# DR3: Value-Based Deep Reinforcement Learning Requires Explicit Regularization

**Aviral Kumar**[1,2]    **Rishabh Agarwal**[2,3]
**Tengyu Ma**[4]    **Aaron Courville**[3]    **George Tucker**[2]    **Sergey Levine**[1,2]
[1] UC Berkeley   [2] Google Research   [3] MILA   [4] Stanford University
`aviralk@berkeley.edu`

## ABSTRACT

Despite overparameterization, deep networks trained via supervised learning are easy to optimize and exhibit excellent generalization. One hypothesis to explain this is that overparameterized deep networks enjoy the benefits of implicit regularization induced by stochastic gradient descent, which favors parsimonious solutions that generalize well on test inputs. It is reasonable to surmise that deep reinforcement learning (RL) methods could also benefit from this effect. In this paper, we discuss how the implicit regularization effect of SGD seen in supervised learning could in fact be harmful in the offline deep RL setting, leading to poor generalization and degenerate feature representations. Our theoretical analysis shows that when existing models of implicit regularization are applied to temporal difference learning, the resulting derived regularizer favors degenerate solutions with excessive "aliasing", in stark contrast to the supervised learning case. We back up these findings empirically, showing that feature representations learned by a deep network value function trained via bootstrapping can indeed become degenerate, aliasing the representations for state-action pairs that appear on either side of the Bellman backup. To address this issue, we derive the form of this implicit regularizer and, inspired by this derivation, propose a simple and effective *explicit* regularizer, called DR3, that counteracts the undesirable effects of this implicit regularizer. When combined with existing offline RL methods, DR3 substantially improves performance and stability, alleviating unlearning in Atari 2600 games, D4RL domains and robotic manipulation from images.

## 1 INTRODUCTION

Deep neural networks are overparameterized, with billions of parameters, which in principle should leave them vulnerable to overfitting. Despite this, supervised learning with deep networks still learn representations that generalize well. A widely held consensus is that deep nets find simple solutions that generalize due to various *implicit* regularization effects [6, 48, 4, 23, 47, 30]. We may surmise that using deep neural nets in reinforcement learning (RL) will work well for the same reason, learning effective representations that generalize due to such implicit regularization effects. But is this actually the case for value functions trained via bootstrapping?

In this paper, we argue that, while implicit regularization leads to effective representations in supervised deep learning, it may lead to poor learned representations when training overparameterized deep network value functions. In order to rule out confounding effects from exploration and non-stationary data distributions, we focus on the offline RL setting – where deep value networks must be trained from a static dataset of experience. There is already evidence that value functions trained via bootstrapping learn poor representations: value functions trained with offline deep RL eventually degrade in performance [2, 28] and this degradation is correlated with the emergence of low-rank features in the value network [28]. Our goal is to understand the underlying cause of the emergence of poor representations during bootstrapping and develop a potential solution. Building on the theoretical framework developed by Blanc et al. [6], Damian et al. [11], we characterize the implicit regularizer that arises when training deep value functions with TD learning. The form of this implicit regularizer implies that TD-learning would co-adapt feature representations at state-action tuples that appear on either side of a Bellman backup.

We show that this theoretically predicted aliasing phenomenon manifests in practice as feature **co-adaptation**, where the features of consecutive state-action tuples learned by the Q-value network

become very similar in terms of their dot product (Section 3). This co-adaptation co-occurs with oscillatory learning dynamics, and training runs that exhibit feature co-adaptation typically converge to poorly performing solutions. Even when Q-values are not overestimated, prolonged training in offline RL can result in performance degradation as feature co-adaptation increases. To mitigate this co-adaptation issue, which arises as a result of implicit regularization, we propose an *explicit regularizer* that we call DR3 (Section 4). While exactly estimating and cancelling the effects of the theoretically derived implicit regularizer is computationally difficult, DR3 provides a simple and tractable theoretically-inspired approximation that mitigates the issues discussed above. In practice, DR3 amounts to regularizing the features at consecutive state-action pairs to be dissimilar in terms of their dot-product similarity. Empirically, we find that DR3 prevents previously noted pathologies such as feature rank collapse [28], gives methods that train for longer and improves performance relative to the base offline RL method employed in practice.

Our first contribution is the derivation of the implicit regularizer that arises when training deep net value functions via TD learning, and an empirical demonstration that it manifests as *feature co-adaptation* in the offline deep RL setting. Feature co-adaptation accounts at least in part for some of the challenges of offline deep RL, including degradation of performance with prolonged training. Second, we propose a simple and effective *explicit* regularizer for offline value-based RL, DR3, which minimizes the feature similarity between state-action pairs appearing in a bootstrapping update. DR3 is inspired by the theoretical derivation of the implicit regularizer, it alleviates co-adaptation and can be easily combined with modern offline RL methods, such as REM [2], CQL [27], and BRAC [49]. Empirically, using DR3 in conjunction with existing offline RL methods provides about **60%** performance improvement on the harder D4RL [18] tasks, and **160%** and **25%** stability gains for REM and CQL, respectively, on offline RL tasks in 17 Atari 2600 games. Additionally, we observe large improvements on image-based robotic manipulation tasks [38].

## 2 PRELIMINARIES

The goal in RL is to maximize the long-term discounted reward in an MDP, defined as $(\mathcal{S}, \mathcal{A}, R, P, \gamma)$ [36], with state space $\mathcal{S}$, action space $\mathcal{A}$, a reward function $R(\mathbf{s}, \mathbf{a})$, dynamics $P(\mathbf{s}'|\mathbf{s}, \mathbf{a})$ and a discount factor $\gamma \in [0, 1)$. The Q-function $Q^\pi(\mathbf{s}, \mathbf{a})$ for a policy $\pi(\mathbf{a}|\mathbf{s})$ is the expected sum of discounted rewards obtained by executing action $\mathbf{a}$ at state $\mathbf{s}$ and following $\pi(\mathbf{a}|\mathbf{s})$ thereafter. $Q^\pi(\mathbf{s}, \mathbf{a})$ is the fixed point of $Q(\mathbf{s}, \mathbf{a}) := R(\mathbf{s}, \mathbf{a}) + \gamma \mathbb{E}_{\mathbf{s}' \sim P(\cdot|\mathbf{s}, \mathbf{a}), \mathbf{a}' \sim \pi(\cdot|\mathbf{s}')}[Q(\mathbf{s}', \mathbf{a}')]$. We study the offline RL setting, where the algorithm must learn a policy only using a given dataset $\mathcal{D} = \{(\mathbf{s}_i, \mathbf{a}_i, \mathbf{s}'_i, r_i)\}$, generated from some behavior policy, $\pi_\beta(\mathbf{a}|\mathbf{s})$, without active data collection. The Q-function is parameterized with a neural net with parameters $\theta$. We will denote the penultimate layer of the deep network (the learned *features*) $\phi_\theta(\mathbf{s}, \mathbf{a})$, such that $Q_\theta(\mathbf{s}, \mathbf{a}) = \mathbf{w}^T \phi(\mathbf{s}, \mathbf{a})$, where $\mathbf{w} \in \mathbb{R}^d$. Standard deep RL methods [33, 24] convert the Bellman equation into a squared temporal difference (TD) error objective for $Q_\theta$:

$$\mathcal{L}_{\text{TD}}(\theta) = \sum_{\mathbf{s}, \mathbf{a}, \mathbf{s}' \sim \mathcal{D}} \left(R(\mathbf{s}, \mathbf{a}) + \gamma \overline{Q}_\theta(\mathbf{s}', \mathbf{a}') - Q_\theta(\mathbf{s}, \mathbf{a})\right)^2, \tag{1}$$

where $\bar{Q}_\theta$ is a delayed copy of same Q-network, referred to as the *target network* and $\mathbf{a}'$ is computed by maximizing the target Q-function at state $\mathbf{s}'$ for Q-learning (i.e., when computing $Q^*$) and by sampling $\mathbf{a}' \sim \pi(\cdot|\mathbf{s})$ when computing the Q-value $Q^\pi$ of a policy $\pi$.

A major problem in offline RL is the issue of distributional shift between the learned policy and the behavior policy [29]. Since our goal is to study the effect of implicit regularization in TD-learning and not distributional shift, we build on top of existing offline RL methods in our experiments: CQL [27], which penalizes erroneous Q-values during training, REM [2], which utilizes an ensemble of Q-functions, and BRAC [49], which applies a policy constraint. An overview of these methods is provided in Appendix E.

## 3 IMPLICIT REGULARIZATION IN DEEP RL VIA TD-LEARNING

While the "deadly-triad" [39] suggests that training value function approximators with bootstrapping off-policy can lead to divergence, modern deep RL algorithms have been able to successfully combine these properties [43]. However, making too many TD updates to the Q-function in offline deep RL is known to sometimes lead to performance degradation and unlearning, even for otherwise effective modern algorithms [17, 16, 2, 28]. Such unlearning is not typically observed when training overparameterized models via supervised learning, so what about TD learning is responsible for it? We show that one possible explanation behind this pathology is the implicit regularization induced

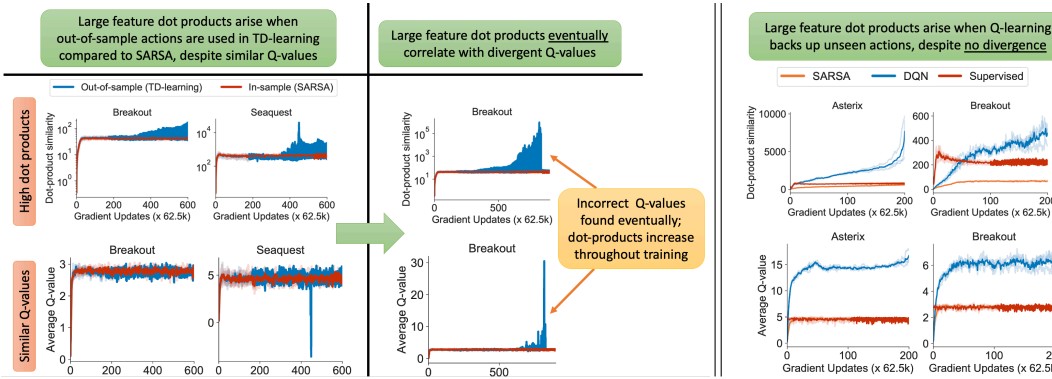

Figure 1: Feature dot-products $\phi(\mathbf{s}, \mathbf{a})^\top \phi(\mathbf{s}', \mathbf{a}')$ increase during training when backing up from *out-of-sample* but in-distribution actions (**TD-learning**: left, **Q-learning**: right), though the average Q-value converges and stays relatively constant. Using only seen state-action pairs for backups (**offline SARSA**) or not performing Bellman backups (i.e., **supervised regression**) avoids this issue, with stable and relatively low dot products. *Left*: TD-learning with high feature dot products eventually destabilizes and produces incorrect Q-values, *Right*: DQN attains extremely large feature dot products, despite a relatively stable trend in Q-values.

by minimizing TD error on a deep Q-network. Our theoretical results suggest that this implicit regularization "co-adapts" the representations of state-action pairs that appear in a Bellman backup (we will define this more precisely below). Empirically, this typically manifests as "co-adapted" features for consecutive state-action tuples, even with specialized TD-learning algorithms that account for distributional shift, and this in turn leads to poor final performance both in theory and in practice. We first provide empirical evidence of this co-adaptation phenomenon in Section 3.1 (additional evidence in Appendix A.1) and then theoretically characterize the implicit regularization in TD learning, and discuss how it can explain the co-adaptation phenomenon in Section 3.2.

## 3.1 Feature Co-Adaptation And How It Relates To Implicit Regularization

In this section, we empirically identify a *feature co-adaptation* phenomenon that appears when training value functions via bootstrapping, where the feature representations of consecutive state-action pairs exhibit a large value of the dot product $\phi(\mathbf{s}, \mathbf{a})^\top \phi(\mathbf{s}', \mathbf{a}')$. Note that feature co-adaptation may arise because of high cosine similarity or because of high feature norms. Feature co-adaptation appears even when there is no explicit objective to increase feature similarity.

**Experimental setup.** We ran supervised regression and three variants of approximate dynamic programming (ADP) on an offline dataset consisting of 1% of uniformly-sampled data from the replay buffer of DQN on two Atari games, previously used in Agarwal et al. [2]. First, for comparison, we trained a Q-function via **supervised regression** to Monte-Carlo (MC) return estimates on the offline dataset to estimate the value of the behavior policy. Then, we trained variants of ADP which differ in the selection procedure for the action $\mathbf{a}'$ that appears in the target value in $\mathcal{L}_{\mathrm{TD}}(\theta)$ (Equation 1). The **offline SARSA** variant aims to estimate the value of the behavior policy, $Q^{\pi_\beta}$, and sets $\mathbf{a}'$ to the actual action observed at the next time step in the dataset, such that $(\mathbf{s}', \mathbf{a}') \in \mathcal{D}$. The **TD-learning** variant also aims to estimate the value of the behavior policy, but utilizes the expectation of the target Q-value over actions $\mathbf{a}'$ sampled from the behavior policy $\pi_\beta$, $\mathbf{a}' \sim \pi_\beta(\cdot|\mathbf{s}')$. We do not have access to the functional form of $\pi_\beta$ for the experiment shown in Figure 1 since the dataset corresponds to the behavior policy induced by the replay buffer of an online DQN, so we train a model for this policy using supervised learning. However, we see similar results comparing **offline SARSA** and **TD-learning** on a gridworld domain where we can access the exact functional form of the behavior policy in Appendix A.6.2. All of the methods so far estimate $Q^{\pi_\beta}$ using different target value estimators. We also train **Q-learning**, which chooses the action $\mathbf{a}'$ to maximize the learned Q-function. While Q-learning learns a different Q-function, we can still compare the relative stability of these methods to gain intuition about the learning dynamics. In addition to feature dot products $\phi(\mathbf{s}, \mathbf{a})^\top \phi(\mathbf{s}', \mathbf{a}')$, we also track the average prediction of the Q-network over the dataset to measure whether the predictions diverge or are stable in expectation.

**Observing feature co-adaptation empirically.** As shown in Figure 1 (right), the average dot product (top row) between features at consecutive state-action tuples continuously increases for both Q-learning and TD-learning (after enough gradient steps), whereas it flatlines and converges to a

small value for supervised regression. We might at first think that this is simply a case of Q-learning failing to converge. However, the bottom row shows that the average Q-values do in fact converge to a stable value. Despite this, the optimizer drives the network towards higher feature dot products. There is no explicit term in the TD error objective that encourages this behavior, indicating the presence of some implicit regularization phenomenon. This *implicit* preference towards maximizing the dot products of features at consecutive state-action tuples is what we call "feature co-adaptation."

**When does feature co-adaptation emerge?** Observe in Figure 1 (right) that the feature dot products for offline SARSA converge quickly and are relatively flat, similarly to supervised regression. This indicates that utilizing a bootstrapped update alone is not responsible for the increasing dot-products and instability, because while offline SARSA uses backups, it behaves similarly to supervised MC regression. Unlike offline SARSA, feature co-adaptation emerges for TD-learning, which is surprising as TD-learning also aims to estimate the value of the behavior policy, and hence should match offline SARSA in expectation. The key difference is that while offline SARSA always utilizes actions $\mathbf{a}'$ observed in the training dataset for the backup, TD-learning may utilize potentially unseen actions $\mathbf{a}'$ in the backup, even though these actions $\mathbf{a}' \sim \pi_\beta(\cdot|\mathbf{s}')$ are *within* the distribution of the data-generating policy. This suggests that utilizing **out-of-sample** actions in the Bellman backup, even when they are not out-of-distribution, critically alters the learning dynamics. This is distinct from the more common observation in offline RL, which attributes training challenges to out-of-distribution actions [29], but not out-of-sample actions. The theoretical model developed in Section 3.2 will provide an explanation for this observation with a discussion about how feature co-adaption caused due to out-of-sample actions can be detrimental in offline RL.

## 3.2  THEORETICALLY CHARACTERIZING IMPLICIT REGULARIZATION IN TD-LEARNING

Why does feature co-adaptation emerge in TD-learning and what do *out-of-sample* actions have to do with it? To answer this question, we theoretically characterize the implicit regularization effects in TD-learning. We analyze the learning dynamics of TD learning in the overparameterized regime, where there are many different parameter vectors $\theta$ that fully minimize the training set temporal difference error. We base our analysis of TD learning on the analysis of implicit regularization in supervised learning, previously developed by Blanc et al. [6], Damian et al. [11].

**Background.** When training an overparameterized $f_\theta(\mathbf{x})$ via supervised regression using the squared loss, denoted by $L$, many different values of $\theta$ will satisfy $L(\theta) = 0$ on the training set due to overparameterization, but Blanc et al. [6] show that the dynamics of stochastic gradient descent will only find fixed points $\theta^*$ that additionally satisfy a condition which can be expressed as $\nabla_\theta R(\theta^*) = 0$, along certain directions (that we will describe shortly). This function $R(\theta)$ is referred to as the implicit regularizer. The noisy gradient updates analyzed in this model have the form:

$$\theta_{k+1} \leftarrow \theta_k - \eta \nabla_\theta L(\theta) + \eta \varepsilon_k, \ \ \varepsilon_k \sim \mathcal{N}(0, M). \tag{2}$$

Blanc et al. [6] and Damian et al. [11] show that some common SGD techniques fall into this framework, for example, when the regression targets in supervised learning are corrupted with $\mathcal{N}(0,1)$ label noise, then the resulting $M = \sum_{i=1}^{|\mathcal{D}|} \nabla_\theta f_\theta(\mathbf{x}_i) \nabla_\theta f_\theta(\mathbf{x}_i)^\top$ and the induced implicit regularizer $R$ is given by $R(\theta) = \sum_i^{|\mathcal{D}|} ||\nabla_\theta f_\theta(\mathbf{x}_i)||_2^2$. Any solution $\theta^*$ found by Equation 2 must satisfy $\nabla_\theta R(\theta^*) = 0$ along directions $\mathbf{v} \in \mathbb{R}^{|\theta|}$ which lie in the null space of the Hessian of the loss $\nabla_\theta^2 L(\theta^*)$ at $\theta^*$, $\mathbf{v} \in \text{Null}(\nabla_\theta^2 L(\theta^*))$. The intuition behind the implicit regularization effect is that along such directions in the parameter space, the Hessian is unable to contract $\theta_k$ when running noisy gradient updates (Equation 2). Therefore, the only condition that the noisy gradient updates converge/stabilize at $\theta^*$ is given by the condition that $\nabla R(\theta^*) = 0$. This model corroborates empirical findings [34, 11] about the solutions found by SGD, which motivates our use.

**Our setup.** Following this framework, we analyze the fixed points of noisy TD-learning. We consider noisy pseudo-gradient (or semi-gradient) TD updates with a general noise covariance $M$:

$$\theta_{k+1} = \theta_k - \eta \underbrace{\left( \sum_i \nabla_\theta Q(\mathbf{s}_i, \mathbf{a}_i) \left( Q_\theta(\mathbf{s}_i, \mathbf{a}_i) - (r_i + \gamma Q_\theta(\mathbf{s}'_i, \mathbf{a}'_i)) \right) \right)}_{:=g(\theta)} + \eta \varepsilon_k, \ \ \varepsilon_k \sim \mathcal{N}(0, M) \tag{3}$$

We use a deterministic policy $\mathbf{a}'_i = \pi(\mathbf{s}'_i)$ to simplify exposition. Following Damian et al. [11], we can set the noise model $M$ as $M = \sum_i \nabla_\theta Q(\mathbf{s}_i, \mathbf{a}_i) \nabla_\theta Q(\mathbf{s}_i, \mathbf{a}_i)^\top$, or utilize a different choice of

$M$, but we will derive the general form first. Let $\theta^*$ denote a stationary point of the training TD error, such that the pseudo-gradient $g(\theta^*) = 0$. Further, we denote the derivative of $g(\theta)$ w.r.t. $\theta$ as the matrix $G(\theta) \in \mathbb{R}^{|\theta| \times |\theta|}$, and refer to it as the *pseudo-Hessian*: although $G(\theta)$ is not actually the second derivative of any well-defined objective, since TD updates are not proper gradient updates, as we will see it will play a similar role to the Hessian in gradient descent. For brevity, define $G = G(\theta^*)$, $g = g(\theta^*)$, $\nabla G = \nabla_\theta G(\theta^*) \in \mathbb{R}^{|\theta| \times |\theta| \times |\theta|}$, and let $\lambda_i(P)$ denote the $i$-th eigenvalue of matrix $P$, when arranged in decreasing order of its (complex) magnitude $|\lambda_i(P)|$ (note that an eigenvalue can be complex).

**Assumptions.** To simplify analysis, we assume that matrices $G$ and $M$ (i.e., the noise covariance matrix) span the same $n$-dimensional basis in $d$-dimensional space, where $d$ is the number of parameters and $n$ is the number of datapoints, and $n \ll d$ due to overparameterization. We also require $\theta^*$ to satisfy a technical criterion that requires approximate alignment between the eigenspaces of $G$ and the gradient of the Q-function, without which noisy TD may not be stable at $\theta^*$. We summarize all the assumptions in Appendix C, and present the resulting regularizer below.

**Theorem 3.1** (Implicit regularizer at TD fixed points). *Under the assumptions so far, a fixed point of TD-learning, $\theta^*$, where $Q_{\theta^*}(\mathbf{s}_i, \mathbf{a}_i) = r_i + \gamma Q_{\theta^*}(\mathbf{s}_i', \mathbf{a}_i')$ for every $(\mathbf{s}_i, \mathbf{a}_i, \mathbf{s}_i') \in \mathcal{D}$ is stable (atttractive) if: (1) it satisfies $\mathrm{Re}(\lambda_i(G)) \geq 0, \forall i$ and $\mathrm{Re}(\lambda_i(G)) > 0$ if $|\mathrm{Imag}(\lambda_i(G))| > 0$, and (2) along directions $\mathbf{v} \in \mathbb{R}^{dim(\theta)}, \mathbf{v} \in Null(G)$, $\theta^*$ is the stationary point of the implicit regularizer:*

$$R_{\mathrm{TD}}(\theta) = \eta \underbrace{\sum_{i=1}^{|\mathcal{D}|} \nabla Q_\theta(\mathbf{s}_i, \mathbf{a}_i)^\top \Sigma_M^* \nabla Q_\theta(\mathbf{s}_i, \mathbf{a}_i)}_{\text{implicit regularizer for noisy GD in supervised learning}} - \eta\gamma \underbrace{\sum_{i=1}^{|\mathcal{D}|} \mathrm{tr}\left( \left[ [\nabla Q_\theta(\mathbf{s}_i', \mathbf{a}_i')^\top] \right]^\top \Sigma_M^* \nabla Q_\theta(\mathbf{s}_i, \mathbf{a}_i) \right)}_{\text{additional term in TD learning}},$$

(4)

*where $(\mathbf{s}_i, \mathbf{a}_i)$ and $(\mathbf{s}_i', \mathbf{a}_i')$ denote state-action pairs that appear together in a Bellman update, $[[\square]]$ denotes the stop-gradient function, which does not pass partial gradients w.r.t. $\theta$ into $\square$. $\Sigma_M^*$ is the fixed point of the discrete Lyapunov equation: $\Sigma_M^* := (I - \eta G)\Sigma_M^*(I - \eta G)^\top + \eta^2 M$.*

A proof of Theorem 3.1 is provided in Appendix C. Next, we explain the intuition behind this result and provide a proof sketch. To derive the induced implicit regularizer for a stable fixed point $\theta^*$ of TD error, we study the learning dynamics of noisy TD learning (Equation 3) initialized at $\theta^*$, and derive conditions under which this noisy update would stay close to $\theta^*$ with multiple updates. This gives rise to the two conditions shown in Theorem 3.1 which can be understood as controlling stability in mutually exclusive directions in the parameter space. If condition **(1)** is not satisfied, then even under-parameterized TD will diverge away from $\theta^*$, since $I - \eta G$ would be a non-contraction as the spectral radius, $\rho(I - \eta G) \geq 1$ in that case. Thus, $\theta_k - \theta^*$ will grow or not decrease in some direction. When **(1)** is satisfied for all directions in the parameter space, there are still directions where both the real and imaginary parts of the eigenvalue $\lambda_i(G)$ are 0 due to overparameterization[1]. In such directions, learning is governed by the projection of the noise under the tensor $\nabla G$, which appears in the Taylor expansion of $\theta_k - \theta^*$ around the point $\theta^*$:

$$\theta_{k+1} = \theta_k - \eta \left( g + G(\theta_k - \theta^*) + \frac{1}{2}\nabla G[\theta_k - \theta^*, \theta_k - \theta^*] \right) + \varepsilon_k, \quad \varepsilon_k \sim \mathcal{N}(0, M) \quad (5)$$

$$\implies \nu_{k+1} = (I - \eta G)\nu_k - \frac{\eta}{2}\nabla G[\nu_k, \nu_k] + \varepsilon_k, \quad (6)$$

where we reparameterize in terms of $\nu_k := \theta_k - \theta^*$. The proof shows that $\theta^*$ is stable if it is a stationary point of the implicit regularizer $R_{\mathrm{TD}}$ (condition **(2)**), which ensures that total noise (i.e., accumulated $\varepsilon_k$ over iterations $k$) accumulated by $\nabla G$ does not lead to a large deviation in $\nu_k$ in directions where $I - \eta G$ does not contract.

**Interpretation of Theorem 3.1.** While the choice of the noise model $M$ will change the form of the implicit regularizer, in practice, the form of $M$ is not known as this corresponds to the noise induced via SGD. We can consider choices of $M$ for interpretation, but Theorem 3.1 is easy to qualitatively interpret for $M$ such that $\Sigma_M^* = I$. In this case, we find that the implicit preference towards local minima of $R_{\mathrm{TD}}(\theta)$ can explain feature co-adaptation. In this case, the regularizer is simpler:

$$R_{\mathrm{TD}}(\theta) := \sum_i ||\nabla Q_\theta(\mathbf{s}_i, \mathbf{a}_i)||_2^2 - \gamma \nabla Q_\theta(\mathbf{s}_i, \mathbf{a}_i)\nabla[[Q_\theta(\mathbf{s}_i', \mathbf{a}_i')]].$$

---

[1]To see why this is the case, note that $\mathrm{rank}(G) \leq |\mathcal{D}| \ll \dim(\theta)$, and so some eigenvalues of $G$ are 0.

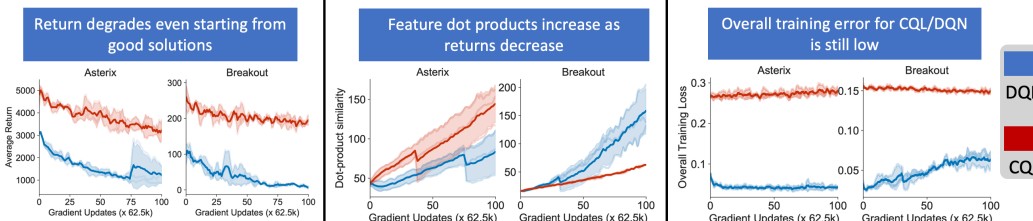

Figure 2: Even when current offline RL algorithms are initialized at a high-performing checkpoint that attains small feature dot products, feature dot products increase with further training and the performance degrades.

The first term is equal to the squared per-datapoint gradient norm, which is same as the implicit regularizer in supervised learning obtained by Blanc et al. [6], Damian et al. [11] with label noise. However, $R_{\mathrm{TD}}(\theta)$ additionally includes a second term that is equal to the dot product of the gradient of the Q-function at the current and next states, $\nabla_\theta Q_\theta(\mathbf{s}_i, \mathbf{a}_i)^\top \nabla_\theta Q_\theta(\mathbf{s}'_i, \mathbf{a}'_i)$, and thus this term is effectively *maximized*. When restricted to the last-layer parameters of a neural network, this term is equal to the dot product of the features at consecutive state-action tuples: $\sum_i \nabla_\theta Q_\theta(\mathbf{s}_i, \mathbf{a}_i)^\top \nabla_\theta Q_\theta(\mathbf{s}'_i, \mathbf{a}'_i) = \sum_i \phi(\mathbf{s}_i, \mathbf{a}_i)^\top \phi(\mathbf{s}'_i, \mathbf{a}'_i)$. The tendency to maximize this quantity to attain a local minimizer of the implicit regularizer corroborates the empirical findings of increased dot product in Section 3.1.

**Explaining the difference between utilizing seen and unseen actions in the backup.** If all state-action pairs $(\mathbf{s}'_i, \mathbf{a}'_i)$ appearing on the right-hand-side of the Bellman update also appear in the dataset $\mathcal{D}$, as in the case of offline SARSA (Figure 1), the preference to increase dot products will be balanced by the affinity to reduce gradient norm (first term of $R_{\mathrm{TD}}(\theta)$ when $\Sigma_M^* = I$): for example, for offline SARSA, when $(\mathbf{s}'_i, \mathbf{a}'_i)$ are permutations of $(\mathbf{s}_i, \mathbf{a}_i)$, $R_{\mathrm{TD}}$ is lower bounded by $(1 - \gamma) \sum_i ||\nabla_\theta Q_\theta(\mathbf{x}_i)||_2^2$ and hence minimizing $R_{\mathrm{TD}}(\theta)$ would minimize the feature norm instead of maximizing dot products. This also corresponds to the implicit regularizer we would obtain when training Q-functions via supervised learning and hence, our analysis predicts that offline SARSA with in-sample actions (i.e., when $(\mathbf{s}', \mathbf{a}') \in \mathcal{D}$) would behave similarly to supervised regression.

However, the regularizer behaves very differently when unseen state-action pairs $(\mathbf{s}'_i, \mathbf{a}'_i)$ appear only on the right-hand-side of the backup. This happens with any algorithm where $\mathbf{a}'$ is not the dataset action, which is the case for all deep RL algorithms that compute target values by selecting $\mathbf{a}'$ according to the current policy. In this case, we expect the dot product of gradients at $(\mathbf{s}, \mathbf{a})$ and $(\mathbf{s}', \mathbf{a}')$ to be large at any attractive fixed point, since this minimizes $R_{\mathrm{TD}}(\theta)$. This is precisely a form of co-adaptation: *gradients at out-of-sample state-action tuples are highly similar to gradients at observed state-action pairs measured by the dot product*. This observation is also supported by the analysis in Section 3.1. Finally, note that the choice of $M$ is a modelling assumption, and to derive our explicit regularizer, later in the paper, we will make a simplifying choice of $M$. However, we also empirically verify that a different choice of $M$, given by label noise, works well.

**Why is implicit regularization detrimental to policy performance?** To answer this question, we present theoretical and empirical evidence that illustrates the adverse effects of this implicit regularizer. Empirically, we ran two algorithms, DQN and CQL, initialized from a high-performing Q-function checkpoint, which attains relatively small feature dot products (i.e., the second term of $R_{\mathrm{TD}}(\theta)$ is small). Our goal is to see if TD updates starting from such a "good" initialization still stay around it or diverge to poorer solutions. Our theoretical analysis in Section 3.2 would predict that TD learning would destabilize from such a solution, since it would not be a stable fixed point. Indeed, as shown in Figure 2, the policy immediately degrades, and the the dot-product similarities start to increase. This even happens with CQL, which explicitly corrects for distributional shift confounds, implying that the performance drop cannot be directly explained by the typical out-of-distribution action explanations. To investigate the reasons behind this drop, we also measured the training loss function values for these algorithms (i.e., TD error for DQN and TD error + CQL regularizer for CQL) and find in Figure 2 that the loss values are generally small for both CQL and DQN. This indicates that the preference to increase dot products is not explained by an inability to minimize TD error. In Appendix A.7, we show that this drop in performance when starting from good solutions can be effectively mitigated with our proposed DR3 explicit regularizer for both DQN and CQL. Thus we find that not only standard TD learning degrades from a good solution in favor of increasing feature dot products, but keeping small dot products enables these algorithms to remain stable near the good solution.

To motivate why co-adapted features can lead to poor performance in TD-learning, we study the convergence of linear TD-learning on co-adapted features. Our theoretical result characterizes a lower bound on the feature dot products in terms of the feature norms for state-action pairs in the dataset $\mathcal{D}$, which if satisfied, will inhibit convergence:

**Proposition 3.2** (TD-learning on co-adapted features). *Assume that the features* $\Phi = [\phi(\mathbf{s}, \mathbf{a})]_{\mathbf{s}, \mathbf{a}}$ *are used for linear TD-learning. Then, if* $\sum_{\mathbf{s}, \mathbf{a}, \mathbf{s}' \in \mathcal{D}} \phi(\mathbf{s}, \mathbf{a})^\top \phi(\mathbf{s}', \mathbf{a}') \geq \frac{1}{\gamma} \sum_{\mathbf{s}, \mathbf{a} \in \mathcal{D}} \phi(\mathbf{s}, \mathbf{a})^\top \phi(\mathbf{s}, \mathbf{a})$, *linear TD-learning using features* $\Phi$ *will not converge.*

A proof of Proposition 3.2 is provided in Appendix D and it relies on a stability analysis of linear TD. While features change during training for TD-learning with neural networks, and arguably linear TD is a simple model to study consequences of co-adapted features, even in this simple linear setting, Proposition 3.2 indicates that TD-learning may be non-convergent as a result of co-adaptation.

## 4 DR3: EXPLICIT REGULARIZATION FOR DEEP TD-LEARNING

Since the implicit regularization effects in TD-learning can lead to feature co-adaptation, which in turn is correlated with poor performance, can we instead derive an *explicit* regularizer to alleviate this issue? Inspired by the analysis in the previous section, we will propose an *explicit* regularizer that attempts to counteract the second term in Equation 4, which would otherwise lead to co-adaptation and poor representations. The *explicit* regularizer that offsets the difference between the two implicit regularizers is given by: $\Delta(\theta) = \sum_i \text{trace} \left[ \Sigma_M^{*\top} \nabla_\theta Q_\theta(\mathbf{s}_i, \mathbf{a}_i) \nabla_\theta Q_\theta(\mathbf{s}_i', \mathbf{a}_i')^\top \right]$, which represents the second term of $R_{\text{TD}}(\theta)$. Note that we drop the stop gradient on $Q_\theta(\mathbf{s}_i', \mathbf{a}_i')$ in $\Delta(\theta)$, as it performs slightly better in practice (Table A.1), although as shown in that Table, the version with the stop gradient also significantly improves over the base method. The first term of $R_{\text{TD}}(\theta)$ corresponds to the regularizer from supervised learning. Our proposed method, DR3, simply combines approximations to $\Delta(\theta)$ with various offline RL algorithms. For any offline RL algorithm, ALG, with objective $\mathcal{L}_{\text{ALG}}(\theta)$, the training objective with DR3 is given by: $\mathcal{L}(\theta) := \mathcal{L}_{\text{ALG}}(\theta) + c_0 \Delta(\theta)$, where $c_0$ is the DR3 coefficient. See Appendix E.3 for details on how we tune $c_0$ in this paper.

**Practical version of DR3.** In order to practically instantiate DR3, we need to choose a particular noise model $M$. In general, it is not possible to know beforehand the "correct" choice of $M$ (Equation 3), even in supervised learning, as this is a complicated function of the data distribution, neural network architecture and initialization. Therefore, we instantiate DR3 with two heuristic choices of $M$: **(i)** $M$ induced by label noise studied in prior work for supervised learning and for which we need to run a computationally heavy fixed-point computation for $M$, and **(ii)** a simpler alternative that sets $\Sigma_M^* = I$. We find that both of these variants generally perform well empirically (Figure 6), and improve over the base offline RL method, and so we utilize **(ii)** in practice due to low computational costs. Additionally, because computing and backpropagating through per-example gradient dot products is slow, we instead approximate $\Delta(\theta)$ with the contribution only from the last layer parameters (*i.e.*, $\sum_i \nabla_\mathbf{w} Q_\theta(\mathbf{s}_i, \mathbf{a}_i)^\top \nabla_\mathbf{w} Q_\theta(\mathbf{s}_i', \mathbf{a}_i')$), similarly to tractable Bayesian neural nets. As shown in Appendix A.5, the practical version of DR3 performs similarly to the label-noise version.

$$\textbf{Explicit DR3 regularizer}: \qquad \overline{\mathcal{R}}_{\text{exp}}(\theta) = \sum_{i \in \mathcal{D}} \phi(\mathbf{s}_i, \mathbf{a}_i)^\top \phi(\mathbf{s}_i', \mathbf{a}_i'). \qquad (7)$$

## 5 EXPERIMENTAL EVALUATION OF DR3

Our experiments aim to evaluate the extent to which DR3 improves performance in offline RL in practice, and to study its effect on prior observations of rank collapse. To this end, we investigate if DR3 improves offline RL performance and stability on three offline RL benchmarks: Atari 2600 games with discrete actions [2], continuous control tasks from D4RL [18], and image-based robotic manipulation tasks [38]. Following prior work [18, 22], we evaluate DR3 in terms of final offline RL performance after a given number of iterations. Additionally, we report *training stability*, which is important in practice as offline RL does not admit cheap validation of trained policies for model selection. To evaluate stability, we train for a large number of gradient steps (2-3x longer than prior work) and either report the **average performance** over the course of training or the final performance at the end of training. We expect that a stable method that does not unlearn with more gradient steps, should have better average performance, as compared to a method that attains good peak performance but degrades with more training. See Appendix E for further details.

**Offline RL on Atari 2600 games.** We compare DR3 to prior offline RL methods on a set of offline Atari datasets of varying sizes and quality, akin to Agarwal et al. [2], Kumar et al. [28]. We evaluated

Table 1: IQM normalized average performance (training stability) across 17 games, with 95% CIs in parenthesis, after 6.5M gradient steps for the 1% setting and 12.5M gradient steps for the 5%, 10% settings. Individual performances reported in Tables F.4-F.9. DR3 improves the stability over both CQL and REM.

| Data | CQL | CQL + DR3 | REM | REM + DR3 |
|------|-----|-----------|-----|-----------|
| 1% | 43.7 (39.6, 48.6) | **56.9** (52.5, 61.2) | 4.0 (3.3, 4.8) | **16.5** (14.5, 18.6) |
| 5% | 78.1 (74.5, 82.4) | **105.7** (101.9, 110.9) | 25.9 (23.4, 28.8) | **60.2** (55.8, 65.1) |
| 10% | 59.3 (56.4, 61.9) | **65.8** (63.3, 68.3) | 53.3 (51.4, 55.3) | **73.8** (69.3, 78) |

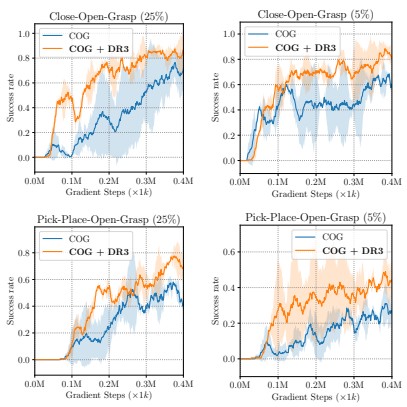

Figure 3: **Performance of DR3 + COG** on two manipulation tasks using only 5% and 25% of the data used by Singh et al. [38] to make these more challenging. COG + DR3 outperforms COG in training and attains higher average and final performance.

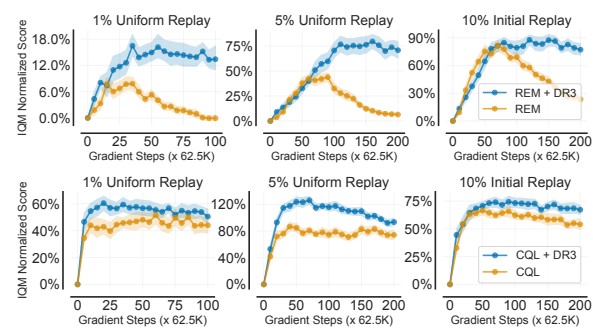

Figure 4: **Normalized performance across 17 Atari games for REM + DR3 (top), CQL + DR3 (bottom)**. x-axis represents *gradient steps*; no new data is collected. While naïve REM suffers from a degradation in performance with more training, REM + DR3 not only remains generally stable with more training, but also attains higher final performance. CQL + DR3 attains higher performance than CQL. We report IQM with 95% stratified bootstrap CIs [3].

on three datasets: **(1)** 1% and 5% samples drawn uniformly at random from DQN replay; **(2)** a dataset with more suboptimal data consisting of the first 10% samples observed by an online DQN. Following Agarwal et al. [3], we report the interquartile mean (IQM) normalized scores across 17 games over the course of training in Figure 4 and report the IQM average performance in Table 1. Observe that combining DR3 with modern offline RL methods (CQL, REM) attains the best final and average performance across the 17 Atari games tested on, directly improving upon prior methods across all the datasets. When DR3 is used in conjunction with REM, it prevents severe unlearning and performance degradation with more training. CQL + DR3 improves by **20%** over CQL on final performance and attains **25%** better average performance. While DR3 is not unequivocally "stable", as its performance also degrades relative to the peak it achieves (Figure 4), it is more stable relative to base offline RL algorithms. We also compare DR3 to the srank($\Phi$) penalty proposed to counter rank collapse [28]. Directly taking median normalized score improvements reported by Kumar et al. [28], CQL + DR3 improves by over **2x** (31.5%) over naïve CQL relative to the srank penalty (14.1%), indicating DR3's efficacy.

**Offline RL on robotic manipulation from images.** Next, we aim to evaluate the efficacy of DR3 on two image-based robotic manipulation tasks [38] (visualized on the right) that require composition of skills (e.g., opening a drawer, closing a drawer, picking an obstructive object, placing an object, etc.) over extended horizons using only a sparse 0-1 reward. As shown in Figure 3, combining DR3 with COG improves over COG.

**Offline RL on D4RL tasks.** Finally, we evaluate DR3 in conjunction with CQL on the antmaze-v2 domain in D4RL [18]. To assess if DR3 is stable and able to prevent unlearning that eventually appears in CQL, we trained CQL+DR3 for **9x** longer: 2M and 3M steps with 3x higher learning rate. This is different from prior works [18] that report performance at the end of 1M steps. Observe in Table 2, that CQL + DR3 outperforms CQL (statistical significance shown in Appendix A.8), indicating that DR3 significantly improves CQL. We also evalute DR3 on kitchen domains in D4RL in Appendix A.8, where we also find that DR3 improves CQL. Finally, we also compare CQL+DR3 and CQL in terms of performance and stability on MuJoCo tasks previously studied in Kumar et al.

Table 2: **Performance of CQL, CQL + DR3 after 2M and 3M gradient steps with a learning rate of 3e-4** for the Q-function averaged over 3 seeds. This is training for **6x** and **9x** longer compared to CQL defaults. Observe that CQL + DR3 outperforms CQL at 2M and 3M steps, indicating is efficacy in preventing unlearning. We present the statistical significance of these results in Appendix A.8.

| D4RL Task | CQL (2M) | CQL + DR3 (2M) | CQL (3M) | CQL + DR3 (3M) |
|---|---|---|---|---|
| antmaze-umaze-v2 | $84.00 \pm 2.67$ | $85.33 \pm 4.16$ | $87.00 \pm 1.73$ | $90.00 \pm 4.00$ |
| antmaze-umaze-diverse-v2 | $45.67 \pm 8.50$ | $40.67 \pm 11.84$ | $36.33 \pm 7.09$ | $\mathbf{52.00 \pm 11.26}$ |
| antmaze-medium-play-v2 | $24.00 \pm 28.16$ | $\mathbf{73.00 \pm 4.00}$ | $16.00 \pm 26.85$ | $\mathbf{71.33 \pm 1.52}$ |
| antmaze-medium-diverse-v2 | $32.67 \pm 9.29$ | $\mathbf{67.00 \pm 2.00}$ | $48.33 \pm 6.11$ | $\mathbf{61.67 \pm 3.21}$ |
| antmaze-large-play-v2 | $3.33 \pm 2.51$ | $\mathbf{28.00 \pm 4.35}$ | $0.33 \pm 0.57$ | $\mathbf{26.33 \pm 11.93}$ |
| antmaze-large-diverse-v2 | $1.33 \pm 2.30$ | $\mathbf{25.67 \pm 0.57}$ | $0.00 \pm 0.00$ | $\mathbf{28.33 \pm 1.52}$ |

[28] in Appendix A.3. These tasks are constructed by uniformly subsampling transitions from the full-replay-v2 MuJoCo datasets in D4RL and are much harder than the typical Gym-MuJoCo tasks from Fu et al. [18] because succeeding on these tasks critically relies on estimating accurate Q-values for out-of-sample actions and all actions at certain states are out-of-sample. As shown in Appendix A.3, CQL+DR3 is significantly more stable, and does not unlearn with more training, unlike CQL whose performance degrades very quickly. We also evaluate DR3 in conjunction with BRAC [49], a policy constraint method, and find that BRAC+DR3 improves over BRAC in **13.8** median normalized performance (Table F.2).

**DR3 does not suffer from rank collapse.** Prior work [28] has shown that implicit regularization can lead to a rank collapse issue in TD-learning, preventing Q-networks from using full capacity. To see if DR3 addresses the rank collapse issue, we follow Kumar et al. [28] and plot the effective rank of learned features with DR3 in Figure 5 (DQN, REM in Appendix A.4). While the value of the effective rank decreases during training with naïve bootstrapping, we find that rank of DR3 features typically does not collapse, despite no explicit term encouraging this. Finally, we test the

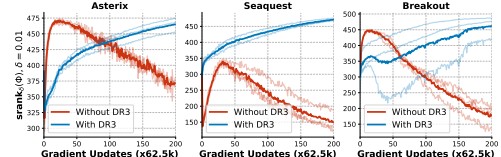

Figure 5: **Trend of effective rank,** $\mathrm{srank}(\Phi)$ of features $\Phi$ learned by the Q-function when trained with TD error (red, "Without DR3") and with TD error + DR3 (blue, "With DR3") on three Atari games using the 5% dataset. Note that DR3 alleviates rank collapse, without explicitly aiming to.

robustness/sensitivity of each layer in the learned Q-network to re-initialization [52] during training and find that DR3 alters the features to behave similarly to supervised learning (Figure A.2).

**Comparing explicit regularizers for different choices of noise covariance $M$.** Finally, we investigate the behavior of different implicit regularizers derived via two choices of $M$ in Equation 4 and the corresponding explicit regularizers. While the explicit regularizer we use in practice is a simplifying choice that works well, another choice of $M$ is the covariance matrix induced by label noise, which requires explicit computation of $\Sigma_M^*$. Observe in Figure 6 that the explicit regularizer for our simplifying choice is not worse than the different choice of $M$. This justifies utilizing our simplified, heuristic choice of setting $\Sigma_M^* = I$ in practice. Results on five Atari games are shown in Appendix A.5.

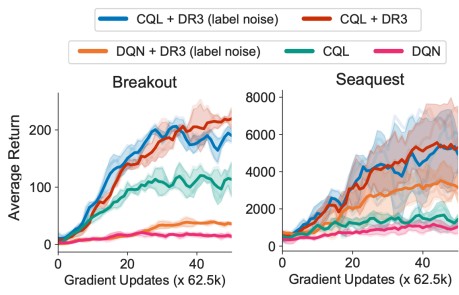

Figure 6: Comparing DR3 regularizers for our simplifying choice of $M$ and $M$ induced by label noise, with base CQL and DQN algorithms. Note that both of these penalties when applied over CQL improve performance.

**Discussion.** We characterized the implicit preference of TD-learning towards solutions that maximally co-adapt gradients (or features) at consecutive state-action tuples that appear in Bellman backup. This regularization effect is exacerbated when out-of-sample state-action samples are used for the Bellman backup and it can lead to poor policy performance. Inspired by the theory, we propose a practical explicit regularizer, DR3, that yields substantial improvements in stability and performance on a wide range of offline RL problems. We believe that understanding the learning dynamics of deep Q-learning will lead to more robust and stable deep RL algorithms and enable predicting such instability issues, well in advance, which can inspire cross-validation and model selection strategies. This is an important, open challenge in offline RL, for which existing off-policy evaluation techniques are not practically sufficient [19].

## ACKNOWLEDGEMENTS

We thank Dibya Ghosh, Xinyang Geng, Dale Schuurmans, Marc Bellemare, Pablo Castro and Ofir Nachum for informative discussions, discussions on experimental setup and for providing feedback on an early version of this paper. We thank the members of RAIL at UC Berkeley for their support and suggestions. We thank anonymous reviewers for feedback on an early version of this paper. This research is funded in part by the DARPA Assured Autonomy Program and in part, by compute resources from Microsoft Azure and Google Cloud. TM acknowledges support of Google Faculty Award, NSF IIS 2045685, the Sloan fellowship, and JD.com.

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

# Appendices

## A  ADDITIONAL VISUALIZATIONS AND EXPERIMENTS FOR DR3

In this section, we provide visualizations and diagnostic experiments evaluating various aspects of feature co-adaptation and the DR3 regularizer. We first provide more empirical evidence showing the presence of feature co-adaptation in modern deep offline RL algorithms. We will also visualize DR3 inspired from the implicit regularizer term in TD-learning alleviates rank collapse discussed in Kumar et al. [28]. We will compare the efficacies of the explicit regularizer induced for different choices of the noise covariance matrix $M$ (Equation 4), understand the effect of dropping the stop gradient term in ou practical regularizer and finally, perform diagnostic experiments visualizing if the Q-networks learned with DR3 resemble more like neural networks trained via supervised learning, measured in terms of sensitivity and robustness to layer reinitialization [52].

### A.1  MORE EMPIRICAL EVIDENCE OF FEATURE CO-ADAPTATION

In this section, we provide more empirical evidence demonstrating the existence of the feature co-adaptation issue in modern offline RL algorithms such as DQN and CQL. As shown below in Figure A.1, while the average dataset Q-value for both CQL and DQN exhibit a flatline trend, the dot product similarity for consecutive state-action tuples generally continues to increase throughout training and does not flatline. While DQN eventually diverges in Seaquest, the dot products increase with more gradient steps even before divergence starts to appear.

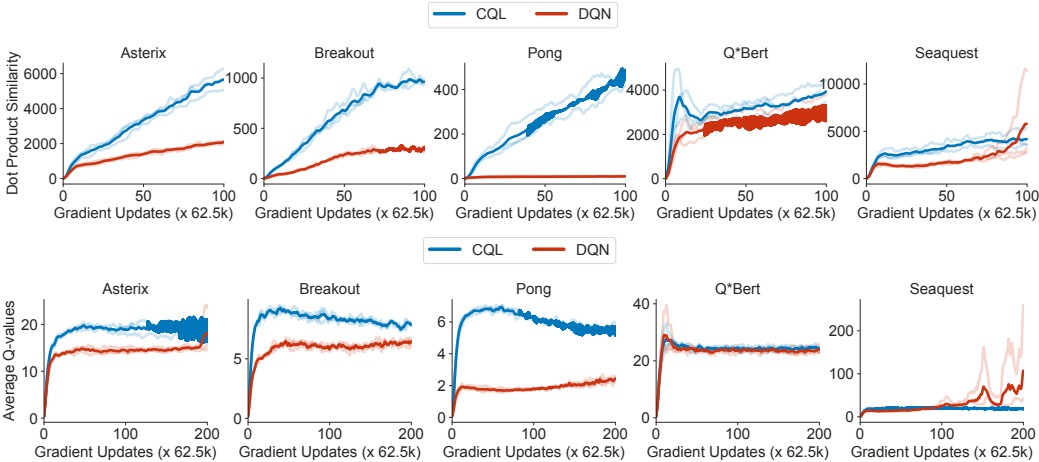

Figure A.1: **Demonstrating feature co-adaptation on five Atari games with standard offline DQN and CQL, averaged over 3 seeds.** Observe that the feature dot products continue to rise with more training for both CQL and DQN, indicating the presence of co-adaptation. On the other hand, the average Q-values exhibit a converged trend, except on Seaquest. Further, note that the dot products continue to increase for CQL even though CQL explicitly corrects for out-of-distribution action inputs.

### A.2  LAYER-WISE STRUCTURE OF A Q-NETWORK TRAINED WITH DR3

To understand if DR3 indeed makes Q-networks behave as if they were trained via supervised learning, utilizing the empirical analysis tools from Zhang et al. [52], we test the robustness/sensitivity of each layer in the learned network to re-initialization, while keeping the other layers fixed. This tests if a particular layer is *critical* to the predictions of the learned neural network and enables us to reason about generalization properties [52, 8]. We ran CQL and REM and saved all the intermediate checkpoints. Then, as shown in Figure A.2, we first loaded a checkpoint ($x$-axis), and computed policy performance (shaded color; colorbar) by re-initializing a given layer ($y$-axis) of the network to its initialization value before training for the same run.

Note in Figure A.2, that while almost all layers are absolutely critical for the base CQL algorithm, utilizing DR3 substantially reduces sensitivity to the latter layers in the Q-network over the course of training. This is similar to what Zhang et al. [52] observed for supervised learning, where the initial

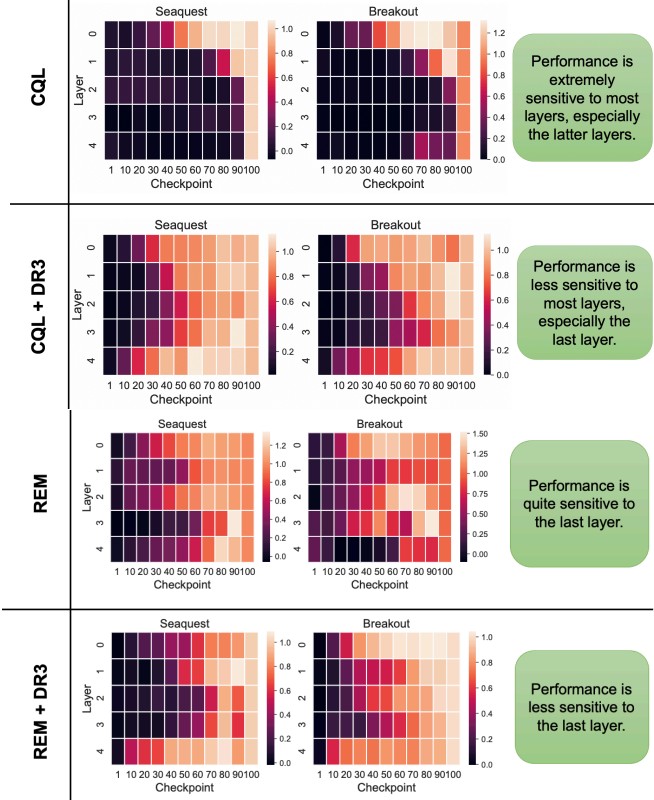

Figure A.2: **CQL vs CQL + DR3 and REM vs REM + DR3.** Average robustness of the learned Q-function to re-initialization of all layers to different checkpoints over the course of training created based on the protocol from Zhang et al. [52]. The colors in the heatmap indicate performance of the reinitialized checkpoint, normalized w.r.t. the checkpoint without any change to layers. Note that while CQL and REM are more sensitive (i.e., less robust) to reinitialization of all the layers especially the last layer, CQL + DR3 and REM + DR3 behave closer to supervised learning, in the sense that they are more robust to reinitialization of layers of the network, especially the last layer.

layers of a network were the most critical, and the latter layers primarily performed near-random transformations without affecting the performance of the network. This indicates that utilizing DR3 alters the internal layers of a Q-network trained with TD to behave closer to supervised learning.

## A.3 RESULTS ON MUJOCO DOMAINS

In this section, we provide the results of applying DR3 on the MuJoCo tasks shown in Figure A.5. (Appendix A) of Kumar et al. [28]. To briefly describe the setup, in these tasks we train on the three gym tasks (Hopper-v2, Ant-v2, Walker2d-v2) using 20% of the offline data, uniformly subsampled from the run of an online SAC agent, mimicking the setup from Kumar et al. [28]. Rather than retraining an SAC agent to collect data, we subsampled the Gym-MuJoCo `*-full-replay-v2` replay buffers from the latest D4RL [18]. In these cases we plot the $\text{srank}_\delta$ values, the feature dot products and the corresponding performance values with and without the DR3 regularizer for 4M steps (Kumar et al. [28] showed their plots for just under 4M steps) in Figure A.3.

Observe in Figure A.3, that while the standard CQL algorithm performs poorly and suffers from performance degradation within about 1M-1.5M steps for Walker2d and Ant, CQL + DR3 is able to prevent the performance degradation and trains stably. Base CQL demonstrates oscillatory performance on Hopper, but CQL + DR3 stabilizes the performace of CQL. This indicates that DR3 is effective on MuJoCo domains, and prevents the instabilities with CQL.

For details, the weight on the CQL regularizer in this case is equal to 5.0 across all the tasks, and weight on the DR3 regularizer is 0.01. We also attempted to tune the CQL coefficient for the baseline CQL algorithm within $\{1.0, 2.0, 5.0, 10.0, 20.0\}$ to see if it address the performance degradation

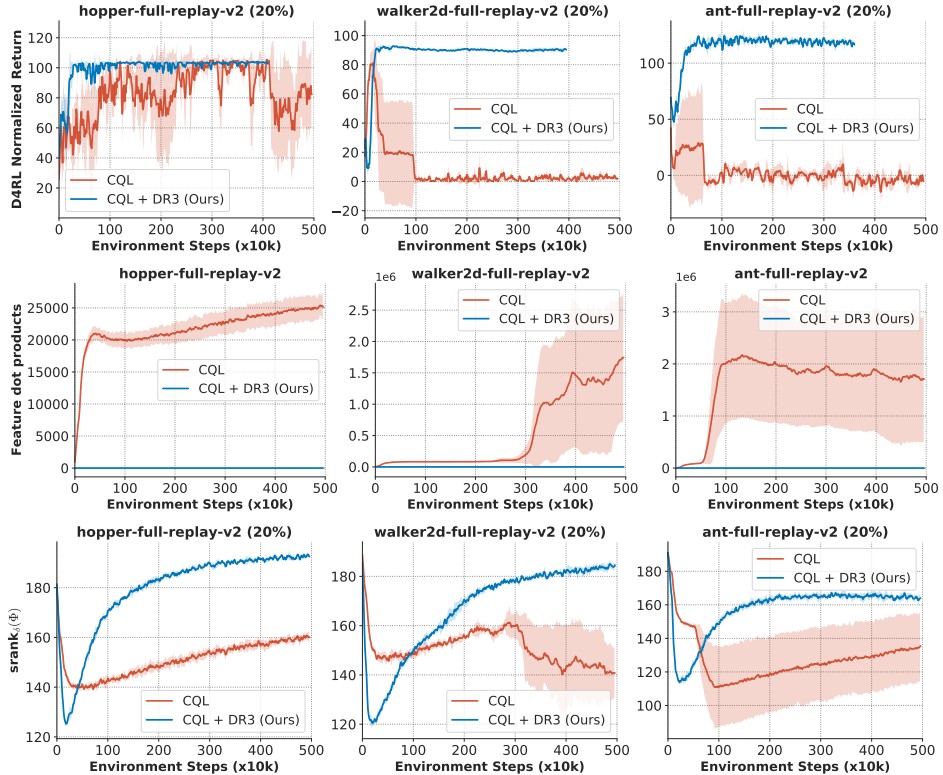

Figure A.3: **Comparison of CQL and CQL + DR3 on the offline MuJoCo Gym domains**, mimicking the setup of Kumar et al. [28]. The data is generated by randomly sampling 20% of the transitions of the D4RL [18] full-replay-v2 datasets, which are collected via the run of an online SAC agent. The performance is shown in terms of the D4RL normalized score, where 0.0 denotes the performance of a random policy and 100.0 denotes the performance of an expert online SAC policy. Observe that adding DR3 stabilizes the performance on Hopper, and prevents performance collapse on Walker2d and Ant. In addition note that the srank values attained by CQL + DR3 is higher than base CQL and more importantly, the feature dot products are much smaller for CQL + DR3 compared to CQL.

issues, but did not find any difference in the collapsing behavior of base CQL. Our CQL baseline is therefore well-tuned, and DR3 improves the performance over this baseline.

## A.4 RANK COLLAPSE IS ALLEVIATED WITH DR3

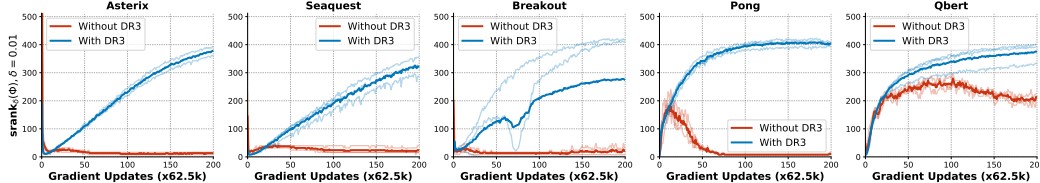

Figure A.4: **Comparing the feature ranks for CQL and CQL + DR3.** Observe that utilizing DR3 successfully alleviates the rank collapse issue noted in prior work without explicitly correcting for it.

Prior work [28] has shown that implicit regularization in TD-learning can lead to a feature rank collapse phenomenon in the Q-function, which hinders the Q-function from using its full representational capacity. Such a phenomenon is absent in supervised learning, where the feature rank does not collapse. Since DR3 is inspired by mitigating the effects of the term in the implicit regularizer (Equation 4) that only appears in the case of TD-learning, we wish to understand if utilizing DR3 also alleviates rank collapse. To do so, we compute the effective rank $\mathrm{srank}_\delta(\phi)$ metric of the features learned by Q-functions trained via CQL and CQL with DR3 explicit regularizer. As shown in Figure A.4, for the case of five Atari games, utilizing DR3 alleviates the rank collapse

issue completely (i.e., the ranks do not collapse to very small values when CQL + DR3 is trained for long). We do not claim that the ranks with DR3 are necessarily higher, and infact as we show below, a higher srank of features may not always imply a better solution. The fact that DR3 can prevent rank collapse is potentially surprising, because no term in the practical DR3 regularizer explicitly aims to increase rank: feature dot products can be made smaller while retaining low ranks by simply rescaling the feature vectors. But, as we observe, utilizing DR3 enables learning features that do not exhibit collapsed ranks, thus we hypothesize that correcting for appropriate terms in $\mathrm{R}_{\mathrm{TD}}(\theta)$ can address some of the previously observed pathologies in TD-learning.

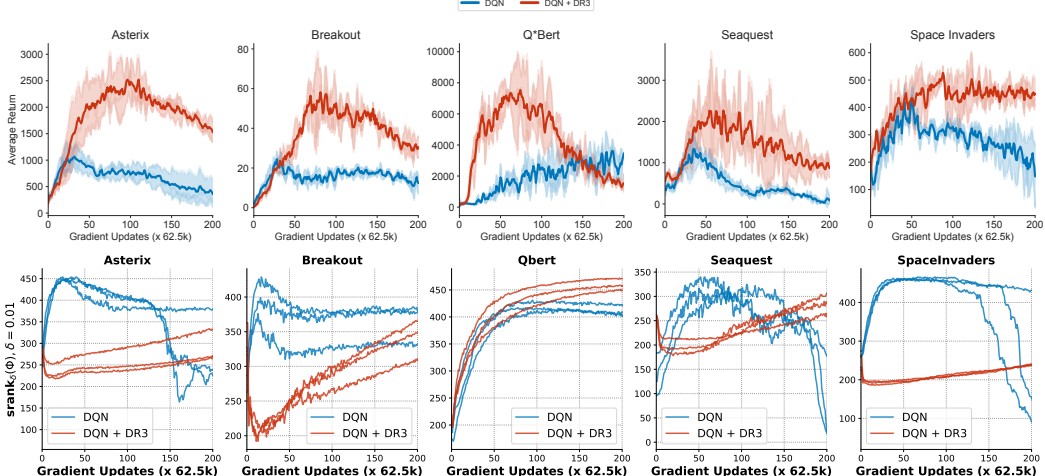

Figure A.5: **Performance and** srank **values for DQN and DQN + DR3.** Observe that the srank values increase for DQN + DR3, while they collapse for DQN on Asterix, Seaquest and SpaceInvaders with more training. Thus, DQN + DR3 does not suffer from a sudden rank collapse. However, a higher srank does not imply a better return, and so while initially DQN does have a high rank, DQN + DR3 performs superiorly.

We now investigate the feature ranks of a Q-network trained when DR3 is applied in conjunction with a standard DQN and REM [2] on the Atari domains. We plot the values of $\mathrm{srank}_\delta(\phi)$, the feature dot products and the performance of the algorithm for DQN in Figure A.5 and for REM in Figure A.6. In the case of DQN, we find that unlike the base DQN algorithm for which feature rank does begin to collapse with more training, the srank for DQN + DR3 is increasing. We also note that DQN + DR3 attains a better performance compared to DQN, throughout training.

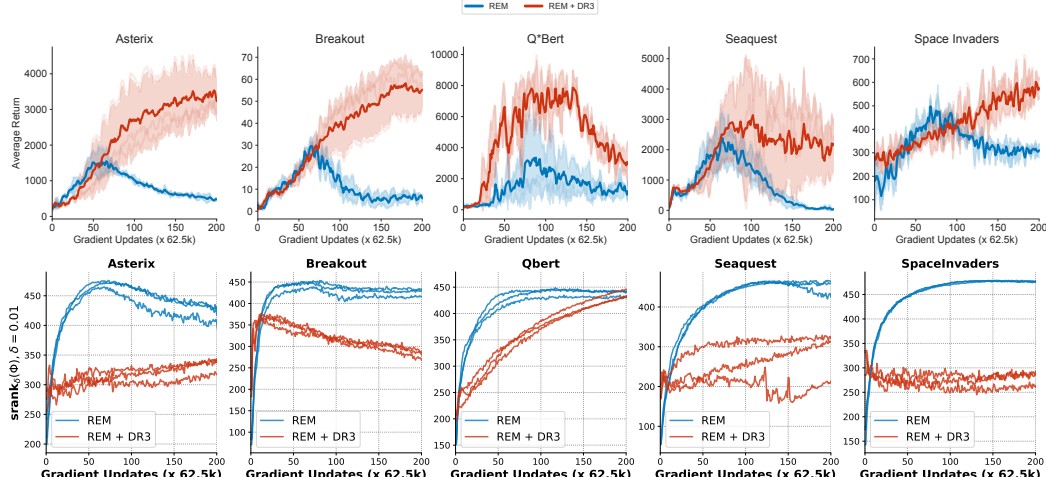

Figure A.6: **Comparing the performance and** srank **values for REM and REM + DR3.** Observe that while REM + DR3 outperforms REM, the srank values attained by REM are much larger than REM + DR3, and none of these ranks have collapsed. Thus, while REM + DR3 maintains non-collapsed features, for the case of REM, it reduces the value of srank and attains better performance. This does not contradict the observations from Kumar et al. [28] as we discuss in the text.

Table A.1: Normalized interquartile mean performance with 95% stratified bootstrap CIs [3] across 17 Atari games of REM, REM + $\Delta'(\Phi)$ (Stop gradient in DR3), REM + DR3 after 6.5M gradient steps for the 1% setting and 12.5M gradient steps for the 5%, 10% settings. Observe that REM + $\Delta'(\phi)$ also improves over the base REM method significantly, by about 130%, even though $\Delta'(\phi)$ is generally comparable and somewhat worse than the DR3 regularizer used in the main paper.

| Data | REM | REM + $\Delta'(\Phi)$ | REM+DR3 |
|------|-----|----------------------|---------|
| 1% | 4.0 (3.3, 4.8) | 15.0 (13.4, 16.6) | 16.5 (14.5, 18.6) |
| 5% | 25.9 (23.4, 28.8) | 55.5 (50.8, 59.8) | 60.2 (55.8, 65.1) |
| 10% | 53.3 (51.4, 55.3) | 67.7 (64.7, 71.3) | 73.8 (69.3, 78) |

However, we note that the opposite trend is true for the case of REM: while REM + DR3 attains a better performance than REM, adding DR3 leads to a reduction in the srank value compared to base REM. At a first glance, this might seem contradicting Kumar et al. [28], but this is not the case: to our understanding, Kumar et al. [28] establish a correlation between extremely low rank values (i.e., rank collapse) and poor performance, but this does not mean that all high rank features will lead to good performance. We suspect that since REM trains a multi-headed Q-function with shared features and randomized target values, it is able to preserve high-rank features, but this need not mean that these features are useful. In fact, as shown in Figure A.7, we find that the base REM algorithm does exhibit feature co-adaptation. This case is an example where the srank metric from Kumar et al. [28] may not indicate poor performance.

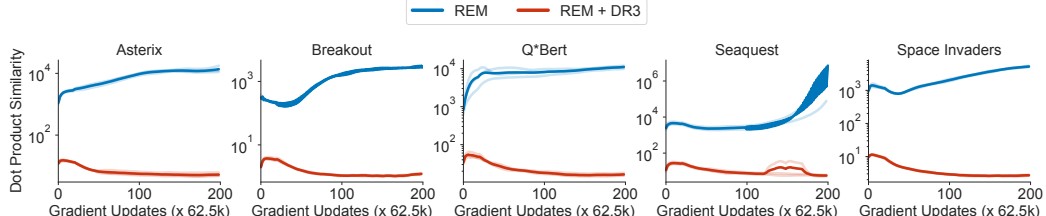

Figure A.7: **Feature dot products for REM and REM + DR3 on log scale.** REM does suffer from feature co-adaptation despite high-rank features.

## A.5 INDUCED IMPLICIT REGULARIZER: THEORY AND PRACTICE

In this section, we compare the performance of our practical DR3 regularizer to the regularizers (Equation 4) obtained for different choices of $M$, such as $M$ induced by noise, studied in previous work, and also evaluate the effect of dropping the stop gradient function from the practical version of our regularizer.

**Empirically comparing the explicit regularizers for different noise covariance matrices, $M$.** The theoretically derived regularizer (Equation 4) suggests that for a given choice of $M$, the following equivalent of feature dot products should increase over the course of training:

$$\Delta_M(\theta) := \sum_{\mathbf{s},\mathbf{a} \in \mathcal{D}} \text{trace} \left[ \Sigma_M^* \nabla Q(\mathbf{s}, \mathbf{a}) \nabla Q(\mathbf{s}', \mathbf{a}')^\top \right]. \quad \text{(Generalized dot products)} \quad \text{(A.1)}$$

We evaluate the efficacy of the explicit regularizer that penalizes the generalized dot products, $\Delta_M(\theta)$, in improving the performance of the policy, with the goal of identifying if our practical method performs similar to this regularizer on generalized dot products.. While $\Sigma_M^*$ must be explicitly computed by running fixed point iteration for every parameter iterate $\theta$ found during TD-learning – which makes this method significantly computationally expensive[2], we evaluated it on five Atari games for $50 \times 62.5k$ gradient steps as a proof of concept. As shown in Figure A.8, the DR3 penalty with the choice of $M$ which corresponds to label noise, and the dot product DR3 penalty, which is our main practical approach in this paper generally perform similarly on these domains, attaining almost identical learning curves on **4/5 games**, and clearly improving over the base algorithm. This hints at the possibility of utilizing other noise covariance matrices to derive an explicit regularizer.

---

[2]In our implementation, we run 20 steps of the fixed-point computation of $\Sigma$ as shown in Theorem 3.1 for each gradient step on the Q-function, and this increases the runtime to about 8 days for 50 iterations on a P100 GPU.

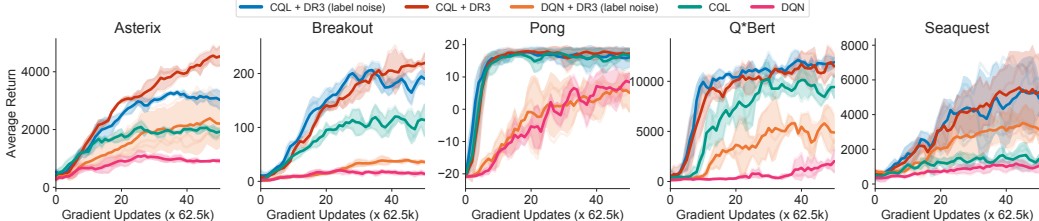

**Figure A.8: Comparing the performance of explicit penalties for two different choices of the covariance matrix** $M$**.** Observe that in all the five games the DR3 regularizer derived for the choice of $M$ from Blanc et al. [6] also leads to a substantial increase in performance over the base algorithm, and in four of five games, DR3 (label-noise) works just as well as DR3.

Deriving more computationally efficient versions of the regularizer for a general $M$ and identifying the best choice of $M$ are subject to future work.

**Effect of stop gradient.** Finally, we investigate the effect of utilizing a stop gradient in the DR3 regularizer. We run a variant of DR3: $\Delta'(\phi) = \sum_{\mathbf{s},\mathbf{a},\mathbf{s}'} \phi(\mathbf{s},\mathbf{a})^\top [[\phi(\mathbf{s}',\mathbf{a}')]]$, with the stop gradient on the second term $(\mathbf{s}',\mathbf{a}')$ and present a comparison to the one without the stop gradient in Table A.1 for REM as the base offline method, averaged over 17 games. Note that this version of DR3, with the stop gradient, also improves upon the baseline offline RL method (i.e., REM) by **130%**. While this performs largely similar, but somewhat worse than the complete version without the stop gradient, these results do indicate that utilizing $\Delta'(\phi)$ can also lead to significant gains in performance.

## A.6 Understanding Feature Co-Adaptation Some More

In this section, we present some more empirical evidence to understand feature co-adaptation. The three factors we wish to study are: **(1)** the effect of target update frequency on feature co-adaptation; **(2)** understand the trend in normalized similarities and compare these to the trend in dot products; and **(3)** understand the effect of out-on-sample actions in TD-learning and compare it to offline SARSA on a simpler gridworld domain. We answer these questions one by one via experiments aiming to verify each hypothesis.

### A.6.1 Effect of Target Update Frequency on Feature Co-Adaptation

We studied the effect of target update frequency on feature co-adaptation, on some gridworld domains from Fu et al. [17]. We utilized the `grid16smoothobs` environment, where the goal of the agent is to navigate from the center of a $16 \times 16$ gridworld maze to one of its corners while avoiding obstacles and "lava" cells. The observations provided to the RL algorithm are given by a high-dimensional random transformation of the $(x, y)$ coordinates, smoothed over neighboring cells in the gridworld. We sampled an offline dataset of 256 transitions and trained a Q-network with two hidden layers of size $(1024, 1024)$ via fitted Q-iteration (FQI) [37].

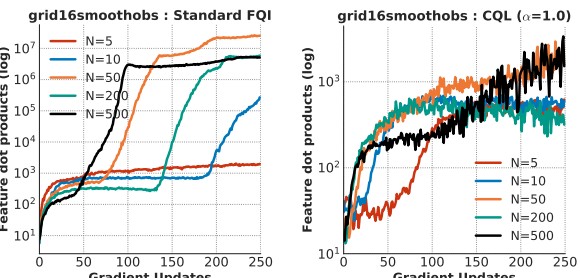

**Figure A.9: Comparing the feature dot products for various target update delays**, where a smaller $N$ implies a faster update and a larger $N$ corresponds to a slower target update. Observe that while slower updates to the target network may reduce co-adaptation, very slow target updates may still lead to excesssive co-adaptation.

We evaluated the feature dot products for Q-functions trained with a varying target update frequencies, given generically as: updating the target network using a hard target update once per $N$ gradient steps, where $N$ takes on values $N = 5, 10, 50, 100, 200, 500$, and present the results in Figure A.9 (left), averaged over 3 random seeds. Thee feature dot products initially decrease from $N = 5$ to $N = 10$, because the target network is updated slower, but then starts to rapidly increase when when the target network is slowed down further to $N = 50$ and $N = 200$ in one case and $N = 500$ in the other case.

We also evaluated the feature dot products when using CQL as the base offline RL algorithm. As shown in Figure A.9 (right), while CQL does reduce the absolute range of the feature dot products, slow target updates with $N = 500$ still lead to the highest feature dot products as training progresses. **Takeaway:** While it is intuitive to think that a slower target network might alleviate co-adaptation, we see that this is not the case empirically with both FQI and CQL, suggesting a deeper question that is an interesting avenue for future study.

### A.6.2 GRIDWORLD EXPERIMENTS COMPARING TD-LEARNING VS OFFLINE SARSA

To supplement the analysis in Section 3.1, we ran some experiments in the gridworld domains from Fu et al. [17]. In this case, we used the `grid16smoothsparse` and `grid16randomsparse` domains, which present challenging navigation tasks in a maze under a 0-1 sparse reward signal, provided at the end of the trajectory. Additionally, the observations available to the offline RL agent do not consist of the raw $(x, y)$ locations of the agent in the maze, but rather high-dimensional randomly chosen transformations of $(x, y)$ in the case of `grid16randomsparse`, which are additionally smoothed locally around a particular state to obtain `grid16smoothsparse`.

Since our goal is to compare feature co-adaptation in TD-learning and offline SARSA, we consider a case where we evaluate a "mixed" behavior policy that chooses the optimal action with a probability of 0.7 at a given state, and chooses a random, suboptimal action with 0.3. We then generate a dataset of size 256 transitions and train offline SARSA and TD-learning on this data. While SARSA backups the next action observed in the offline dataset, TD-learning computes a full expectation of the Q-function $\mathbb{E}_{\mathbf{a}' \sim \pi_\beta(\cdot|\mathbf{s}')}[Q(\mathbf{s}', \mathbf{a}')]$ under the behavior policy for computing Bellman backup targets. The behavior policy is fully known to the TD-learning agent. Our Q-network consists of two hidden layers of size $(1024, 1024)$ as before.

We present the trends in the feature dot products for TD-learning and offline SARSA in Figure A.10, averaged over three seeds. Observe that the trends in the dot product values for TD-learning and offline SARSA closely follow each other for the initial few gradient steps, soon, the dot products in TD-learning start growing faster. In contrast, the dot products for SARSA either saturate or start decreasing. The only difference between TD-learning and SARSA is the set of actions used to compute Bellman targets – while the actions used for computing Bellman backup

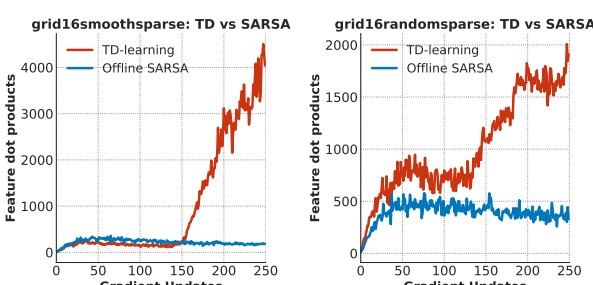

Figure A.10: **Comparing the feature dot products for TD-learning and offline SARSA**, used to compute the value of the behavior policy using a dataset of size 256 on two gridworld domains. Observe that the feature dot products are higher in the case of TD-learning compared to offline SARSA.

targets in SARSA are in-sample actions and are observed in the dataset, the actions used by TD-learning may be out-of-sample, but are still within the distribution of the data-generating behavior policy. This supports our empirical evidence in the main paper showing that out-of-sample actions can lead to feature co-adaptation.

### A.6.3 FEATURE CO-ADAPTATION AND NORMALIZED FEATURE SIMILARITIES

Note that we characterized feature co-adaptation via the dot products of features. In this section, we explore the trends in other notions of similarity, such as cosine similarity between $\phi(\mathbf{s}, \mathbf{a})$ and $\phi(\mathbf{s}', \mathbf{a}')$ which measures the dot product of feature vectors at consecutive state-action tuples after normalization. Formally,

$$\cos(\phi(\mathbf{s}, \mathbf{a}), \phi(\mathbf{s}', \mathbf{a}')) := \frac{\phi(\mathbf{s}, \mathbf{a})^\top \phi(\mathbf{s}', \mathbf{a}')}{||\phi(\mathbf{s}, \mathbf{a})||_2 \cdot ||\phi(\mathbf{s}', \mathbf{a}')||_2}.$$

We plot the trend in the cosine similarity with and without DR3 for five Atari games in Figure A.11 with CQL, DQN and REM, and for the three MuJoCo tasks studied in Appendix A.3 in Figure A.12. We find that the cosine similarity is generally very high on the Atari domains, close to 1, and not indicative of performance degradation. On the Ant and Walker2d MuJoCo domains, we find that

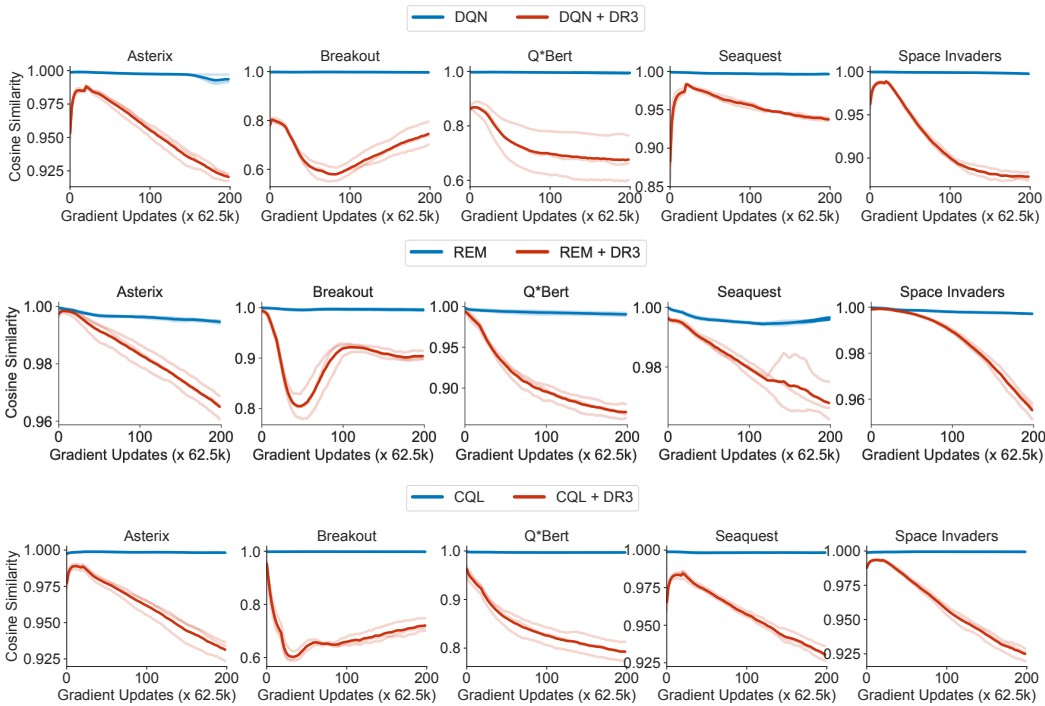

Figure A.11: **Cosine similarities of DQN, DQN + DR3, REM, REM + DR3 and CQL, CQL + DR3.** Note that DQN, REM and CQL attain close to 1 cosine similarities, and addition of DR3 does reduce the cosine similarities of consecutive state-action features.

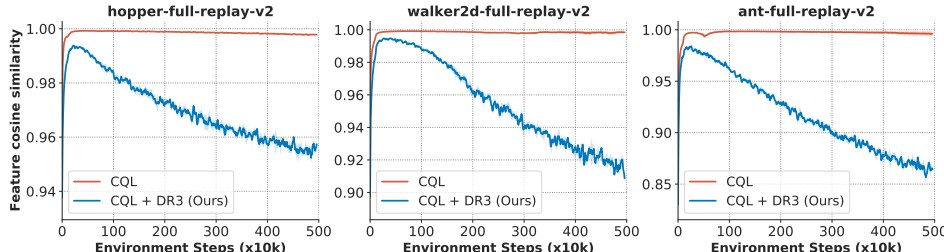

Figure A.12: **Cosine similarities of CQL and CQL + DR3 on MuJoCo domains.** Note that the cosine similarities of CQL grow to 1 and roughly stabilize for Ant and Walker2d, but start decreasing for Hopper. This happens despite the oscillatory trends in performance of CQL on Hopper A.3. This means that a low cosine similarity need not imply poor performance, and DR3 can improve performance even when cosine similarity of base CQL is decreasing. We also notice that DR3 does actually reduce cosine similarity.

the cosine similarity first rises up close to 1 and roughly saturates there. On the Hopper domain, the cosine similarity even decreases over training. However we observe that the feature dot products are increasing for all the domains. Applying DR3 in both cases improves performance (as shown in earlier Appendix), and generally gives rise to reduced cosine similarity values, though it can also increase the cosine similarity values occasionally. Furthermore, even when the cosine similarities were decreasing for the base algorithm (e.g., in the case of Hopper), addition of DR3 reduced the feature dot products and helped improve performance. This indicates that both the norm and directional alignment are contributors to the co-adaptation issue, which is what DR3 aims to fix and independently directional alignment does not indicate poor performance.

## A.7   STABILITY OF DR3 FROM A GOOD SOLUTION

In this appendix, we study the trend of CQL + DR3 when starting learning from a good initialization, which was studied in Figure 2. As shown in Figure A.13, while the performance for baseline CQL degrades significantly (from 5000 at initialization on Asterix, performance degrades to ∼2000 by

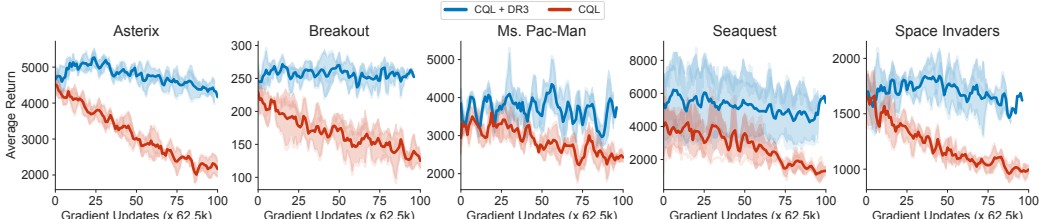

Figure A.13: **Running CQL + DR3 and CQL in the setup of Figure 2 to evaluate the stability of CQL + DR3 when starting training from a good solution.** Observe that the performance of base CQL decays quickly from the good solution, but CQL + DR3 is *relatively* more stable.

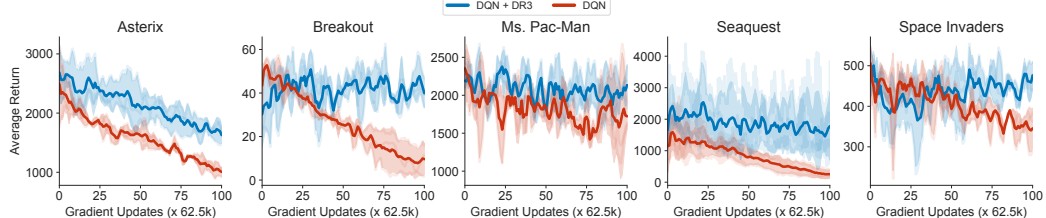

Figure A.14: **Running DQN + DR3 and DQN in the setup of Figure 2 to evaluate the stability of DQN + DR3 when starting training from a good solution.** Observe that the performance of base DQN decays quickly from the good solution, but DQN + DR3 is *relatively* more stable.

100 iterations for base CQL), whereas the performance of DR3 only moves from 5000 to ~4300. A similar trend holds for Breakout. This means that the addition of DR3 does stabilize the learning relative to the baseline algorithm. Please note that we are not claiming that DR3 is unequivocally stable, but that improves stability relative to the base method.

## A.8   STATISTICAL SIGNIFICANCE OF DR3 AND FRANKA KITCHEN RESULTS

Table A.2: **Performance of CQL, CQL + DR3 after 2M gradient steps on the Franka Kitchen domains** averaged over 3 seeds. This is training for **6x** longer compared to CQL defaults. Observe that CQL + DR3 outperforms CQL at 2M steps, indicating is efficacy in preventing long term performance degradation.

| D4RL Task | CQL | CQL + DR3 |
|---|---|---|
| kitchen-mixed | $27.67 \pm 12.66$ | **$37.00 \pm 11.53$** |
| kitchen-partial | $20.67 \pm 15.57$ | **$40.67 \pm 4.04$** |
| kitchen-complete | $28.00 \pm 14.73$ | **$38.67 \pm 6.66$** |

We present the results comparing CQL and CQL+DR3 on the Franka Kitchen tasks from D4RL in Table A.2. Observe that CQL+DR3 outperforms CQL, and to test the statistical significance of these results, we analyze the probability of improvement of CQL+DR3 over CQL next.

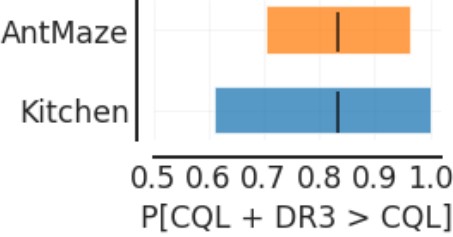

Figure A.15: **Statistical significance of the results of CQL + DR3 vs CQL (Table 2) as measured by average probability of improvement [3],** with stratified bootstrap confidence intervals for this statistic. Since the lower CI for this statistic is $> 0.5$, CQL + DR3 **significantly** improves over base CQL, and since the mean and upper CI are $\geq 0.75$, this improvement is also **meaningful**.

In order to assess the statistical significance of our D4RL Antmaze and Kitchen results, we follow the recommendedations by Agarwal et al. [3] for comparing deep RL algorithms considering their

statistical uncertainties. Specifically, we computed the average probability of improvement [3] of CQL + DR3 over CQL on the antmaze and kitchen domains, and we find that DR3 **does** significantly and meaningfully improve over CQL on both the Kitchen and AntMaze domains. Before presenting the results, let us first describe the metric we compute.

**Probability of improvement and statistical significance**. For two given algorithms $\mathsf{Alg}_1$ and $\mathsf{Alg}_2$, and runs $X_{k,1}, X_{k,2}, \cdots, X_{k,m}$ from $\mathsf{Alg}_1$ and runs $Y_{k,1}, Y_{k,2}, \cdots, Y_{k,n}$ from $\mathsf{Alg}_2$ on task $k$, the probability of improvement of $\mathsf{Alg}_1$ over $\mathsf{Alg}_2$ is given by $P(\mathsf{Alg}_1 > \mathsf{Alg}_2) = \frac{1}{K} \sum_{k=1}^{K} P(\mathsf{Alg}_1^k > \mathsf{Alg}_2^k)$. The probability of improvement on a given task $k$, $P(\mathsf{Alg}_1^k > \mathsf{Alg}_2^k)$ is computed using the Mann-Whitney U-statistic and is given by:

$$P(\mathsf{Alg}_1^k > \mathsf{Alg}_2^k) = \frac{1}{MN} \sum_{i=1}^{M} \sum_{j=1}^{N} S(X_{k,i}, Y_{k,j}) \quad \text{where} \quad S(x,y) = \begin{cases} 1, & \text{if } y < x, \\ \frac{1}{2}, & \text{if } y = x, \\ 0, & \text{if } y > x. \end{cases}$$

$\mathsf{Alg}_1$ leads to statistically *significant* improve over $\mathsf{Alg}_2$ if the lower CI for $P(\mathsf{Alg}_1 > \mathsf{Alg}_2)$ is larger than 0.5. Per the Neyman-Pearson statistical testing criterion in Bouthillier et al. [7], $\mathsf{Alg}_1$ leads to statistically *meaningful* improvement over $\mathsf{Alg}_2$ if the upper confidence interval (CI) of $P(\mathsf{Alg}_1 > \mathsf{Alg}_2)$ is larger than 0.75.

Figure A.15 presents the value of $P(\text{CQL} + \text{DR3} > \text{CQL})$ on the AntMaze and Kitchen domains at 2M gradient steps along with the 95% CI for this statistic. DR3 improves over CQL on the AntMaze domains with probability **0.83** with 95% CI (0.7, 0.96) and on the Kitchen domains with probability **0.8** with 95% CI (0.6, 1.0). These values pass the criterion of being both statistically significant and meaningful per the above definitions, implying that DR3 does significantly and meaningfully improve upon CQL on these domains.

## B  RELATED WORKS

In this section, we briefly review some extended related works, and in particular, try to connect feature co-adaptation and implicit regularization to various interesting results pertaining to RL lower-bounds with function approximation and self-supervised learning.

### B.1  BRIEF SUMMARY OF RELATED WORK

Prior analyses of the learning dynamics in RL has focused primarily on analyzing error propagation in tabular or linear settings [*e.g.*, 9, 13, 50, 45, 46, 15, 12], understanding instabilities in deep RL [1, 5, 26, 44] and deriving weighted TD updates that enjoy convergence guarantees [31, 32, 40], but these methods do not reason about implicit regularization or any form of representation learning. Ghosh & Bellemare [20] focuses on understanding the stability of TD-learning in underparameterized linear settings, whereas our focus is on the overparameterized setting, when optimizing TD error and learning representations via SGD. Kumar et al. [28] studies the learning dynamics of Q-learning and observes that the rank of the feature matrix, $\Phi$, drops during training. While this observation is related, our analysis characterizes the implicit preference of learning towards feature co-adaptation (Theorem 3.1) on out-of-sample actions as the primary culprit for aliasing. Additionally, while the goal of our work is not to increase srank($\Phi$), utilizing DR3 not only outperforms the srank($\Phi$) penalty in Kumar et al. [28] by more than **100%**, but it also alleviates rank collapse, with no apparent term that explicitly enforces high rank values. Somewhat related to DR3, Durugkar & Stone [14], Pohlen et al. [35] heuristically constrain gradients of TD to prevent changes in target Q-values to prevent divergence. Contrary to such heuristic approaches, DR3 is inspired from a theoretical model of implicit regularization, and does not prevent changes in target values, but rather reduces feature dot products.

### B.2  EXTENDED RELATED WORK

**Lower-bounds for offline RL.** Zanette [51] identifies hard instances for offline TD learning of linear value functions when the provided features are "aliased". Note that this work does not consider feature learning or implicit regularization, but their hardness result relies heavily on the fact the given linear features are aliased in a special sense. Aliased features utilized in the hard instance inhibit learning along certain dimensions of the feature space with TD-style updates, necessitating an

exponential sample size for near-accurate value estimation, even under strong coverage assumptions. A combination of Zanette [51]'s argument, which provides a hard instance given aliased features, and our analysis, which studies the emergence of co-adapted/similar features in the offline deep RL setting, could imply that the co-adaptation can lead to failure modes from the hard instance, even on standard Offline RL problems, when provided with limited data.

**Connections to self-supervised learning (SSL).** Several modern self-supervised learning methods [21, 10] can be viewwed as utilizing some form of bootstrapping where different augmentations of the same input $(\mathbf{x} + \mathrm{Aug}_1, \mathbf{x} + \mathrm{Aug}_2)$ serve as consecutive state-action tuples that appear on two sides of the backup. If we may extrapolate our reasoning of feature co-adaptation to this setting, it would suggest that performing noisy updates on a self-supervised bootstrapping loss will give us feature representations that are highly similar for consecutive state-action tuples, i.e., the representations for $\phi(\mathbf{x} + \mathrm{Aug}_1)^\top \phi(\mathbf{x} + \mathrm{Aug}_2)$ will be high. Intuitively, an easy way for obtaining high feature dot products is for $\phi(\cdot)$ to capture only that information in $\cdot$, which is agnostic to data augmentation, thus giving rise to features that are invariant to transformations. This aligns with what has been shown in self-supervised learning [41, 42]. Another interesting point to note is that while such an explanation would indicate that highly co-adapted features are beneficial in SSL, such features can be adverse in value-based RL as discussed in Section 3.

**Preventing divergence in deep TD-learning.** Finally, we discuss Achiam et al. [1] which proposes to pre-condition the TD-update using the inverse the neural tangent kernel [25] matrix so that the TD-update is always a contraction, for every $\theta_k$ found during TD-learning. Intuitively, this can be overly restrictive in several cases: we do not need to ensure that TD always contracts, but that is eventually stabilizes at good solution over long periods of running noisy TD updates, Our implicit regularizer (Equation '4) derives this condition, and our theoretically-inspired DR3 regularizer shows that empirically, it suffices to penalize the dot product similarity in practice.

## C  PROOF OF THEOREM 3.1

In this section, we will derive our implicit regularizer $R_{\mathrm{TD}}(\theta)$ that emerges when performing TD updates with a stochastic noise model with covariance matrix $M$. We first introduce our notation that we will use throughout the proof, then present our assumptions and finally derive the regularizer. Our proof utilizes the analysis techniques from Blanc et al. [6] and Damian et al. [11], which analyze label-noise SGD for supervised learning, however key modifications need to be made to their arguments to account for non-symmetric matrices that emerge in TD learning. As a result, the form of the resulting regularizer is very different. To keep the proof concise, we will appeal to lemmas from these prior works which will allow us to bound certain concentration terms.

### C.1  NOTATION

The noisy TD-learning update for training the Q-function is given by:

$$\theta_{k+1} = \theta_k - \eta \underbrace{\left( \sum_i \nabla_\theta Q(\mathbf{s}_i, \mathbf{a}_i) \left( Q_\theta(\mathbf{s}_i, \mathbf{a}_i) - (r_i + \gamma Q_\theta(\mathbf{s}'_i, \mathbf{a}'_i)) \right) \right)}_{:= g(\theta)} + \eta \varepsilon_k, \quad \varepsilon_k \sim \mathcal{N}(0, M) \quad \text{(C.1)}$$

where $g(\theta)$ denotes the parameter update. Note that $g(\theta)$ is not a full gradient of a scalar objective, but it is a form of a "pseudo"-gradient or "semi"-gradient. Let $\varepsilon_k$ denote an i.i.d.random noise that is added to each update. This noise is sampled from a zero-mean Gaussian random variable with covariance matrix $M$, i.e., $\mathcal{N}(0, M)$.

Let $\theta^*$ denote a point in the parameter space such that in the vicinity of $\theta^*$, $g(\theta) \leq \mathscr{C}$, for a small enough $\mathscr{C}$. Let $G(\theta)$ denote the derivative of $g(\theta)$ w.r.t. $\theta$: $G(\theta) = \nabla_\theta g(\theta)$ and let $\nabla G(\theta)$ denote the third-order tensor $\nabla_\theta^2 g(\theta)$. For notation clarity, let $G = G(\theta^*), \nabla G = \nabla G(\theta^*)$. Let $e_i$ denote the signed TD error for a given transition $(\mathbf{s}_i, \mathbf{a}_i, \mathbf{s}'_i) \in \mathcal{D}$ at $\theta^*$:

$$e_i = Q_{\theta^*}(\mathbf{s}_i, \mathbf{a}_i) - (r_i + \gamma Q_{\theta^*}(\mathbf{s}'_i, \mathbf{a}'_i)). \quad \text{(C.2)}$$

Since $\theta^*$ is a fixed point of the training TD error, $e_i = 0$. Following Blanc et al. [6], we will assume that the learning rate in gradient descent, $\eta$, is small and we will ignore terms that scale as $\mathcal{O}(\eta^{1+\delta})$, for $\delta > 0$. Our proof will rely on using a reference Ornstein-Uhlenbeck (OU) process which the TD

parameter iterates will be compared to. Let $\zeta_k$ denote the $k$-th iterate of an OU process, which is defined as:

$$\zeta_{k+1} = (I - \eta G)\zeta_k + \eta\varepsilon_k, \quad \varepsilon_k \sim \mathcal{N}(0, M) \tag{C.3}$$

We will drop $\theta$ from $\nabla_\theta$ to indicate that the gradient is being computed at $\theta^*$, and drop $(\mathbf{s}_i, \mathbf{a}_i)$ from $Q(\mathbf{s}_i, \mathbf{a}_i)$ and instead represent it as $Q_i$ for brevity; we will represent $Q(\mathbf{s}'_i, \mathbf{a}'_i)$ as $Q'_i$. We assume that $\nabla^2 Q_i$ is $\mathscr{L}_2$-Lipschitz and $\nabla^3 Q_i$ is $\mathscr{L}_3$-Lipschitz throughout the parameter space $\Theta$.

## C.2  PROOF STRATEGY

For a given point $\theta^*$ to be an attractive fixed point of TD-learning, our proof strategy would be to derive the condition under which it mimics a given OU noise process, which as we will show stays close to the parameter $\theta^*$. This condition would then be interpreted as the gradient of a "induced" implicit regularizer. If the point $\theta^*$ is not a stationary point of this regularizer, we will show that the movement $\theta$ is large when running the noisy TD updates, indicating that the regularizer, atleast in part guides the dynamics of TD-learning. To show this, we would write out the gradient update, isolate some terms that will give rise to the implicit regularizer, and bound the remaining terms using contraction and concentration arguments. The contraction arguments largely follow prior work (though with key exceptions in handling contraction with asymmetric and complex eigenvalue matrices), while the form of the implicit regularizer is different. Finally, we will interpret the resulting update over large timescales to show that learning is indeed guided by the implicit regularizer.

## C.3  ASSUMPTIONS AND CONDITIONS

Next, we present some key assumptions we will need for the proof. Our first assumption is that the matrix $G \in \mathbb{R}^{d \times d}$ is of maximal rank possible, which is equal to the number of datapoints $n$ and $n \ll d$, the dimensionality of the parameter space. Crucially, this assumption do not imply that $G$ is of full rank – it cannot be, because we are in the overparameterized regime.

**Assumption A1** ($G$ spans an $n$-dimensional basis.). *Assume that the matrix $G$ spans $n$-possible directions in the parameter space and hence, attains the maximal possible rank it can.*

The second condition we require is that the matrices $\sum_i \nabla Q_i \nabla Q_i^\top$ and $M$ share the same $n$-dimensional basis as matrix $G$:

**Assumption A2.** $\sum_i \nabla Q_i \nabla Q_i^\top$, $M$, and $G$ span identical $n$-dimensional subspaces.

This is a technical condition that is required. If this condition is not met, as we will show the learning dynamics of noisy TD will not be a contraction in certain direction in the parameter space and TD-learning will not stabilize at such a solution $\theta^*$. In fact, we will utilize a stronger version of this statement for TD-learning to converge, and we will discuss this shortly.

## C.4  LEMMAS USED IN THE PROOF

Next, we present some lemmas that would be useful for proving the theoretical result.

**Lemma C.1** (Expressions for the first and and second-order derivatives of $g(\theta)$.). *The following definitions and expansions apply to our proof:*

$$G(\theta^*) = \sum_i \nabla^2 Q_i e_i + \sum_i \nabla Q_i (\nabla Q_i - \gamma \nabla Q'_i)^\top$$

$$\nabla G(\theta^*)[\mathbf{v}, \mathbf{v}] = 2\sum_i \nabla^2 Q_i \mathbf{v}\mathbf{v}^\top (\nabla Q_i - \gamma \nabla Q'_i) + \sum_i \operatorname{tr}\left((\nabla^2 Q_i - \gamma \nabla^2 Q'_i)\mathbf{v}\mathbf{v}^\top\right) \nabla Q_i + \nabla^3 Q_i e_i$$

Lemma C.1 presents a decomposition of the matrix $G$ and the directional derivative of the third order tensor $\nabla G[\mathbf{v}, \mathbf{v}]$ in directions $\mathbf{v}$ and $\mathbf{v}$, which will appear in the Taylor expansion layer. Note that at $\theta^*$ since $e_i = 0$, the first term in $G(\theta^*)$ and the third term in $\nabla G(\theta^*)[\mathbf{v}, \mathbf{v}]$ vanish. Lemma C.2 derives a fixed-point recursion for the covariance matrix of the total noise accumulated in the OU-process with covariance matrix $M$ and this will appear in our proof.

**Lemma C.2** (Covariance of the random noise process $\zeta_k$). *Let $\zeta_k$ denote the OU process satisfying: $\zeta_{k+1} = (I - \eta G)\zeta_k + \eta\varepsilon_k$, where $\varepsilon_k \sim \mathcal{N}(0, M)$, where $M \succeq 0$. Then, $\zeta_{k+1} \sim \mathcal{N}(0, \Sigma)$, where $\Sigma$ satisfies the discrete Lyapunov equation:*

$$\Sigma_M^* = (I - \eta G)\Sigma_M^*(I - \eta G)^\top + \eta^2 M.$$

*Proof.* For the OU process, $\zeta_{k+1} = (I - \eta G)\zeta_k + \eta\varepsilon_k$, since $\varepsilon_k$ is a Gaussian random variable, by induction so is $\zeta_{k+1}$, and therefore the covariance matrix of $\zeta_{k+1}$ is given by:

$$\Sigma_{k+1} := (I - \eta G)\Sigma_k(I - \eta G^\top) + \eta^2 M. \tag{C.4}$$

Solving for the fixed point for $\Sigma_k$ gives the desired expression. $\qquad\square$

In our proofs, we will require the following contraction lemmas to tightly bound the magnitude of some zero-mean terms that will appear in the noisy TD update under certain scenarios. Unlike the analysis in Damian et al. [11] and Blanc et al. [6] for supervised learning with label noise, where the contraction terms like $(I - \eta G)^k G$ are bounded by $\approx \frac{1}{k\eta}$ intuitively because $I - \eta G$ is a contraction in the subspace spanned by matrix $G$. However, this is not true for TD-learning directly since terms like $(I - \eta G)^k S$ appear for a different matrix $S$. Therefore, TD-learning will diverge from $\theta^*$ unless matrices $G$ and $M$ have their corresponding eigenvectors assigned to the top eigenvalues be approximately "aligned". We formalize this definition next, and then provide a proof of the concentration guarantee.

**Definition 1** (($\omega, C_0$)-alignment). *Given a positive semidefinite matrix $A$, let $A = U_A\Lambda_A U_A^\top$ denote its eigendecomposition. Without loss of generality assume that the eiegenvalues are arranged in decreasing order, i.e., $\forall i > j, \Lambda_A(i) \leq \Lambda_A(j)$. Given another matrix $B$, let $B = U_B\Lambda_B U_B^H$ denote its complex eigendecomposition, where eigenvalues in $\Lambda_B$ are arranged in decreasing order of their complex magnitudes, i.e., $\forall i > j, |\Lambda_B(i)| \leq |\Lambda_B(j)|$. Then the matrix pair $(A, B)$ is said to be $(\omega, C_0)$-aligned if $|U_B^H(i)U_A(i)| \leq \omega$ and if $\forall\, i, \Lambda_A(i) \leq C_0|\Lambda_B(i)|$ for a constant $C_0$.*

If two matrices are $(\omega, C_0)$-aligned, this means that the corresponding eigenvectors when arranged in decreasing order of eigenvalue magnitude roughly align with each other. This condition would be crucial while deriving the implicit regularizer as it will quantify the rate of contraction of certain terms that define the neighborhood that the iterates of noisy TD-learning will lie in with high probability. We will operate in the setting when the matrix $G$ and $\sum_i \nabla Q_i \nabla Q_i^\top$ are $(\omega, C_0)$-aligned with each other, and matrix $M$ and $G$ are also $(\omega, C_0)$-aligned (note that we can consider $\omega', C_0'$), which will not change our bounds and therefore we go for less notational clutter). Next we utilize this notion of alignment to show a particular contraction bound that extends the weak contraction bound in Damian et al. [11].

**Lemma C.3.** *Assume we are given a matrix $G$ such that $|\lambda_i(I - \eta G)| \leq \rho_0 < 1$ for all $\lambda_i$ such that $\lambda_i \neq 0$. Let $G = U\Lambda U^H$ be the complex eigenvalue decomposition of $G$ (since almost every matrix is complex-diagonalizable). For a positive semi-definite matrix $S$ that is $(\omega, C_0)$-aligned with $G$, if $S = U_S\Lambda_S U_S^\top$ is its eigenvalue decomposition, the following contraction bound holds:*

$$||(I - \eta G)^k S|| = \mathcal{O}\left(\frac{\omega C_0}{\eta k}\right)$$

*Proof.* To prove this statement, we can expand $(I - \eta G)$ using its eigenvalue decomposition only in the subspace that is jointly shared by $G$ and $M$, and then utilize the definition of $\omega$-alignment to bound the terms.

$$||(I - \eta G)^k S|| = ||(I - \eta U\Lambda U^H)^k U_S\Lambda_S U_S^\top|| \tag{C.5}$$

$$= ||(UU^H - \eta U\Lambda_U U^H)^k U_S\Lambda_S U_S^\top|| \tag{C.6}$$

$$= \left|\left| U(I - \eta\Lambda)^k U^H U_S\Lambda_S U_S^\top \right|\right| \tag{C.7}$$

$$\leq \omega \cdot ||(I - \eta\Lambda)^k|| \cdot \Lambda_S \tag{C.8}$$

$$\leq \omega \cdot C_0 \cdot \left(\max_i\ |1 - \eta\Lambda(i)|^k|\Lambda(i)|\right) \tag{C.9}$$

Now we need to solve for the inner maximization term. When $\Lambda(i)$ is not complex for any $i$, the term above is $\lesssim 1/\eta k$ using the result from Damian et al. [11], but when $\Lambda(i)$ is complex, this bound can only hold under certain conditions. To note when this quantity is bounded, we expand $|1 - \eta x|^k$ for some complex number $x = r(\cos\theta + \iota\sin\theta)$:

$$|1 - \eta x|^k = |(1 - \eta r\cos\theta) + \iota\eta r\sin\theta| \tag{C.10}$$

$$= \left[\sqrt{(1 - \eta r\cos\theta)^2 + \eta^2 r^2\sin^2\theta}\right]^k = \left(1 + \eta^2 r^2 - 2\eta r\cos\theta\right)^{k/2} \tag{C.11}$$

$$\implies |1 - \eta x|^k|x| = \left(1 + \eta^2 r^2 - 2\eta r\cos\theta\right)^{k/2} r \tag{C.12}$$

$$\lesssim \frac{1}{\eta k} \quad \text{if } \eta \leq \min_i \frac{\operatorname{Re}(\Lambda(i))}{|\Lambda(i)|} \quad \text{and} \quad \infty \text{ otherwise.} \tag{C.13}$$

Plugging back the above expression in the bound above completes the proof. $\qquad\square$

The proof of Lemma C.3 indicates that unless the learning rate $\eta$ and the matrix $G$ are such that the $|\lambda_i(I - \eta G)| \leq \rho < 1$ in directions spanned by matrix $S$, such an expression may not converge. This is expected since the matrix $I - \eta G$ will not contract in directions of non-zero eigenvalues if the real part $r\cos\theta$ is negative or zero. Additionally, we note that under Definition 1, we can extend several weak-contraction bounds from Damian et al. [11] (Lemmas 9-14 in Damian et al. [11]) to our setting.

Next, Lemma C.4 shows that the OU noise iterates are bounded with high probability when Definition 1 holds:

**Lemma C.4** ($\zeta_k$ is bounded with high probability). *With probability atleast $1 - \delta$ and under Definition 1, $\|\zeta_k\| \leq n\omega\sqrt{\eta C_0}\log\frac{1}{\delta} = \mathcal{O}(\sqrt{\eta})$.*

*Proof.* To prove this lemma, we first bound the trace of the covariance matrix $\Sigma_{k+1}$ and then apply high probability bounds on the Martingale norm concentration. The trace of the covariance matrix $\Sigma_{k+1}$ can be bounded as follows (all the equations below are restricted to the dimensions of non-zero eigenvalues of $G$):

$$\operatorname{tr}[\Sigma_{k+1}] = \sum_{j\leq k} \operatorname{tr}\left[(I - \eta G)^j M(I - \eta G^\top)^j\right] \tag{C.14}$$

$$= \sum_{j\leq k} \operatorname{tr}\left[(UU^H - \eta U\Lambda U^H)^j M(UU^H - \eta U\Lambda U^H)^j\right] \tag{C.15}$$

$$= \sum_{j\leq k} \operatorname{tr}\left[U(I - \eta\Lambda)^j U^H U_M \Lambda_M U_M^\top U(I - \eta\Lambda)^j U^H\right] \tag{C.16}$$

$$= \sum_{j\leq k} n\omega^2 C_0 \operatorname{tr}\left[|I - \eta\Lambda|^j \cdot |\Lambda| \cdot |I - \eta\Lambda|^j\right] \tag{C.17}$$

$$\leq n\omega^2 C_0 \sum_{j\leq k} n \cdot \max_\lambda(|1 - \eta\lambda|^{2j} \cdot |\lambda|) \leq \eta n^2 C_0 \omega^2 \tag{C.18}$$

Now, we can apply Corollary 1 from Damian et al. [11] to obtain a bound on $\|\zeta_k\|$ as with high probability, atleast $1 - \delta$, $\|\zeta_k\| \leq \sqrt{2\operatorname{tr}(\Sigma)\log\frac{1}{\delta}} = n\omega\sqrt{\eta C_0}\log\frac{1}{\delta}$. $\qquad\square$

## C.5  MAIN PROOF OF THEOREM 3.1

In this section, we present the main proof of Theorem 3.1. The proof involves two components: **(1)** the part where we derive the regularizer, and **(2)** bounding additional terms via concentration inequalities. Part **(1)** is specific to TD-learning, while a lot of the machinery for part **(2)** is directly taken from prior work [11] and Blanc et al. [6]. We focus on part **(1)** here.

Our strategy is to analyze the learning dynamics of noisy TD updates that originate at $\theta^*$. In a small neighborhood around $\theta^*$, we can expand the noisy TD update (Equation 3) using Taylor's expansion

around $\theta^*$ which gives:

$$\theta_{k+1} = \theta_k - \eta g(\theta_k) + \eta \varepsilon_k, \ \ \varepsilon_k \sim \mathcal{N}(0, M) \tag{C.19}$$

$$\implies \theta_{k+1} = \theta_k - \eta \left( g + G(\theta_k - \theta^*) - \frac{\eta}{2} G[\theta_k - \theta^*, \theta_k - \theta^*] \right) + \eta \varepsilon_k + \mathcal{O}(\eta \|\theta_k - \theta^*\|^3). \tag{C.20}$$

Denoting $\nu_k := \theta_k - \theta^*$, using the fact that $\|g(\theta^*)\| \leq \mathscr{C}$, we find that $\nu_k$ can be written as:

$$\nu_{k+1} = (I - \eta G)\nu_k + \varepsilon_k + \frac{\eta}{2} G[\nu_k, \nu_k] + \mathcal{O}(\eta \|\nu_k\|^3 + \eta \mathscr{C}) \tag{C.21}$$

Since the OU process $\zeta_k$ stays in the vicinity of the point $\theta^*$, and follows a similar recursion to the one above, our goal would be to design a regularizer so that Equation C.21 closely follows the OU process. Thus, we would want to bound the difference between the variable $\nu_k$ and the variable $\zeta_k$, denoted as $r_k$ to be within a small neighborhood:

$$r_{k+1} = \nu_{k+1} - \zeta_{k+1} = (I - \eta G) \underbrace{(\nu_k - \zeta_k)}_{r_k} + \frac{1}{2} G[\nu_k, \nu_k] + \mathcal{O}(\eta \|\nu_k\|^3 + \eta \mathscr{C}).$$

We can write down an expression for $r_k$ summing over all the terms:

$$r_{k+1} = \underbrace{-\frac{\eta}{2} \sum_{j \leq k} (I - \eta G)^{k-j} \nabla G[\nu_k, \nu_k]}_{\text{term (a)}} + \underbrace{\sum_{j \leq k} (I - \eta G)^j \left[ \mathcal{O}(\eta \|\nu_k\|^3 + \eta \mathscr{C}) \right]}_{\text{term (b)}}. \tag{C.22}$$

Term (a) in the above equation is the one that can induce a displacement in $r_k$ as $k$ increases and would be used to derive the regularizer, whereas term (b) primarily consists of terms that concentrate to 0. We first analyze term (a) and then we will analyze the concentration terms later.

To analyze term (a), note that the term $\nabla G[\nu_k, \nu_k]$, by Lemma C.1, only depends on $\nu_k$ via the covariance matrix $\nu_k \nu_k^\top$. So we will partition this term into two terms: **(i)** a term that utilizes the asymptotic covariance matrix of the OU process and **(ii)** errors due to a finite $k$ and stochasticity that will concentrate.

$$2 \times \text{(a)} = \eta \sum_{j \leq k} (I - \eta G)^{k-j} \nabla G[\nu_k, \nu_k] \tag{C.23}$$

$$= \sum_{j \leq k} (I - \eta G)^{k-j} \nabla G[\zeta^*, \zeta^*] + \sum_{j \leq k} (I - \eta G)^{k-j} \nabla G([\nu_k, \nu_k] - [\zeta^*, \zeta^*]), \tag{C.24}$$

The first term is a "bias" term and doesn't concentrate to 0, and will give rise to the regularizer. We can break this term using Lemma C.1 as:

$$\nabla G[\zeta^*, \zeta^*] = 2 \sum_i \nabla^2 Q_i \Sigma_M^* (\nabla Q_i - \gamma \nabla Q_i') + \sum_i \text{tr} \left[ (\nabla^2 Q_i - \gamma \nabla^2 Q_i') \Sigma_M^* \right] \nabla Q_i \tag{C.25}$$

The regularizer $R_{\text{TD}}(\theta)$ is the function such that:

$$\nabla_\theta R_{\text{TD}}(\theta) = \sum_i \nabla^2 Q_i \Sigma_M^* (\nabla Q_i - \gamma \nabla Q_i') \tag{C.26}$$

$$\implies R_{\text{TD}}(\theta) = \sum_i \nabla Q_i \Sigma_M^* \nabla Q_i^\top - \gamma \sum_i \text{trace} \left( \Sigma_M^* \nabla Q_i [[\nabla Q_i']]^\top \right), \tag{C.27}$$

where $[[\cdot]]$ denotes the stop gradient operator. If the point $\theta^*$ is a stationary point of the regularizer $R_{\text{TD}}(\theta)$, then Equations C.26 and C.27 imply that the first term of Equation C.25 must be 0. Therefore in this case to show that $\theta^*$ is attractive, we need to show that the other terms in Equations C.25, C.24 and term (b) in Equation C.22 concentrate around 0 and are bounded in magnitude. The remaining part of the proof shown in Appendix C.7 provides these details, but we first summarize the main takeaways in the proof to conclude the argument.

## C.6   SUMMARY OF THE ARGUMENT

We will show how to concentrate terms in Equation C.26 besides the regularizer largely following the techniques from prior work, but we first summarize the entire proof. The overall update to the vector $r_k$ which measures the displacement between the parameter vector $\theta_k - \theta^*$ and the OU-process $\zeta_k$ can be written as follows, and it is governed by the derivative of the implicit regularizer (modulo error terms):

$$r_{k+1} = -\frac{\eta}{2}\sum_{j \le k}(I - \eta G)^{k-j}\nabla_\theta R_{\text{TD}}(\theta^*) + \mathcal{O}\left(\sqrt{\eta t} \cdot \text{poly}(\mathscr{C}, \mathscr{L}_2, \mathscr{L}_3, \omega, C_0)\right). \quad \text{(C.28)}$$

An important detail to note here is that since the regularizer consists of $\Sigma_M^*$ and the size of $\Sigma_M^*$ (i.e, its eigenvalues), as shown in Lemma C.4 depends on one factor of $\eta$. So, effectively the first term in Equation C.28 does depend on two factors of $\eta$. Using Equation C.28, we can write the deviation between $\theta^*$ and $\theta_k$ as:

$$\nu_{k+1} = \zeta_{k+1} - \frac{\eta}{2}\sum_{j \le k}(I - \eta G)^{k-j}\nabla_\theta R_{\text{TD}}(\theta^*) + \mathcal{O}\left(\sqrt{\eta t} \cdot \text{poly}(\mathscr{C}, \mathscr{L}_2, \mathscr{L}_3, \omega, C_0)\right). \quad \text{(C.29)}$$

The OU process $\zeta_k$ converges to $\theta^*$ in the subspace spanned by $G$, since the condition $\rho(I - \eta G) < 1$ is active in this subspace (if the condition that $\rho(I - \eta G) < 1$ in the subspace spanned by $G$ is not true, then as Ghosh & Bellemare [20] show, TD can diverge). Now, given $G$ satisfies this spectral radius condition, $\zeta_k$ would converge to $\theta^*$ within a timescale of $\mathcal{O}\left(\frac{1}{\eta}\right)$ within this subspace, which as Blanc et al. [6] put it is the strength of the "mean-reversion" term. On the remaining directions (note that $d \gg n$), the dynamics is guided by the regularizer, although with a smaller weight of $\eta^2$.

## C.7   ADDITIONAL PROOF DETAILS: CONCENTRATING OTHER TERMS

We first concentrate the terms in Equation C.25. The cumulative effect of the second term in Equation C.25 is given by:

$$\eta \sum_{j \le k}(I - \eta G)^{j-k}\nabla Q_i \text{tr}\left[(\nabla^2 Q_i - \gamma \nabla^2 Q_i')\Sigma_M^*\right] \quad \text{(C.30)}$$

$$\le \eta \sum_{j \le k}(I - \eta G)^{j-k}\nabla Q_i \cdot \mathcal{O}\left(\mathscr{L}_2(1+\gamma)\sigma\right) \le \mathcal{O}\left(\eta\sqrt{\frac{k}{\eta}}\omega_0 C_0 \mathscr{L}_2(1+\gamma)\sigma\right), \quad \text{(C.31)}$$

which follows from the fact that $\nabla^2 Q_i$ is $\mathscr{L}_2$-Lipschitz, and using Lemma C.3 for contracting the remaining terms.

Next, we turn to concentrating the second term in Equation C.24. This term corresponds to the contribution of difference between the empirical covariance matrix $\nu_k \nu_k^\top$ and the asymptotic covariance matrix $\zeta^*\zeta^{*\top}$. We expand this term below using the form of $G$ from Lemma C.1, and bound it one by one.

$$\sum_{j \le k}(I - \eta G)^{k-j}\nabla G([\nu_k, \nu_k] - [\zeta^*, \zeta^*]) \quad \text{(C.32)}$$

$$= \sum_{j \le k}\sum_i (I - \eta G)^{k-j}\nabla^2 Q_i\left(\nu_k \nu_k - \zeta^*\zeta^{*\top}\right)(\nabla Q_i - \gamma \nabla Q_i') + \mathcal{O}\left(\sqrt{\eta k}\omega_0 C_0 \mathscr{L}_2(1+\gamma)\sigma\right)$$

$$\quad \text{(C.33)}$$

Now, we note that the term $\Delta_{k+1} := \nu_{k+1}\nu_{k+1}^\top - \zeta^*\zeta^{*\top}$ can itself be written as a recursion:

$$\Delta_{k+1} = (I - \eta G)(\Delta_k)(I - \eta G)^\top + \underbrace{(I - \eta G)\zeta_k \varepsilon^\top + \varepsilon\zeta_k^\top(I - \eta G)^\top}_{A_k} + \underbrace{\varepsilon\varepsilon^\top - \eta M}_{B_k} \quad \text{(C.34)}$$

Expanding the term $\Delta_{k+1}$ in terms of a summation over $k$, and plugging it into the expression from Equation C.33 we get

$$\sum_i \sum_{j \leq k} (I - \eta G)^{k-j} \nabla^2 Q_i (I - \eta G)^j \Delta_0 (I - \eta G^\top)^j \tag{C.35}$$

$$+ \sum_i \sum_{j \leq k} \sum_{p \leq j} (I - \eta G)^{k-j} \nabla^2 Q_i (I - \eta G)^{j-p-1} (A_p + B_p)(I - \eta G^\top)^{j-p-1}$$

Now by noting that if $G$ and $\nabla Q_i$ are $(\omega, C_0)$-aligned, then so are $G^\top$ and $\nabla Q_i$, we can finish the proof by repeating the calculations used by Damian et al. [11] (Appendix B, Equations 67-73) to bound the terms in Equation C.35 by $\mathcal{O}(\sqrt{\eta k})$, but with an additional factor of $\omega^2 C_0^2$.

**Term (b) in Equation C.22.** When $\mathscr{C}$ is small enough, we can bound the term (b) using $\mathcal{O}(\sqrt{\eta k})$, similar to Damian et al. [11].

## D    PROOF OF PROPOSITION 3.2

In this section, we will prove Proposition 3.2. First, we refer to Proposition 3.1 in Ghosh & Bellemare [20], which shows that TD-learning is stable and converges if and only if the matrix $M_\phi = \Phi^\top (\Phi - \gamma \Phi')$ has eigenvalues with all positive real entries. Now note that if,

$$\sum_{\mathbf{s},\mathbf{a}} \phi(\mathbf{s},\mathbf{a})^\top \phi(\mathbf{s},\mathbf{a}) \leq \gamma \sum_{\mathbf{s},\mathbf{a},\mathbf{s}'} \phi(\mathbf{s}',\mathbf{a}')^\top \phi(\mathbf{s},\mathbf{a}) \tag{D.1}$$

$$\implies \text{trace}\left(\Phi^\top \Phi\right) \leq \gamma \text{trace}\left(\Phi^\top \Phi'\right) \tag{D.2}$$

$$\implies \text{trace}\left[\Phi^\top (\Phi - \gamma \Phi')\right] \leq 0. \tag{D.3}$$

Since the trace of a real matrix is the sum of real components of eigenvalues, if for a given matrix $M$, $\text{trace}(M) \leq 0$, then there exists atleast one eigenvalue $\lambda_i$ such that $\text{Re}(\lambda_i) \leq 0$. If $\lambda_i < 0$, then the learning dynamics of TD would diverge, while if $\lambda_i = 0$ for all $i$, then learning will not contract towards the TD fixed point. This concludes the proof of this result.

## E    EXPERIMENTAL DETAILS OF APPLYING DR3

In this section, we discuss the practical experimental details and hyperparameters in applying our method, DR3 to various offline RL methods. We first discuss an overview of the offline RL methods we considered in this paper, and then provide a discussion of hyperparameters for DR3.

### E.1    BACKGROUND ON VARIOUS OFFLINE RL ALGORITHMS

In this paper, we consider four base offline RL algorithms that we apply DR3 on. These methods are detailed below:

**REM**. Random ensemble mixture [2] is an uncertainty-based offline RL algorithm uses multiple parameterized Q-functions to estimate the Q-values. During the Bellman backup, REM computes a random convex combination of the target Q-values and then trains the Q-function to match this randomized target estimate. The randomized target value estimate provides a robust estimate of target values, and delays unlearning and performance degradation that we typically see with standard DQN-style algorithms in the offline setting. For instantiating REM, we follow the instantiation provided by the authors and instantiate a multi-headed Q-function with 200 heads, each of which serves as an estimate of the target value. These multiple heads branch off the last-but-one layer features of the base Q-network. The objective for REM is given by:

$$\min_\theta \mathbb{E}_{\mathbf{s},\mathbf{a},r,\mathbf{s}' \sim \mathcal{D}} \left[ \mathbb{E}_{\alpha_1,\ldots,\alpha_K \sim \Delta} \left[ \ell_\lambda \left( \sum_k \alpha_k Q_\theta^k(\mathbf{s},\mathbf{a}) - r - \gamma \max_{\mathbf{a}'} \sum_k \alpha_k Q_{\theta'}^k(\mathbf{s}',\mathbf{a}') \right) \right] \right] \tag{E.1}$$

where $l_\lambda$ denotes the Huber loss while $P_\Delta$ denotes the probability distribution over the standard (K 1)-simplex.

**CQL**. Conservative Q-learning [27] is an offline RL algorithm that learns a conservative value function such that the estimated performance of the policy under this learned value function lower-bounds its true value. CQL modifies the Q-function training to incorporate a term that minimizes the overestimated Q-values in expectation, while maximizing the Q-values observed in the dataset, in addition to standard TD error. This CQL regularizer is typically multiplied by a coefficient $\alpha$, and we pick $\alpha = 0.1$ for all our Atari experiments following Kumar et al. [28] and $\alpha = 5.0$ for all our kitchen and antmaze D4RL experiments. Using $\overline{y}_k(\mathbf{s}, \mathbf{a})$ to denote the target values computed via the Bellman backup (we use actor-critic backup for D4RL experiments and the $\max_{\mathbf{a}'}$ backup for standard Q-learning in our Atari experiments following Kumar et al. [27]), the objective for training CQL is given by:

$$\min_Q \alpha \left( \mathbb{E}_{\mathbf{s}\sim\mathcal{D}} \left[ \log \sum_{\mathbf{a}} \exp(Q(\mathbf{s}, \mathbf{a})) \right] - \mathbb{E}_{\mathbf{s},\mathbf{a}\sim\mathcal{D}} \left[ Q(\mathbf{s}, \mathbf{a}) \right] \right) + \frac{1}{2}\mathbb{E}_{\mathbf{s},\mathbf{a},\mathbf{s}'\sim\mathcal{D}} \left[ (Q(\mathbf{s}, \mathbf{a}) - \overline{y}_k(\mathbf{s}, \mathbf{a}))^2 \right].$$

The deep Q-network utilized by us is a ReLU network with four hidden layers of size $(256, 256, 256, 256)$ for the D4RL experiments, while for Atari we utilize the standard convolutional neural network from Agarwal et al. [2], Kumar et al. [28] with 3 convolutional layers borrowed from the nature DQN network and then a hidden feedforward layer of size $512$.

**BRAC**. Behavior-regularized actor-critic [49] is a policy-constraint based actor-critic offline RL algorithm which regularizes the policy to stay close to the behavior policy $\pi_\beta$ to prevent the selection of "out-of-distribution" actions. In addition, BRAC subtracts this divergence estimate from the target Q-values when performing the backup, to specifically penalize target values that come from out-of-distribution action inputs at the next state $(\mathbf{s}', \mathbf{a}')$.

Q-function:  $\min_\theta \ \mathbb{E}_{\mathbf{s},\mathbf{a}\sim\mathcal{D}} \left[ \left( r(\mathbf{s}, \mathbf{a}) + \gamma \mathbb{E}_{\mathbf{a}'\sim\pi_\phi(\cdot|\mathbf{s}')}[\bar{Q}_\theta(\mathbf{s}', \mathbf{a}') + \beta \log \hat{\pi}_\beta(\mathbf{a}'|\mathbf{s}')] - Q_\theta(\mathbf{s}, \mathbf{a}) \right)^2 \right].$

Policy:  $\max_\phi \ \mathbb{E}_{\mathbf{s}\sim\mathcal{D},\mathbf{a}\sim\pi_\phi(\cdot|\mathbf{s})} \left[ Q_\theta(\mathbf{s}, \mathbf{a}) + \beta \log \hat{\pi}_\beta(\mathbf{a}|\mathbf{s}) - \alpha \log \pi_\phi(\mathbf{a}|\mathbf{s}) \right].$  (E.2)

**COG**. COG [38] is an algorithmic framework for utilizing large, unlabeled datasets of diverse behavior to learn generalizable policies via offline RL. Similar to real-world scenarios where large unlabeled datasets are available alongside limited task-specific data, the agent is provided with two types of datasets. The task-specific dataset consists of behavior relevant for the task, but the prior dataset can consist of a number of random or scripted behaviors being executed in the same environment/setting. The goal in this task is to actually stitch together relevant and overlapping parts of different trajectories to obtain a good policy that can work from a new initial condition that was not seen in a trajectory that actually achieved the reward. COG utilizes CQL as the base offline RL algorithm, and following Singh et al. [38], we fix the hyperparameter $\alpha = 1.0$ in the CQL part for both base COG and COG + DR3. All other hyperparameters including network sizes, etc are kept fixed as the prior work Singh et al. [38] as well.

### E.2  TASKS AND ENVIRONMENTS USED

**Atari 2600 games used**. For all our experiments, we used the same set of 17 games utilized by Kumar et al. [28] to test rank collapse. In the case of Atari, we used the 5 standard games (ASTERIX, QBERT, PONG, SEAQUEST, BREAKOUT) for tuning the hyperparameters, a strategy followed by several prior works [22, 2, 28]. The 17 games we test on are:  ASTERIX, QBERT, PONG, SEAQUEST, BREAKOUT, DOUBLE DUNK, JAMES BOND, MS. PACMAN, SPACE INVADERS, ZAXXON, WIZARD OF WOR, YARS' REVENGE, ENDURO, ROAD RUNNER, BEAMRIDER, DEMON ATTACK, ICE HOCKEY.

Following Agarwal et al. [3], we report interquartile mean (IQM) normalized scores across all runs as mean scores can be dominated by performance on a few outlier tasks while median is independent of performance on all except 1 task – zero score on half of the tasks would not affect the median. IQM which corresponds to 25% trimmed mean and considers the performance on middle 50% of the runs. IQM interpolates between mean and median, which correspond to 0% and almost 50% trimmed means across runs.

**D4RL tasks used.** For our experiments on D4RL, we utilize the Gym-MuJoCo-v0 environments for evaluating BRAC, since BRAC performed somewhat reasonably on these domains [18], whereas we use the harder AntMaze and Franka Kitchen domains for evaluating CQL, since these domains are challenging for CQL [27].

Table E.1: **Hyperparameters used by the offline RL Atari agents in our experiments.** Following Agarwal et al. [2], the Atari environments used by us are stochastic due to sticky actions, *i.e.*, there is a 25% chance at every time step that the environment will execute the agents previous action again, instead of the new action commanded. We report offline training results with same hyperparameters over 5 random seeds of the offline dataset, game simulator and network initialization.

| Hyperparameter | Setting (for both variations) |
|---|---|
| Sticky actions | Yes |
| Sticky action probability | 0.25 |
| Grey-scaling | True |
| Observation down-sampling | (84, 84) |
| Frames stacked | 4 |
| Frame skip (Action repetitions) | 4 |
| Reward clipping | [-1, 1] |
| Terminal condition | Game Over |
| Max frames per episode | 108K |
| Discount factor | 0.99 |
| Mini-batch size | 32 |
| Target network update period | every 2000 updates |
| Training environment steps per iteration | 250K |
| Update period every | 4 environment steps |
| Evaluation $\epsilon$ | 0.001 |
| Evaluation steps per iteration | 125K |
| $Q$-network: channels | 32, 64, 64 |
| $Q$-network: filter size | $8 \times 8, 4 \times 4, 3 \times 3$ |
| $Q$-network: stride | 4, 2, 1 |
| $Q$-network: hidden units | 512 |

**Robotic manipulation tasks from COG [38].** These tasks consist of a 6-DoF WidowX robot, placed in front of two drawers and a larger variety of objects. The robot can open or close a drawer, grasp objects from inside the drawer or on the table, and place them anywhere in the scene. The task here consists of taking an object out of a drawer. A reward of +1 is obtained when the object has been taken out, and zero otherwise. There are two variants of this domain: **(1)** in the first variant, the drawer starts out closed, the top drawer starts out open (which blocks the handle for the lower drawer), and an object starts out in front of the closed drawer, which must be moved out of the way before opening, and **(2)** in the second variant, the drawer is blocked by an object, and this object must be removed before the drawer can be opened and the target object can be grasped from the drawer. The prior data for this environment is collected from a collection of scripted randomized policies. These policies are capable of opening and closing both drawers with 40-50% success rates, can grasp objects in the scene with about a 70% success rate, and place those objects at random places in the scene (with a slight bias for putting them in the tray).

### E.3    THE DR3 REGULARIZER COEFFICIENT

We utilize identical hyperparameters of the base offline RL algorithms when DR3 is used, where the base hyper-parameters correspond to the ones provided in the corresponding publications. DR3 requires us to tune the additional coefficient $c_0$, that weights the DR3 explicit regularizer term. In order to find this value on our domains, we followed the tuning strategy typically followed on Atari, where we evaluated four different values of $c_0 \in \{0.001, 0.01, 0.03, 0.3\}$ on 5 games (ASTERIX, SEAQUEST, BREAKOUT, PONG and SPACEINVADERS) on the 5% replay dataset settings, picked $c_0$ that wprked best on just these domains, and used it to report performance on all 17 games, across all dataset settings (1% replay and 10% initial replay) in Section 5. This protocol is standard in Atari and has been used previously in Agarwal et al. [2], Gulcehre et al. [22], Kumar et al. [28] in the context of offline RL. The value of the coefficient found using this strategy was $c_0 = 0.001$ for REM and $c_0 = 0.03$ for CQL.

For CQL on D4RL, we ran DR3 with multiple values of $c_0 \in \{0.0001, 0.001, 0.01, 0.5, 1.0, 10.0\}$, and picked the smallest value of $c_0$ which did not lead to eventually divergent (either negatively diverging or positively diverging) Q-values, in average. For the antmaze domains, this corresponded to $c_0 = 0.001$ and for the FrankaKitchen domains, this corresponded to $c_0 = 1.0$.

# F    COMPLETE RESULTS ON ALL DOMAINS

In this section, we present the results obtained by running DR3 on the Atari and D4RL domains which were not discussed in the main paper due to lack of space. We first understand the effect of applying DR3 on BRAC [49], which was missing from the main paper, and then present the per-game Atari results.

Table F.1: Normalized interquartile mean  (IQM) final performance (last iteration return) of CQL, CQL + DR3, REM and REM + DR3 after 6.5M gradient steps for the 1% setting and 12.5M gradient steps for the 5%, 10% settings. Intervals in brackets show 95% CIs computed using stratified percentile bootstrap [3]

| Data | CQL | CQL + DR3 | REM | REM + DR3 |
|------|-----|-----------|-----|-----------|
| 1%   | 44.4 (31.0, 54.3) | **61.6** (39.1, 71.5) | 0.0 (-0.7, 0.1) | **13.1** (9.9, 18.3) |
| 5%   | 89.6 (67.9, 98.1) | **100.2** (90.6, 102.7) | 3.9 (3.1, 7.6) | **74.8** (59.6, 84.4) |
| 10%  | 57.4 (53.2, 62.4) | **67.0** (62.8, 73.0) | 24.9 (15.0, 29.1) | **72.4** (65.7, 81.7) |

Table F.2: **Performance of DR3 when applied in conjunction with BRAC [49].** Note that DR3 attains a larger final performance (at the end of 2M steps of training) as well as a higher average performance (i.e. stability score) across all iterations of training.

| Task | Average Performance across Iterations | | Final Performance | |
|------|----------------|----------------|----------------|----------------|
|      | BRAC | BRAC + DR3 | BRAC | BRAC + DR3 |
| halfcheetah-expert-v0 | $1.7 \pm 1.9$ | $49.9 \pm 16.7$ | $2.1 \pm 3.3$ | $71.5 \pm 24.9$ |
| halfcheetah-medium-v0 | $43.5 \pm 0.2$ | $43.2 \pm 0.2$ | $45.1 \pm 0.8$ | $44.9 \pm 0.6$ |
| halfcheetah-medium-expert-v0 | $17.0 \pm 5.4$ | $6.0 \pm 5.5$ | $24.8 \pm 9.3$ | $6.7 \pm 7.3$ |
| halfcheetah-random-v0 | $24.4 \pm 0.4$ | $18.4 \pm 0.3$ | $24.9 \pm 0.8$ | $18.2 \pm 1.0$ |
| halfcheetah-medium-replay-v0 | $44.9 \pm 0.3$ | $44.1 \pm 0.4$ | $45.0 \pm 1.4$ | $44.9 \pm 0.5$ |
| hopper-expert-v0 | $15.7 \pm 1.5$ | $21.8 \pm 3.2$ | $16.6 \pm 6.0$ | $20.8 \pm 5.3$ |
| hopper-medium-v0 | $32.8 \pm 1.4$ | $46.3 \pm 7.1$ | $36.2 \pm 1.7$ | $58.3 \pm 13.7$ |
| hopper-medium-expert-v0 | $40.2 \pm 5.7$ | $37.0 \pm 2.9$ | $31.7 \pm 11.8$ | $21.8 \pm 4.9$ |
| hopper-random-v0 | $11.7 \pm 0.0$ | $11.2 \pm 0.0$ | $12.2 \pm 0.0$ | $11.1 \pm 0.0$ |
| hopper-medium-replay-v0 | $31.6 \pm 0.3$ | $30.3 \pm 0.8$ | $31.3 \pm 1.2$ | $36.1 \pm 5.7$ |
| walker2d-expert-v0 | $25.5 \pm 14.4$ | $33.6 \pm 11.8$ | $54.0 \pm 31.0$ | $60.6 \pm 20.2$ |
| walker2d-medium-v0 | $81.3 \pm 0.3$ | $80.8 \pm 0.2$ | $83.8 \pm 0.2$ | $83.4 \pm 0.3$ |
| walker2d-medium-expert-v0 | $5.8 \pm 5.2$ | $6.4 \pm 3.4$ | $22.4 \pm 22.0$ | $39.5 \pm 23.3$ |
| walker2d-random-v0 | $1.4 \pm 0.8$ | $1.7 \pm 0.9$ | $0.0 \pm 0.1$ | $2.9 \pm 2.1$ |
| walker2d-medium-replay-v0 | $26.1 \pm 6.4$ | $47.4 \pm 4.1$ | $11.7 \pm 7.0$ | $38.7 \pm 9.6$ |

Table F.4: **Mean evaluation returns per Atari game across 5 runs with standard deviations for 1% dataset**. The coefficient for DR3 is 0.03 with a CQL coefficient of 1.0. The average performance is computed over 20 checkpoints spaced uniformly over training for 100 iterations where 1 iteration corresponds to 62,500 gradient updates.

| Game | Final Performance | | Average Performance across Iterations | |
| --- | --- | --- | --- | --- |
| | CQL | CQL + DR3 | CQL | CQL + DR3 |
| Asterix | $656.9 \pm 91.0$ | $821.4 \pm 75.1$ | $650.2 \pm 65.3$ | $814.1 \pm 25.1$ |
| Breakout | $23.9 \pm 3.8$ | $32.0 \pm 3.2$ | $23.8 \pm 0.5$ | $32.8 \pm 3.1$ |
| Pong | $16.7 \pm 1.7$ | $14.2 \pm 3.3$ | $15.7 \pm 2.0$ | $15.1 \pm 2.3$ |
| Seaquest | $449.0 \pm 11.0$ | $446.6 \pm 26.9$ | $474.5 \pm 30.3$ | $456.1 \pm 17.0$ |
| Qbert | $8033.8 \pm 1513.2$ | $9162.7 \pm 993.6$ | $7980.0 \pm 379.9$ | $9000.7 \pm 225.2$ |
| SpaceInvaders | $386.0 \pm 123.2$ | $351.9 \pm 77.1$ | $371.7 \pm 47.5$ | $440.6 \pm 29.6$ |
| Zaxxon | $829.4 \pm 813.3$ | $1757.4 \pm 879.4$ | $834.6 \pm 504.0$ | $1634.0 \pm 673.9$ |
| YarsRevenge | $11848.2 \pm 2977.7$ | $16011.3 \pm 1409.0$ | $15077.9 \pm 1301.9$ | $17741.6 \pm 613.6$ |
| RoadRunner | $37000.7 \pm 1148.5$ | $24928.7 \pm 7484.5$ | $35899.9 \pm 653.1$ | $32063.3 \pm 1011.4$ |
| MsPacman | $1869.8 \pm 167.2$ | $2245.7 \pm 193.8$ | $1991.9 \pm 55.1$ | $2224.1 \pm 80.8$ |
| BeamRider | $780.3 \pm 64.5$ | $617.9 \pm 25.1$ | $782.0 \pm 36.1$ | $619.9 \pm 20.9$ |
| Jamesbond | $558.5 \pm 124.8$ | $460.5 \pm 102.0$ | $524.6 \pm 118.5$ | $484.2 \pm 89.4$ |
| Enduro | $198.4 \pm 34.2$ | $253.5 \pm 14.2$ | $259.8 \pm 16.4$ | $276.1 \pm 16.9$ |
| WizardOfWor | $771.1 \pm 358.2$ | $904.6 \pm 343.7$ | $833.7 \pm 168.4$ | $935.2 \pm 174.4$ |
| IceHockey | $-8.7 \pm 1.3$ | $-7.8 \pm 0.9$ | $-8.8 \pm 0.9$ | $-7.9 \pm 0.7$ |
| DoubleDunk | $-15.1 \pm 1.9$ | $-14.0 \pm 2.8$ | $-15.3 \pm 0.9$ | $-14.5 \pm 1.0$ |
| DemonAttack | $1970.2 \pm 161.3$ | $386.2 \pm 75.3$ | $1338.8 \pm 298.4$ | $414.0 \pm 46.0$ |

Table F.5: **Mean evaluation returns per Atari game across 5 runs with standard deviations for 5% dataset.** The coefficient for DR3 is 0.03 with a CQL coefficient of 0.1. The average performance is computed over 20 checkpoints spaced uniformly over training for 200 iterations where 1 iteration corresponds to 62,500 gradient updates.

| Game | Final Performance | | Average Performance across Iterations | |
| --- | --- | --- | --- | --- |
| | CQL | CQL + DR3 | CQL | CQL + DR3 |
| Asterix | $1798.2 \pm 168.6$ | $3318.5 \pm 301.7$ | $1812.7 \pm 64.0$ | $3790.5 \pm 218.0$ |
| Breakout | $94.1 \pm 44.4$ | $166.0 \pm 23.1$ | $105.1 \pm 10.4$ | $196.5 \pm 4.4$ |
| Pong | $13.1 \pm 4.2$ | $17.9 \pm 1.1$ | $15.2 \pm 1.3$ | $17.4 \pm 1.2$ |
| Seaquest | $1815.9 \pm 722.8$ | $2030.7 \pm 822.8$ | $1382.3 \pm 258.1$ | $3722.3 \pm 969.5$ |
| Qbert | $10595.7 \pm 1648.5$ | $9605.6 \pm 1593.5$ | $9552.0 \pm 925.6$ | $10830.7 \pm 783.1$ |
| SpaceInvaders | $758.9 \pm 56.9$ | $1214.6 \pm 281.8$ | $662.0 \pm 58.1$ | $1323.7 \pm 94.4$ |
| Zaxxon | $1501.0 \pm 1165.7$ | $4250.1 \pm 626.2$ | $1508.8 \pm 437.5$ | $3556.5 \pm 531.3$ |
| YarsRevenge | $24036.7 \pm 3370.6$ | $17124.7 \pm 2125.6$ | $22733.1 \pm 1175.3$ | $18339.8 \pm 1299.7$ |
| RoadRunner | $40728.4 \pm 3318.9$ | $38432.6 \pm 1539.7$ | $42338.4 \pm 471.4$ | $41260.2 \pm 1008.6$ |
| MsPacman | $2975.9 \pm 522.1$ | $2790.6 \pm 353.1$ | $2923.6 \pm 251.3$ | $3101.2 \pm 381.6$ |
| BeamRider | $1897.6 \pm 473.7$ | $785.8 \pm 43.5$ | $2218.5 \pm 242.4$ | $775.9 \pm 12.5$ |
| Jamesbond | $108.8 \pm 49.1$ | $96.8 \pm 43.2$ | $76.5 \pm 4.6$ | $106.1 \pm 34.8$ |
| Enduro | $764.3 \pm 168.7$ | $938.5 \pm 63.9$ | $797.7 \pm 47.8$ | $923.2 \pm 40.3$ |
| WizardOfWor | $943.2 \pm 380.3$ | $612.0 \pm 343.3$ | $1004.3 \pm 314.7$ | $1007.4 \pm 313.2$ |
| IceHockey | $-17.3 \pm 0.6$ | $-15.0 \pm 0.7$ | $-16.6 \pm 0.5$ | $-12.0 \pm 0.3$ |
| DoubleDunk | $-18.1 \pm 1.5$ | $-16.2 \pm 1.7$ | $-17.3 \pm 1.0$ | $-16.0 \pm 1.6$ |
| DemonAttack | $4055.8 \pm 499.7$ | $8517.4 \pm 1065.9$ | $4062.4 \pm 465.8$ | $8396.7 \pm 689.4$ |

Table F.6: **Mean returns per Atari game across 5 runs with standard deviations for initial 10% dataset.** The coefficient for DR3 is 0.03 with a CQL coefficient of 0.1. The average performance is computed over 20 checkpoints spaced uniformly over training for 200 iterations.

| Game | Final Performance | | Average Performance across Iterations | |
|---|---|---|---|---|
| | CQL | CQL + DR3 | CQL | CQL + DR3 |
| Asterix | $2803.9 \pm 294.6$ | $3906.2 \pm 521.3$ | $2903.2 \pm 217.7$ | $4692.2 \pm 377.0$ |
| Breakout | $64.7 \pm 7.3$ | $70.8 \pm 5.5$ | $65.6 \pm 5.7$ | $75.4 \pm 6.0$ |
| Pong | $5.3 \pm 6.8$ | $5.5 \pm 6.2$ | $7.3 \pm 5.0$ | $8.1 \pm 5.2$ |
| Seaquest | $222.3 \pm 219.5$ | $1313.0 \pm 220.0$ | $704.9 \pm 254.5$ | $1327.9 \pm 250.0$ |
| Qbert | $4803.2 \pm 489.5$ | $5395.3 \pm 1003.6$ | $4492.5 \pm 240.8$ | $4708.5 \pm 463.0$ |
| SpaceInvaders | $704.9 \pm 121.5$ | $938.1 \pm 80.3$ | $737.8 \pm 23.8$ | $902.1 \pm 60.0$ |
| Zaxxon | $231.6 \pm 450.9$ | $836.8 \pm 434.7$ | $394.4 \pm 385.1$ | $725.7 \pm 370.3$ |
| YarsRevenge | $13076.2 \pm 2427.0$ | $12413.9 \pm 2869.7$ | $12493.2 \pm 543.6$ | $12395.6 \pm 1044.2$ |
| RoadRunner | $45063.5 \pm 1749.7$ | $45336.9 \pm 1366.7$ | $45522.7 \pm 1068.1$ | $44808.0 \pm 911.7$ |
| MsPacman | $2459.5 \pm 381.3$ | $2427.5 \pm 191.3$ | $2528.1 \pm 149.2$ | $2488.3 \pm 109.8$ |
| BeamRider | $4200.7 \pm 470.2$ | $3468.0 \pm 238.0$ | $4729.5 \pm 94.8$ | $3344.3 \pm 289.0$ |
| Jamesbond | $84.6 \pm 25.4$ | $89.7 \pm 15.6$ | $108.7 \pm 34.1$ | $111.7 \pm 10.9$ |
| Enduro | $946.7 \pm 289.7$ | $1160.2 \pm 81.5$ | $1013.9 \pm 29.7$ | $1136.2 \pm 32.5$ |
| WizardOfWor | $520.4 \pm 451.2$ | $764.7 \pm 250.0$ | $499.8 \pm 238.5$ | $792.2 \pm 101.3$ |
| IceHockey | $-18.1 \pm 0.7$ | $-16.0 \pm 1.3$ | $-17.6 \pm 0.5$ | $-15.2 \pm 1.0$ |
| DoubleDunk | $-21.2 \pm 1.1$ | $-20.6 \pm 1.0$ | $-20.6 \pm 0.3$ | $-19.7 \pm 0.5$ |
| DemonAttack | $4145.2 \pm 400.6$ | $7152.9 \pm 723.2$ | $4839.4 \pm 586.7$ | $7278.5 \pm 701.3$ |

Table F.7: **Mean returns per Atari game across 5 runs with standard deviations for 1% dataset.** The coefficient for DR3 is 0.001 while we use a multi-headed REM with 200 Q-heads [2]. The average performance is computed over 20 checkpoints spaced uniformly over training for 100 iterations.

| Game | Final Performance | | Average Performance across Iterations | |
|---|---|---|---|---|
| | REM | REM + DR3 | REM | REM + DR3 |
| Asterix | $240.4 \pm 29.1$ | $405.7 \pm 46.5$ | $304.4 \pm 9.3$ | $413.7 \pm 39.6$ |
| Breakout | $0.7 \pm 0.7$ | $14.3 \pm 2.8$ | $6.3 \pm 1.0$ | $10.3 \pm 1.1$ |
| Pong | $-14.2 \pm 1.7$ | $-7.7 \pm 6.3$ | $-14.1 \pm 2.2$ | $-15.3 \pm 3.0$ |
| Seaquest | $81.0 \pm 78.5$ | $293.3 \pm 191.5$ | $246.6 \pm 49.5$ | $489.9 \pm 128.6$ |
| Qbert | $239.6 \pm 133.2$ | $436.3 \pm 111.5$ | $255.5 \pm 76.0$ | $471.0 \pm 116.5$ |
| SpaceInvaders | $152.8 \pm 27.5$ | $206.6 \pm 77.6$ | $188.6 \pm 5.8$ | $262.7 \pm 22.4$ |
| Zaxxon | $534.9 \pm 731.3$ | $2596.4 \pm 1726.4$ | $1807.9 \pm 478.2$ | $707.7 \pm 577.4$ |
| YarsRevenge | $1452.6 \pm 1631.0$ | $5480.2 \pm 962.3$ | $4018.8 \pm 987.8$ | $7352.0 \pm 574.7$ |
| RoadRunner | $0.0 \pm 0.0$ | $3872.9 \pm 1616.4$ | $1601.2 \pm 637.9$ | $14231.9 \pm 2406.0$ |
| MsPacman | $698.8 \pm 129.5$ | $1275.1 \pm 345.6$ | $690.4 \pm 69.7$ | $860.4 \pm 57.1$ |
| BeamRider | $703.0 \pm 97.4$ | $522.9 \pm 42.2$ | $745.5 \pm 30.7$ | $592.2 \pm 27.7$ |
| Jamesbond | $41.0 \pm 27.0$ | $157.6 \pm 65.0$ | $53.3 \pm 12.1$ | $88.8 \pm 27.2$ |
| Enduro | $0.5 \pm 0.4$ | $132.4 \pm 16.1$ | $21.7 \pm 4.0$ | $197.5 \pm 19.1$ |
| WizardOfWor | $362.5 \pm 321.8$ | $1663.7 \pm 417.8$ | $552.1 \pm 253.1$ | $1460.8 \pm 194.8$ |
| IceHockey | $-16.7 \pm 0.9$ | $-9.1 \pm 5.1$ | $-12.1 \pm 0.8$ | $-4.8 \pm 1.8$ |
| DoubleDunk | $-21.8 \pm 1.0$ | $-17.6 \pm 1.5$ | $-20.4 \pm 0.6$ | $-17.1 \pm 1.6$ |
| DemonAttack | $102.0 \pm 17.3$ | $162.0 \pm 34.7$ | $124.0 \pm 10.7$ | $145.6 \pm 27.2$ |

Table F.8: **Mean returns per Atari game across 5 runs with standard deviations for the 5% dataset.** The coefficient for DR3 is 0.001 while we use a multi-headed REM with 200 Q-heads [2]. The average performance is computed over 20 checkpoints spaced uniformly over training for 200 iterations.

| Game | Final Performance | | Average Performance across Iterations | |
| --- | --- | --- | --- | --- |
| | REM | REM + DR3 | REM | REM + DR3 |
| Asterix | 876.8 ± 201.1 | 2317.0 ± 838.1 | 958.9 ± 50.9 | 1252.6 ± 395.1 |
| Breakout | 15.2 ± 4.9 | 33.4 ± 4.0 | 16.3 ± 3.4 | 17.7 ± 2.4 |
| Pong | 7.5 ± 5.2 | -0.7 ± 9.9 | -4.7 ± 3.0 | -12.0 ± 3.2 |
| Seaquest | 1276.0 ± 417.3 | 2753.6 ± 1119.7 | 1484.3 ± 367.7 | 1602.0 ± 603.7 |
| Qbert | 2421.4 ± 1841.8 | 7417.0 ± 2106.7 | 1330.7 ± 431.0 | 4045.8 ± 898.9 |
| SpaceInvaders | 431.5 ± 23.3 | 443.5 ± 67.4 | 349.5 ± 22.6 | 362.1 ± 33.6 |
| Zaxxon | 6738.2 ± 966.6 | 1609.7 ± 1814.1 | 3630.7 ± 751.4 | 346.1 ± 512.1 |
| YarsRevenge | 14454.2 ± 1644.4 | 16930.4 ± 2625.8 | 14628.3 ± 1945.1 | 12936.5 ± 1286.0 |
| RoadRunner | 15570.9 ± 12795.6 | 46601.6 ± 2617.2 | 22740.3 ± 1977.2 | 33554.1 ± 1880.4 |
| MsPacman | 1272.2 ± 215.3 | 2303.1 ± 202.7 | 1147.7 ± 126.1 | 1438.7 ± 140.4 |
| BeamRider | 1922.5 ± 589.1 | 674.8 ± 21.4 | 886.9 ± 82.1 | 698.3 ± 21.5 |
| Jamesbond | 189.6 ± 77.0 | 130.5 ± 45.7 | 120.2 ± 9.3 | 88.6 ± 41.5 |
| Enduro | 172.7 ± 55.9 | 583.9 ± 108.7 | 236.8 ± 11.3 | 457.7 ± 39.3 |
| WizardOfWor | 838.4 ± 670.0 | 2661.6 ± 371.4 | 1281.3 ± 66.7 | 1863.7 ± 261.2 |
| IceHockey | -9.7 ± 4.2 | -6.5 ± 3.1 | -8.1 ± 0.7 | -4.1 ± 1.5 |
| DoubleDunk | -18.4 ± 0.9 | -17.6 ± 2.6 | -19.6 ± 1.0 | -17.8 ± 1.9 |
| DemonAttack | 507.7 ± 120.1 | 5602.3 ± 1855.5 | 581.6 ± 207.0 | 1452.3 ± 765.0 |

Table F.9: **Mean returns per Atari game across 5 runs with standard deviations for initial 10% dataset.** The coefficient for DR3 is 0.001 while we use a multi-headed REM with 200 Q-heads [2]. The average performance is computed over 20 checkpoints spaced uniformly over training for 200 iterations.

| Game | Final Performance | | Average Performance across Iterations | |
| --- | --- | --- | --- | --- |
| | REM | REM + DR3 | REM | REM + DR3 |
| Asterix | 2254.7 ± 403.6 | 5122.9 ± 328.9 | 2684.6 ± 184.4 | 3432.1 ± 257.5 |
| Breakout | 81.2 ± 13.9 | 96.8 ± 21.2 | 63.5 ± 4.6 | 62.4 ± 6.1 |
| Pong | 8.8 ± 3.1 | 7.6 ± 11.1 | 2.6 ± 2.1 | -2.5 ± 5.6 |
| Seaquest | 1540.2 ± 354.6 | 981.3 ± 605.9 | 1029.5 ± 260.6 | 836.2 ± 234.3 |
| Qbert | 4330.7 ± 250.2 | 4126.2 ± 495.7 | 3478.0 ± 248.0 | 3494.7 ± 380.3 |
| SpaceInvaders | 895.2 ± 68.3 | 799.0 ± 28.3 | 699.7 ± 31.4 | 653.1 ± 21.5 |
| Zaxxon | 950.7 ± 897.4 | 0.0 ± 0.0 | 490.2 ± 306.6 | 0.0 ± 0.0 |
| YarsRevenge | 10913.1 ± 1519.1 | 11924.8 ± 2413.8 | 11508.5 ± 290.0 | 10977.7 ± 1026.9 |
| RoadRunner | 45521.7 ± 2502.1 | 49129.4 ± 1887.9 | 37997.4 ± 638.6 | 41995.2 ± 1482.1 |
| MsPacman | 2177.4 ± 393.0 | 2268.8 ± 455.0 | 1930.5 ± 141.7 | 2126.6 ± 147.6 |
| BeamRider | 2921.7 ± 308.7 | 4154.9 ± 357.2 | 3727.5 ± 304.3 | 2871.0 ± 44.3 |
| Jamesbond | 197.8 ± 73.8 | 149.3 ± 304.5 | 149.0 ± 120.5 | 83.3 ± 162.4 |
| Enduro | 529.5 ± 200.7 | 832.5 ± 65.5 | 584.6 ± 85.3 | 801.8 ± 39.3 |
| WizardOfWor | 606.5 ± 823.2 | 920.0 ± 497.0 | 838.3 ± 343.7 | 926.3 ± 318.5 |
| IceHockey | -4.3 ± 0.6 | -5.9 ± 5.1 | -7.0 ± 1.1 | -5.4 ± 3.7 |
| DoubleDunk | -17.7 ± 3.9 | -19.5 ± 2.5 | -16.9 ± 0.5 | -16.7 ± 1.0 |
| DemonAttack | 6097.9 ± 1251.3 | 9674.7 ± 1600.6 | 4649.1 ± 514.6 | 5141.9 ± 361.4 |

Table F.10: Average returns across 5 runs for the random agent and the average performance of the trajectories in the DQN (Nature) dataset. For Atari normalized scores reported in the paper, the random agent is assigned a score of 0 while the average DQN replay is assigned a score of 100. Note that the random agent scores are also evaluated on Atari 2600 games with sticky actions.

| Game | Random | Average DQN-Replay |
|---|---|---|
| Asterix | 279.1 | 3185.2 |
| Breakout | 1.3 | 104.9 |
| Pong | -20.3 | 14.5 |
| Seaquest | 81.8 | 1597.4 |
| Qbert | 155.0 | 8249.7 |
| SpaceInvaders | 149.5 | 1529.8 |
| Zaxxon | 10.6 | 1854.1 |
| YarsRevenge | 3147.7 | 21015.0 |
| RoadRunner | 15.5 | 38352.3 |
| MsPacman | 248.0 | 3108.8 |
| BeamRider | 362.0 | 4576.4 |
| Jamesbond | 27.6 | 560.3 |
| Enduro | 0.0 | 671.9 |
| WizardOfWor | 686.6 | 1128.5 |
| IceHockey | -9.8 | -8.5 |
| DoubleDunk | -18.4 | -11.3 |
| DemonAttack | 166.0 | 4407.5 |

## F.1 PER-GAME LEARNING CURVES FOR ATARI GAMES

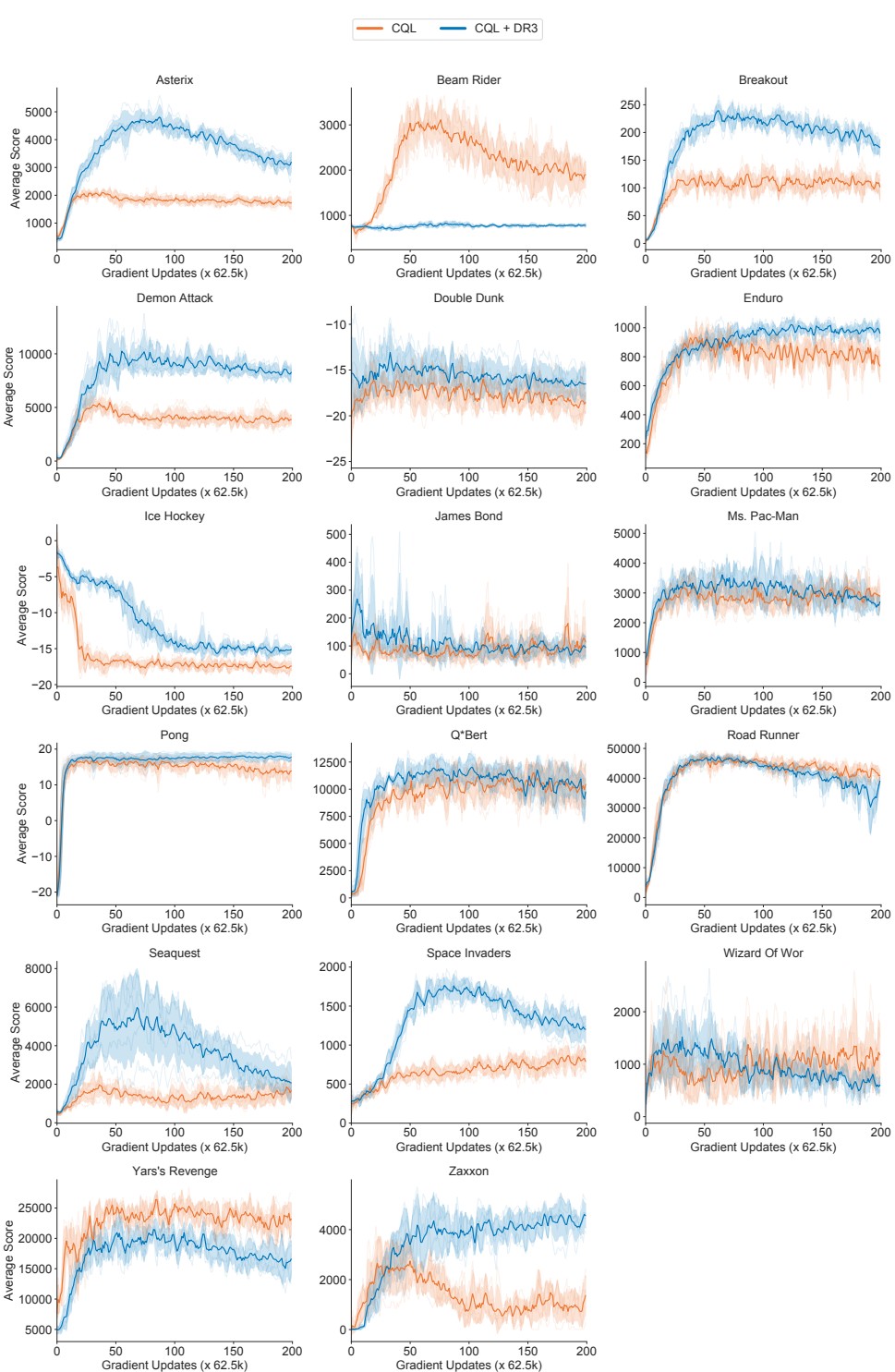

Figure F.1: **Per-game learning curves of CQL and CQL + DR3 on the 5% uniform replay dataset, for which the normalized average learning curve is shown in Figure 4.** Note that CQL + DR3 attains a higher performance than CQL for a majority of games, and rises up to a higher peak.

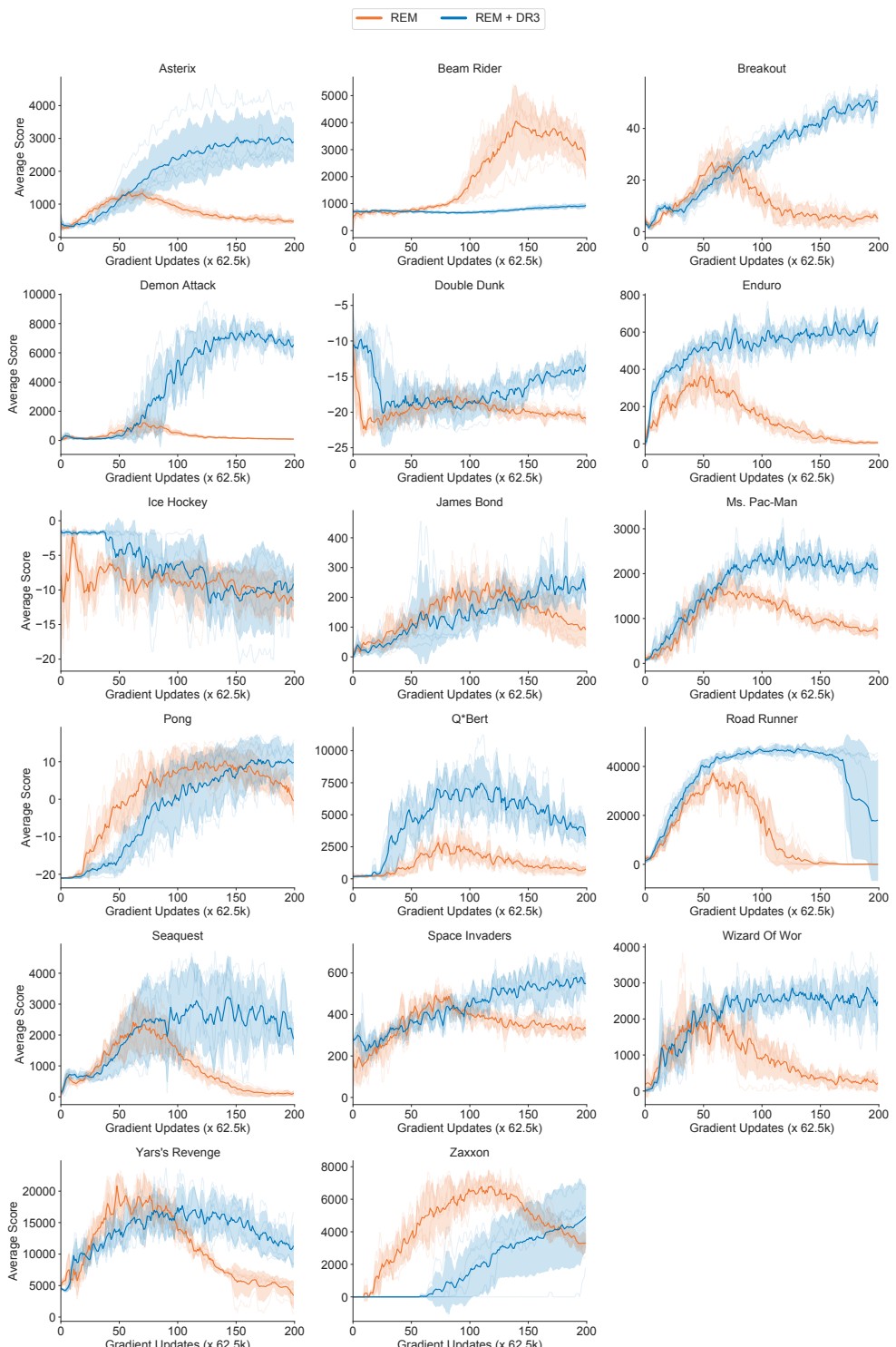

Figure F.2: **Per-game learning curves of REM and REM + DR3 on the 5% uniform replay dataset, for which the normalized average learning curve is shown in Figure 4.** Note that REM + DR3 attains a higher performance than REM for a majority of games.

## F.2  Dot Product Similarities For CQL+DR3 and REM+DR3 on 17 Games

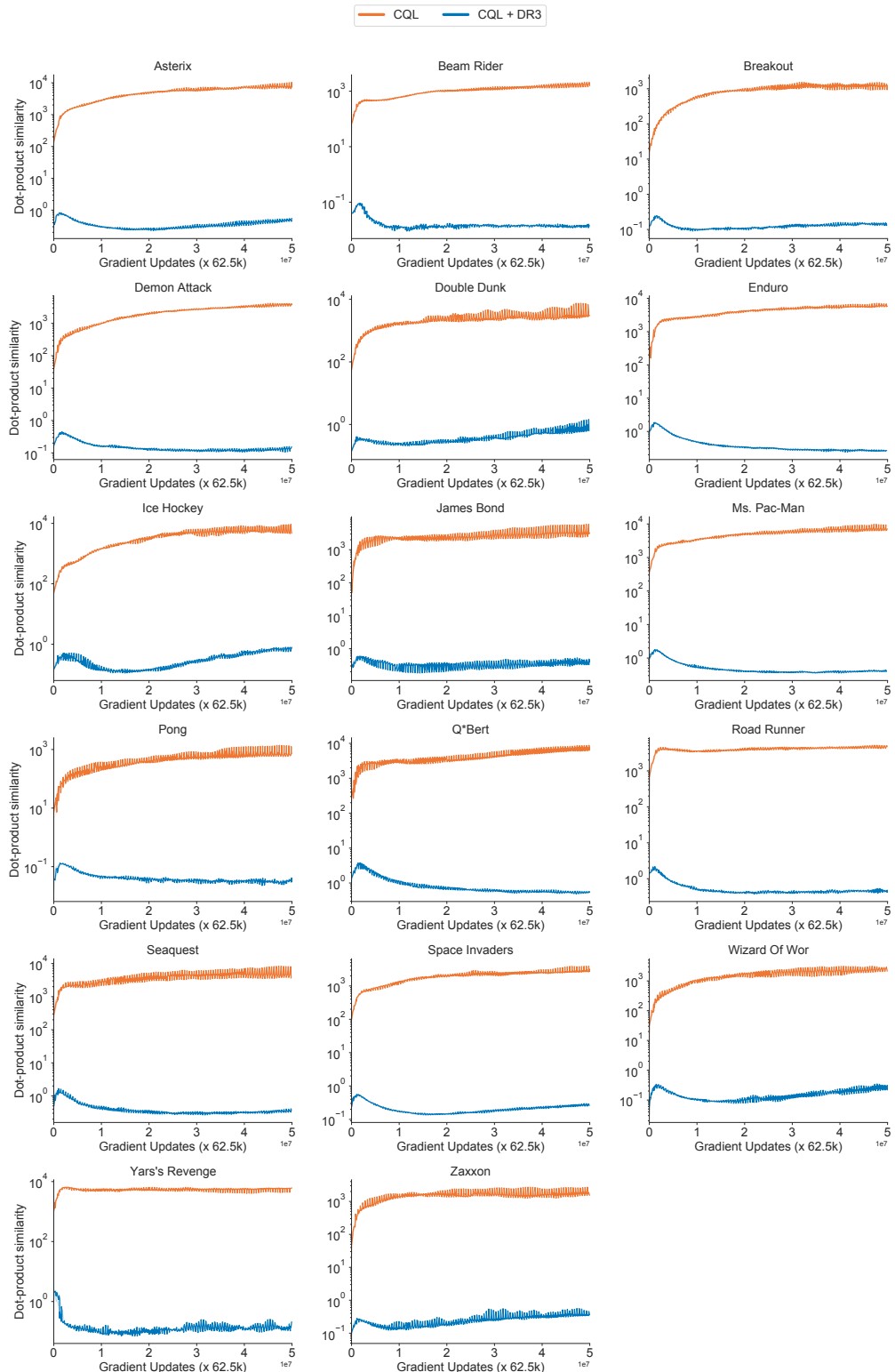

Figure F.3: **Per-game feature dot products (in log scale) of CQL and CQL + DR3 on the 5% uniform replay dataset**. Note that CQL + DR3 attains a smaller value of the feature dot product.

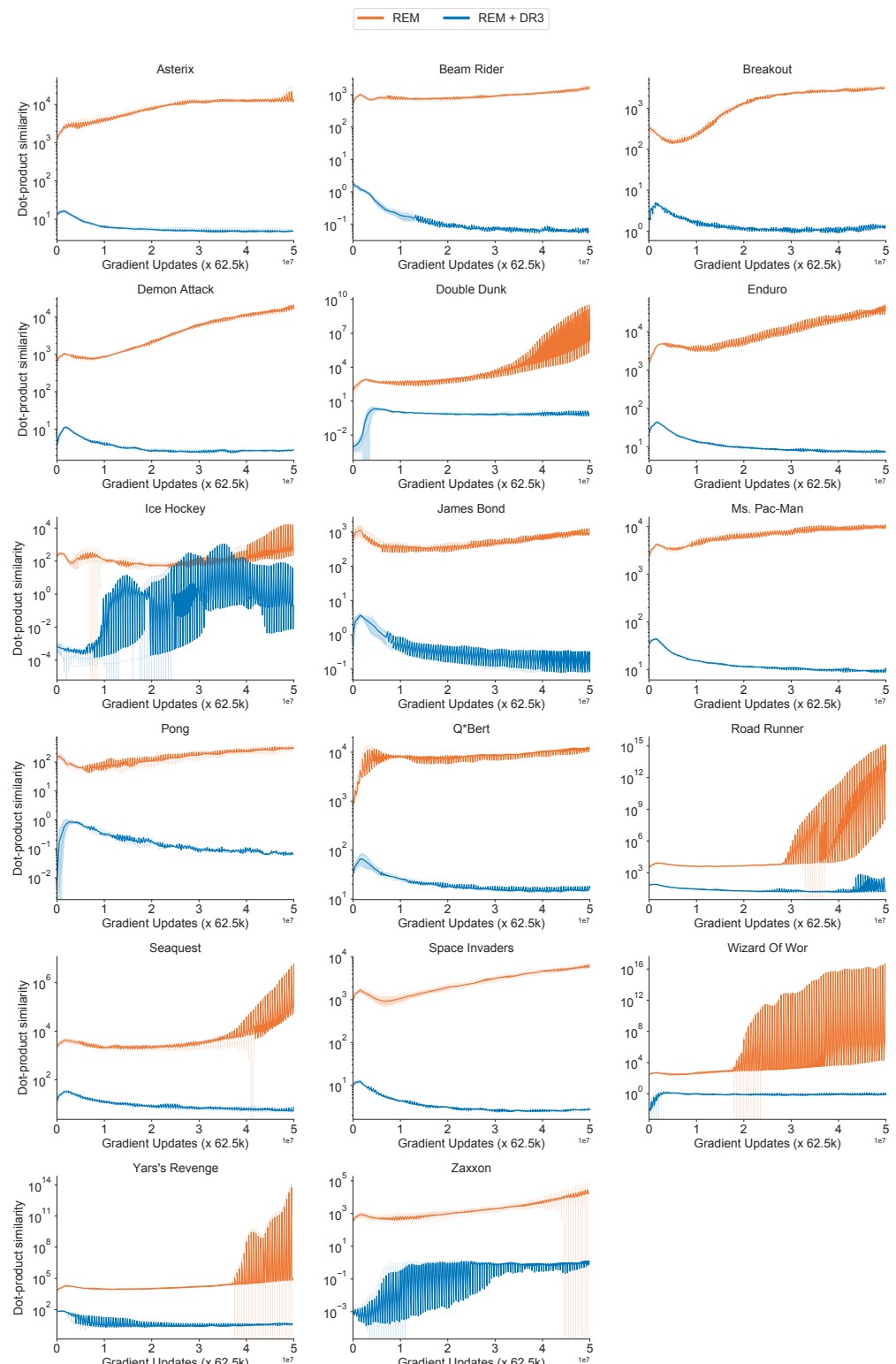

Figure F.4: **Per-game feature dot products (in log scale) of REM and REM + DR3 on the 5% uniform replay dataset** Note that REM + DR3 attains a higher performance than REM for a majority of games. Note that the dot products for REM+DR3 stabilize are small, and decreases for a majority of the training steps for a number of games, or stabilize at a small value.

