# OpenReview forum: "DR3: Value-Based Deep Reinforcement Learning Requires Explicit Regularization"
_ICLR.cc/2022/Conference — ICLR 2022 Spotlight_

### Official Review · Reviewer_oKwL · 2021-10-30

**Correctness:** 4
**Technical Novelty And Significance:** 3
**Empirical Novelty And Significance:** 3
**Recommendation:** 8
**Confidence:** 3

**Main Review:**

Strengths:
1. The identified phenomenon is very critical: When the consecutive state-action pairs are encoded to have similar representations, the learned value functions, as well as corresponding policies, would perform poorly.
2. This paper provides theoretical evidence to explain the feature co-adaptation phenomenon, which meanwhile explains why it is severe for out-of-sample next actions.
3. The empirical evaluation strongly confirms the effectiveness of the proposed regularizer.
Weakness:
1. There are some typos, e.g., "stableif" in Theorem 3.1.
2. The organization can be improved further. It is not easy to follow this story, and there are many preliminaries about offline RL that are indispensable for readers to understand the discussions.

**Summary Of The Paper:**

This paper first empirically notices the feature co-adaptation phenomenon which will be exacerbated in TD-learning with out-of-sample next actions. The authors theoretically show that the implicit regularization of (noisy) SGD encourages maximizing the inner product between the encoded features of current and next state-action pairs, which is responsible for the observed phenomenon. Thus, the authors propose a novel explicit regularizer for offline TD-learning to encourage smaller inner products. Extensive experiments on both video game environments and robotics demonstrate the effectiveness of the proposed regularizer, with various offline TD-learning algorithms as the testbeds.

**Summary Of The Review:**

This is a good paper which studies an important problem from both empirical and theoretical aspects and brings in sufficient novel stuff. I appreciate the contributions made by the authors and recommend this paper.

---

> ### Author Response · Authors · 2021-11-16
> **Thanks you very much!**
>
> Thanks for the valuable feedback and a positive assessment of our work! We are glad that you liked the paper. We have fixed the typos pointed by all the reviewers and will fix the remaining for the final version.
>
> For the preliminaries, we have discussed them in the appendix (space constraint) but would highlight them more prominently for the final. Furthermore, in response to other reviewers' comments, we have added several other results for statistical significance (**Appendix A.6**), new experiments on MuJoCo domains (**Appendix A.3**), and further analysis of the relationship between srank and DR3 (**Appendix A.4**), which further supports our claim regarding the efficacy of DR3.

---

### Official Review · Reviewer_5pgu · 2021-11-02

**Correctness:** 3
**Technical Novelty And Significance:** 3
**Empirical Novelty And Significance:** 3
**Recommendation:** 6
**Confidence:** 3

**Main Review:**

Overall the paper is well written, pedagogical, and the problem at stake is really interesting. But I find the presentation of experimental results misleading which makes me doubt about the performances of the proposed solution.

### Major revisions/questions/remarks
  * Roughly 3 papers out of 5 refer to non peered reviewed papers (arXiv)
  * Which algorithm is used to learn the Q-value in Figure 5? I am assuming it is CQL since the results match the ones for CQL in Kumar et al. 2021
  * "it also alleviates rank collapse, with no apparent term that explicitly increases rank" (Section 5), and "This is potentially suprising, because no term in the practical DR3 regularizer explicitly aims to increase rank" (Appendix A.3):
    * The only srank's behavior exposed in this paper are the ones observed while learning the Q-network with the CQL algorithm. What about DQN, REM, and BRAC? Is the alleviation of rank collapse is specific to CQL+DR3?
    * Looking at the simplified loss the authors based their intuition on, composed of 3 (maybe 4?) terms:
      * 1. TD error: $||r + {\phi^\prime}^\top w$ - $\phi^\top w||^2$;
      * 2. CQL OOD samples regularization: ${\phi^\prime}^\top w$ (at least);
      * 3. Implicit regularization: $||\phi||^2 - \[\[{\phi^\prime}^\top\]\] \phi$;
      * 4. Potential L2 regularization on weight: $||w||^2$ (at least);
      * It looks like adding the explicit regularization is an explicit way to increase rank, even if this regularization is more computationally efficient than Kumar et al. 2020. Am I missing something?
    * Plotting the norms of $\phi$, $\phi^\prime$, and their cosine distance with and without DR3 could be quite informative
  * Comparing Figure 6 (resp. A.4) to Figure 5 (resp. A.3) is misleading the reader:
    * At first sight it looks like the performances are somehow related to the srank not shrinking
    * But at 50x62.5K steps the srank of CQL is higher than CQL+DR3 for Breakout for example.
    * Why the Average Return of Asterix is missing (Figure 6)?
    * Looking further at the results reported in Kumar et al. 2021, the authors are missing what seems like the interesting region w.r.t. their claims. It is only after more than 50x62.5K gradient steps that the performance drops appear. Why did the authors clipped the plots if the experiments were done for at least 200x62.5K gradient updates. This could help conclude about the training stability the authors claim reporting in the paper (e.g. p. 2 " giving rise to methods that train for longer, reach a better solution, and remain stable at this solution ").
    * Nevertheless I notice what looks like a significative improvement w.r.t. the average return
  * Same goes for Figure 4
    * The x-axis' scale for 1% Uniform Replay is different from the other ones, why? It looks like CQL+DR3 suffers a performance drop and that with more gradient step the performances with or without the explicit regularization would be similar.
    * Is that the case for the 1% Uniform Replay experiment at 200x62.5K gradient updates?
  * "we evaluate DR3 in conjunction with CQL on the harder D4RL domains (antmaze, kitchen domains)" (Section 6), "we use the harder AntMaze and Franka Kitchen domains for evaluating CQL, since these domains are challenging for CQL." (Appendix E.2)
    * W.r.t. which metric those domains are harder than the other ones? From Kumar et al. 2021: "These tasks are especially challenging, since they require composing parts of trajectories, precise long-horizon manipulation, and handling human-provided teleoperation data". In this case, does the potentially poor performances are correlated to the co-adaptive phenomenon? Reporting sranks after 2M gradient steps would help conclude.
    * While trying to put emphasis on "stability", what looks like the main focus of the paper, I was expecting the authors to focus on environments for which previous methods suffer from performance drop and the under-parametrization phenomenon such as Ant-v2, Hopper-v2, and Walker2d-v2 as shown in Kumar et al. 2021 or hopper-medium-v0 as shown in Kumar et al. 2020b
  * I do not understand why the effect of DR3 on D4RL is discussed using BRAC and not CQL nor REM. This would have had more impact since the paper mainly focus on the effect of DR3 w.r.t. CQL and REM.
  * Overall a rigorous statistical significance test, to highlight significatively better results in the tables, is missing

### Minor revisions/questions/remarks
  * Does light colored regions on the plots always refer to 95% CI?
  * Does the under-parametrization phenomenon and the drop in performances are specific to over-parametrized NNs?
  * What optimizer the authors are using to optimize their models? This could be interesting since the theory developed in the paper assume an SGD optimization scheme.
  * It would be better if all the algorithms are referenced in the main paper, e.g. COG has only its attached reference in the appendix (?)
  * Writing "This protocol is standard in Atari" seems a bit far-fetched when the references point only to Google related papers
  * What does bold format mean in the paper tables?

### Some typos
  * $\theta$ missing for some gradients
  * p. 12 "A eoretical comparison"
  * p. 13 "in ou practical"
  * p. 16 "(Equation'4)""
  * p. 22 "atleast"
  * p. 24 "wprked"

Authors answering remarks and questions during the rebuttal period might lead to a higher grade. But, as is, important claims of the paper are not well-supported and the way the results are presented is suspicious.

**Summary Of The Paper:**

The paper propose to study the under-parametrization/co-adaptation effect (observed while optimizing over-parametrized networks with SGD) arising in TD learning based RL algorithms which looks correlated to performance drops. They prove an implicit regularization effect, following previous works focussing on supervised learning, that could explained the co-adaptation phenomenon. They derive an explicit regularizer and show its effects on plethora of experiments.

**Summary Of The Review:**

The paper looks theoretically sound some questions remain regarding the experimental efficacy of the proposed regularizer claimed by the authors.

---

> ### Author Response · Authors · 2021-11-16
> **Author Response (Part 1 of 3): Summary of changes, Clarifying our goal and connection to srank, Added MuJoCo results**
>
> We thank the reviewer for their thorough feedback, which has helped improve the paper.
>
> It was not our intention to mislead the reader. We have added all experiments requested by the reviewer and provided explanations for the choices we made. In summary, the changes we made are as follows (changes in $\textcolor{magenta}{magenta}$):
>
> - To resolve the reviewer’s concerns regarding the performance of DR3, as per the reviewer’s suggestion, we have added statistical significance testing (**Appendix A.6**), and included new experiments showing the efficacy of DR3 on MuJoCo tasks from Kumar et al. 2021 (**Appendix A.3**), a detailed study of srank and DR3 regularization (**Appendix A.4**), cosine similarities of features in **Appendix A.8.3**, and per-game plots until 200 x 62.5k epochs (**Appendix F.1**).
>
> - We clarified the purpose of Figures 5 and 6, which correspond to two different explicit regularizers.
>
> - Finally, we have also clarified what we mean by stability and revised the paper to reflect this definition.
>
> In all, our results align with the original findings and support the efficacy of DR3. We are happy to provide additional explanations or conduct further experiments to resolve the reviewer's concerns. **Please let us know if this addresses your concerns.** We are happy to address any remaining concerns.
>
> ___
>
> > **Our goal in this paper; is the regularizer a way to increase rank, clarifications regarding srank:**
>
> We think that perhaps there is a misunderstanding that our method is simply trying to address the rank-collapse issue pointed out by Kumar et al. 2021. That is not the case -- while this prior work is related, the co-adaptation phenomenon we attempt to correct is not the same as fixing srank. Our goal is not to study if higher srank corresponds to higher performance and it may not (e.g., as the reviewer noted in CQL vs CQL + DR3 result on Breakout in Figure 5, the method with a higher srank doesn’t achieve better performance). All we wanted to show in Figure 5, and we have made this explicit in the paper now, is that DR3 prevents a collapse in the rank. This is different from the meaning that DR3 features have a high rank. We have clarified this in the paper now.
>
> **"DR3 explicit way to increase rank"?** : It is not immediately clear to us that minimizing the explicit dot-product regularizer ($R(\phi) = \sum_{s, a, s’, a’} \phi(s, a)^T \phi(s’, a’)$) would increase rank since $R(\phi)$ can be reduced, in principle, by simply rescaling the feature vectors for all $(s, a)$ pairs, while this does not change the rank of the features. Nothing prevents this rescaling from happening, but we find that while DR3 does typically also increase rank, it might decrease rank too as we empirically find. That said, we never found in our experiments, that the rank of DR3 features suddenly collapses.
>
> **In our experiments**, which we added in **Appendix A.4**, comparing the srank and performance indicate that while DR3 does increase srank for DQN and CQL, the **srank for REM + DR3 is smaller than that of REM**, and despite this, REM + DR3 attains a higher performance than REM. This does not contradict Kumar et al. 2021’s findings, because they found that a low value of srank may correspond to poor performance, but performance may still be poor when srank is high. Therefore, *high srank may not be correlated with higher performance and srank may not be a good metric to indicate the hardness of any given task*.
>
> ___
>
> > **MuJoCo domains from Kumar et al. 2021.**
>
> To address the reviewer’s concern, we have now compared CQL and CQL+DR3 on the Hopper-v2, Ant-v2, and Walker2d-v2 domains from Kumar et al. 2021 in **Appendix A.3**. As shown, while CQL suffers from a performance collapse on Walker2d and Ant, and oscillates on Hopper, CQL + DR3 attains a higher performance value, which is stable. The most notable difference is on Ant-v2, where adding DR3 improves performance from random (0.0) for base CQL to better than an online SAC expert ($\geq 100.0$) for CQL + DR3. This means that DR3 is also effective on these MuJoCo tasks.

---

> > ### Author Response · Authors · 2021-11-16
> > **Author Response (Part 2 of 3): Statistically significant results, Addressed concerns regarding Figure 5 and 6, arXiv papers, 1% replay**
> >
> > > **Statistical significance tests:**
> >
> > To add statistical reliability to our reported results, we have updated the paper to report the average probability of improvement with 95% confidence intervals (CIs), as recommended recently by Agarwal et al. NeurIPS 2021 [1] (and follows recommendations of works published in statistical journals [2, 3]), on AntMaze, Kitchen in Appendix A.6 to assess the probability that adding DR3 improves over the baseline algorithm. In summary, we find that DR3 **does** improve over the baseline algorithm in a statistically significant manner.
> >
> > The probability of improvement shown in **Appendix A.6** is computed using the [Mann-Whitney U-statistic](https://en.wikipedia.org/wiki/Mann%E2%80%93Whitney_U_test), averaged across tasks. Adding DR3 improves over CQL with probability **0.83** with 95% CI (0.7, 0.96) on AntMaze, with probability **0.8** with 95% CI (0.6, 1.0) on Kitchen. Since the CIs do not contain 0.5, these results are **statistically significant** (p-value < 0.05). Also, since the mean probabilities are larger than 0.75, our results are **statistically meaningful** as per the Neyman-Pearson statistical testing criterion in [4]. Furthermore, the statistical significance of the aggregate interquartile mean (IQM) results on Atari datasets can be seen from the fact that there is no overlap in the 95% confidence intervals in Table 1.
> >
> > [1] Agarwal et al. 2021. Deep reinforcement learning at the edge of the statistical precipice. NeurIPS 2021.
> >
> > [2] Wasserstein, R et al. 2019. Moving to a world beyond “p< 0.05”. The American Statistician.
> >
> > [3]  Amrhein V et al. 2019.  Scientists rise up against statistical significance.  Nature.
> >
> > [4] Bouthillier, Xavier, et al. "Accounting for variance in machine learning benchmarks." Proceedings of Machine Learning and Systems 3 (2021)
> >
> > ___
> >
> > > **Regarding missing the interesting region in Figure 5 and Figure 6**
> >
> > We would first like to clarify that Figure 5 and Figure 6 correspond to different experiment setups -- Figure 5 plots the $\text{srank}$ values for the practical DR3 regularizer shown in Equation 7. Figure 6 compares the performance of the DR3 regularizer for two choices of $M$ (label-noise, studied in prior work Blanc et al. 2020 and $M = I$, which is our practical approach). The performance plot for our main practical method averaged over all games is already shown for **200 units** on the x-axis in Figure 4, and we have now added per-game srank and performance plots in **Appendix F.1**. These plots must be used for the srank comparison, not the plots in Figure 6 since Figure 6 corresponds to a different regularizer which was not used to compute srank in Figure 5.
> >
> > **x-axis in Figure 6.**  The label-noise regularizer is expensive to compute (8 days for running 50 x 62.5k gradient steps) since we run 20 fixed point iterations for computing $\Sigma_M^*$ (see the statement of Theorem 1) for every gradient update on the Q-function. Additionally, please note that the point of Figure 6 is to show that our simplifying choice of $M$ does not degrade performance and that the label noise version of $M$ also improves over the base algorithm, and this is apparent from the plot in Figure 6. That said, we can run it for longer and put the results in the final version. The average return of Asterix in Figure 6 is provided in Appendix A.3, Figure A.7; we didn’t include it in the main paper due to space reasons, but we are happy to do this in the final.
> >
> > ___
> >
> > > **Roughly 3/5 papers are arXiv papers**
> >
> > Thank you for pointing this out. It was due to the bibliography entries referring to the arXiv versions of peer-reviewed papers, and we have now updated the References to point to the conference/journal version of the papers. Now, **less than 8 of all 52 references** point to arXiv links.
> >
> > ___
> >
> > > **1% uniform replay plots**
> >
> > Our results were not clipped for the 1% dataset as we only evaluated 100 iterations. [Agarwal et al. 2020](https://arxiv.org/abs/1907.04543) reported results on 1% replay using only 20 iterations (1 iteration = 62.5k gradient updates) and 100 iterations for 10% initial replay data,  while [Kumar et al. 2021](https://openreview.net/forum?id=O9bnihsFfXU) use 100 iterations for 5% uniform replay. Instead, we ran **5x** more gradient updates on the 1% dataset and **2x** more updates on the 5% and 10% datasets compared to prior work on this data.
> >
> > That said, we launched a new experiment to report results for REM + DR3 compared to REM on 1% replay dataset for 200 iterations in **Figure A.10**. Note that the performance of REM + DR3 after around 120 iterations (= thrice as long as when the peak in REM appears) is still better than the peak in REM. Additionally, we have launched another experiment to evaluate DR3 +CQL on 200 iterations on 1% data and will update the reviewer if the runs finish in time.

---

> > > ### Author Response · Authors · 2021-11-16
> > > **Author Response (Part 3): Training stability, D4RL and Minor Points**
> > >
> > > > **Training stability of DR3 (“giving rise to methods that train for longer, reach a better solution and remain stable at the better solution”).**
> > >
> > > We did not intend to claim that our approach solves all stability changes in RL, and we've revised the claims in the introduction and elsewhere accordingly to clearly state that our method improves stability and performance relative to prior methods, rather than stating unequivocally that it is "stable." We believe that this better reflects our results, where we measure stability via our proposed average performance metric and show that DR3 improves the results according to this metric over the baselines. As with all prior methods, we do notice degradation in performance with more training even with DR3 in some cases. However, DR3 maintains higher performance throughout training and in a large number of games, the final performance of DR3 is also above the peak in the baseline offline RL method (for example, in per-game plots, Appendix F.1).
> > >
> > > ___
> > >
> > > > **Results on D4RL domains.**
> > >
> > > We have removed the qualitative statements that the Antmaze and Kitchen domains are harder since that is not related to the paper. To clarify, we used these domains, since CQL does collapse on these domains with more training. Additionally results in Table 2 + significance test in Appendix A.6. indicate that adding DR3 improves performance on these domains in a statistically significant manner.
> > >
> > > We only studied BRAC on the D4RL Gym-MuJoCo domains, since we found that the CQL algorithm didn’t suffer from any instabilities over the course of training on the Gym-MuJoCo domains in D4RL. Recent papers (e.g., [Kostrikov et al. 2021](https://arxiv.org/abs/2110.06169)) have also reported good performance of CQL (compared to many algorithms) on these tasks, and therefore we did not use these. If the reviewer suggests, we can run experiments on these domains as well and add them to the paper.
> > >
> > > REM performs poorly on D4RL MuJoco domains (see results in [Fu et al. 2020](https://openreview.net/forum?id=px0-N3_KjA), Table 2, cREM) since it does not handle overestimation at out-of-distribution actions. Since DR3 does not correct OOD actions, we don’t expect it to help improve performance on REM significantly on D4RL and so we didn't explore REM.
> > >
> > > ___
> > >
> > > > **Minor Points**
> > >
> > > 1. Light-colored regions refer to one standard deviation on each side of mean in our non-Atari plots which corresponds to 95% CI under the assumption of a Gaussian random variable.
> > >
> > > 2. **Regarding overparameterized NNs and under-parameterization:** Our theory does apply to overparameterized neural networks and does not apply to, for instance, simple linear Q-functions that are not overparameterized.
> > >
> > > 3. **Optimizer used:** Following guidelines of recent papers [Viellard et al. NeurIPS 2020](https://proceedings.neurips.cc//paper/2020/file/2c6a0bae0f071cbbf0bb3d5b11d90a82-Paper.pdf), [Agarwal et al. ICML 2020](https://arxiv.org/abs/1907.04543), [Agarwal et al. NeurIPS 2021](https://openreview.net/forum?id=uqv8-U4lKBe), we utilized DQN (Adam) for our experiments since this version of DQN attains higher performance. In our early experiments, we did measure feature dot products with DQN (RMSprop) (i.e., Nature DQN) and found a similar increase with this RMSProp optimizer.
> > >
> > > 4. **Protocol is standard in Atari:** We referred to the protocol of tuning to find a single hyperparameter on a limited set of games, which has been used in several papers starting from the DQN (Nature) [Mnih et al. 2014] paper that ran DQN on Atari domains on the first time, and this protocol has been adopted by several follow-ups.
> > >
> > > 5. **Typos** We have fixed the typos in the paper.
> > >
> > > -------------------------------------------------
> > > *We would appreciate it if the reviewer can confirm that their concerns had been addressed and, if so, reconsider their assessment. We’d be happy to engage in further discussions.*

---

> > > > ### Comment · Reviewer_5pgu · 2021-11-19
> > > > **Reviewer response**
> > > >
> > > > I thank the authors for the detailed answers and the revised paper.
> > > >
> > > > ### Some more questions/remarks:
> > > >
> > > >   * I still find the observations of the paper somehow misleading and I would like to see more coherence in the paper:
> > > >       * The authors start with some observations on the dot-product similarity (Figure 2);
> > > >       * The authors correlate the increase of the dot-product similarity with the drop of performance of batch RL algorithms "Why implicit regularization [...] policy performance?" paragraph (page 6);
> > > >       * The authors do not show if, with their explicit regularizer, the dot product similarity does not increase (I can't find the dot-product similarity DR3 for these examples);
> > > >       * The authors plot the average return on Asterix and Breakout starting from a good solution achieved using DR3 (the peak for both DQN+DR3 (Figure A.5) and CQL+DR3 (Figure F.1));
> > > >       * The authors emphasis on the return drop while doing gradient updates with the original algorithms (without DR3);
> > > >       * But we would have observe the same drop and the same final performances even using DR3;
> > > >       * A quick fix would be to add CQL+DR3 and DQN+DR3 starting from the same "good initialization".
> > > >
> > > >   * Looking at Figure 2 (right plots), I cannot conclude about the "non-increasing trend", can you normalize the loss with the initial one (i.e. the one with no gradient update yet)?
> > > >
> > > >   * I would be happy to see in appendices the actual running time to better raise the community's awareness about the actual time cost of optimizing and validating such models (with a table or a footnote as the authors did page 18).

---

> > > > > ### Author Response · Authors · 2021-11-20
> > > > > **Author Response: Added plot for DR3 in the setting of Figure 2**
> > > > >
> > > > > Thanks for your reply! We answer the questions raised below:
> > > > >
> > > > > >  **A quick fix would be to add CQL+DR3 and DQN+DR3 starting from the same "good initialization".**
> > > > >
> > > > > We have added this plot to **Appendix A.9** and we observe that the curve for CQL+DR3 maintains a higher return than CQL. For example, while the return for CQL drops from ~5000 to ~3200 on Asterix, when DR3 is used, the return only moves from ~5000 to ~4300. Similarly, dot product similarity is significantly smaller when using DR3 in this experiment (**Figure A.15**), unlike the base CQL algorithm. We are running DQN + DR3 for the setting of Figure 2 (starting from the initial checkpoint that was used for the DQN run in Figure 2)  and will get back to you about that.
> > > > >
> > > > > As stated in our previous response and in the revised paper, we would again emphasize that DR3 improves stability and performance relative to prior methods, and we are not unequivocally saying that it is "stable." The figure in Appendix A.9 supports the relative stability of CQL + DR3 over CQL when starting from a good initial solution.
> > > > >
> > > > >
> > > > > ___
> > > > >
> > > > >
> > > > > > **The authors do not show if, with their explicit regularizer, the dot product similarity does not increase**
> > > > >
> > > > > The dot product similarity figures with DR3 penalty are already present in Figure A.3 (middle) for MuJoCo tasks from Kumar et al. (2021), REM with and without DR3 on Atari in Figure A.7. However, to fully address your suggestion, we have now added per-game dot product similarity curves (*note that these are in log-scale*) for all 17 games on CQL/REM on 5% Atari datasets in **Appendix F.2**. All these figures show that dot product similarity is significantly smaller with the DR3 regularizer compared to the base offline RL algorithm.
> > > > >
> > > > > ___
> > > > >
> > > > > > **Looking at Figure 2 (right plots), I cannot conclude about the "non-increasing trend", can you normalize the loss with the initial one (i.e. the one with no gradient update yet)?**
> > > > >
> > > > > The intention of this statement of “generally exhibit non-increasing trend” was to simply state that training error remains relatively small while performance degrades and dot product similarities rise significantly. Since the statement about trend created confusion, we have revised the paper to explicitly state that training error remains relatively small.
> > > > >
> > > > > > **I would be happy to see in appendices the actual running time to better raise the community's awareness about the actual time cost of optimizing and validating such models (with a table or a footnote as the authors did on page 18).**
> > > > >
> > > > > We have added this to Appendix E.4 now.
> > > > >
> > > > > _____
> > > > >
> > > > > *Please let us know if this addresses your concerns, and we are happy to clarify any other concerns that might be remaining.*

---

> > > > > > ### Comment · Reviewer_5pgu · 2021-11-20
> > > > > > **Reviewer response**
> > > > > >
> > > > > > I thank the authors for the detailed answers and the revised paper.
> > > > > >
> > > > > > Last question: do the authors have an explanation why the trend observed in Figure A.15 is different from the one observed in Figure F.1 for CQL+DR3 on Asterix?
> > > > > >
> > > > > > Overall, I am quite impressed by the rebuttal!

---

> > > > > > > ### Author Response · Authors · 2021-11-21
> > > > > > > **Author Response**
> > > > > > >
> > > > > > > Thanks for your reply and we are glad that you are quite impressed with the rebuttal! Your questions have helped us improve the paper.
> > > > > > >
> > > > > > > To answer your question regarding the learning trends of CQL + DR3, there is a key difference in the run in Appendix F.1 and Figure A.15: when starting optimization in Figure A.15, we did not retain any gradient mean and variance parameters from Adam and only initialize the parameters of the Q-function. This was done to make sure that no information about how this initial Q-function is obtained confounds the results of this plot. This way, we can also make sure that CQL and CQL + DR3 are comparable to each other in Figure A.15. But this means that the learning trends of CQL+DR3 in Figure A.15 and Appendix F.1 are not comparable to each other.  We have updated Appendix A.9 to add this detail.

---

> > > > > > > > ### Comment · Reviewer_5pgu · 2021-11-21
> > > > > > > > **Reviewer response**
> > > > > > > >
> > > > > > > > It is interesting that, zeroing the moments of Adam at the peak alleviate the unlearning effect.
> > > > > > > >
> > > > > > > > I thank the authors for the discussion, I have updated my score to 6.

---

### Official Review · Reviewer_7Brr · 2021-11-02

**Correctness:** 3
**Technical Novelty And Significance:** 3
**Empirical Novelty And Significance:** 3
**Recommendation:** 6
**Confidence:** 4

**Main Review:**

The paper is overall well-written and clear. The method proposed is simple, quite clearly motivated and the claims are quite well supported by the experiments. Here are a few elements/questions that might need some clarifications:
- the dot products can increase either because of an increased co-adaptation or because the magnitude of the vector of features considered increases. How can be sure that it is due to a co-adaptation and not because of an increase in the magnitude of the features?
- In Fig 3, 5% and 25% of the data used by Singh et al. (2020) are used to make these more challenging. Would it also be useful to provide the results for the case of 100% of the data? It seems that even when data is relatively abundant, it would be interesting to report the performance with and without the regularizer. The same comment applied for Figure where a plot with 100% of the data could also be of interest.
- The authors mention that they will release scores for individual runs as well as open-source their code. Is it possible to have access to that during this rebuttal period?

Minor remarks:
- typo: "(...) and the the dot-product (...)"

**Summary Of The Paper:**

This paper discusses how the implicit regularization effect of SGD could be harmful in the offline deep RL setting, due to degenerate feature representations, aliasing the representations for state-action pairs that appear on either side of the Bellman backup. To address this issue, the paper proposes a simple explicit regularizer (DR3) that counteracts the undesirable effects of this implicit regularizer. When combined with existing offline RL methods, DR3 has strong performance and stability (Atari 2600 games, D4RL domains and robotic manipulation from images).

**Summary Of The Review:**

Paper that investigates a simple regularization technique for off-policy value function learning in deep RL. The idea is quite well motivated by showing a few insightful values through learning (not only the score). The empirical results are quite convincing, even though they could be more complete.

---

> ### Author Response · Authors · 2021-11-16
> **Author Response: Added clarification of dot products, released individual runs and code snippets for DR3 in Jax**
>
> We thank the reviewer for their constructive feedback. We apologize for the delay in responding. We have updated the paper to incorporate the suggestions and changes are shown in $\textcolor{magenta}{magenta}$. We provide clarifications to their questions below and have added a new section to discuss the relationship between co-adaptation, dot products, and normalized similarities (i.e., cosine similarities) [**Appendix A.8.3**]. Furthermore, in response to other reviewers' comments, we have added results for statistical significance (**Appendix A.6**), new experiments on MuJoCo domains (**Appendix A.3**), and further analysis of the relationship between srank and DR3 (**Appendix A.4**), which further supports our claim regarding the efficacy of DR3.
>
> **Please let us know if our answers below address your concerns and if there are any concerns remaining.** We are happy to clarify further.
>
> > **Clarification regarding co-adaptation, dot-products, and aliasing**
>
> We would like to clarify that the term co-adaptation refers to a high value of dot-products, rather than any normalized measure of similarity such as cosine similarity, throughout the paper. Proposition 3.2 shows that even in a linear function approximation setting, high dot-products can lead to bad effects, whether it be due to high alignment of vectors $\phi(s, a)$ and $\phi(s’, a’)$ as measured by cosine similarity or due to high norms. We have updated the paper to indicate that the word “aliasing” is used informally, and replaced it by the word ‘’co-adaptation’’ to clear the confusion.
>
> Further, to understand the contribution of norms and directional alignment (i.e., cosine similarity) in co-adaptation, we have added a new empirical analysis in Appendix. Specifically, we find that the cosine similarities on Atari games and MuJoCo domains (**Appendix A.8.3**) are very high, close to 1 throughout training, and are not indicative of performance degradation on these tasks. However, feature dot-products are continuously increasing and the addition of DR3 improves performance while decreasing both cosine similarity and dot products. We also found a case where the cosine similarities for the base offline RL algorithm (CQL, **Figure A.14**) are decreasing, but the performance is oscillating (**Figure A.3**), indicating that cosine similarities or normalized alignment may not convey the full picture. Therefore, overall, we believe that both norms and directional alignment, combined together, contribute to the co-adaptation issue.
>
> ___
>
> > **individual runs and open-source code.**
>
> We have released the individual runs for Atari in this [anonymous folder](https://drive.google.com/drive/folders/1OVfOue-h3COsZJ1lW3Vca1awVVn_KWL6?usp=sharing). Furthermore, implementing DR3 regularizer is a few lines of code and we show this in an anonymous [python notebook](https://colab.research.google.com/drive/16lrgPQopSXsx3HWGpHsIV5QJBxYIllgs#scrollTo=cBKUmhxj_20V) for a barebones JAX implementation of DR3 loss on a DQN network. That said, our full code is tied to a complex, currently non-public infrastructure, which requires a major clean-up. We’ll make sure to open-source the code for the final version, as we mentioned in the paper.
>
> ___
>
> > **Regarding 100% of the COG data**
>
> Thanks for the suggestion! We have launched the experiments with 100% of the data for Figure 5, which requires more than a week of training with one run per GPU. We will report back here by the end of the rebuttal period if our runs finish. We speculate that in such a scenario with such a broad coverage offline dataset, out-of-sample actions may be less likely.
>
> -------------------------------------------------
> We would appreciate it if the reviewer can confirm that their concerns had been addressed and, if so, reconsider their assessment. We’d be happy to engage in further discussions.

---

> > ### Comment · Reviewer_7Brr · 2021-11-19
> > **thanks for the clarification**
> >
> > Thanks for the clarifications. Even though some points can be improved further in the paper (e.g. some experiments not shown in specific settings), I believe that the paper investigates an interesting question and provides valuable insights. I update my score to 6.

---

### Official Review · Reviewer_T3ZN · 2021-11-03

**Correctness:** 3
**Technical Novelty And Significance:** 4
**Empirical Novelty And Significance:** 2
**Recommendation:** 8
**Confidence:** 4

**Main Review:**

Strengths
---------

-   Interesting theoretical analysis, while somewhat compromised to make
    the algorithm tractable, still provides insight into the problem
    addressed. It might be good to further show evidence that the noise
    model is well-justified in the RL regime, which would justify the
    leap from theory to empirics.
-   Empirical analysis is comprehensive, clearly demonstrating the
    problem and algorithms stability when the problem is addressed
    (through the proposed regularizer).

Weaknesses
----------

-   While the empirical analysis is comprehensive, the performance of
    the algorithms with the DR3 regularizer in the experiments are not
    completely compelling. Although the mean performance seems better,
    the statistical significance is not there in Figure 3 and Table 2
    (and Figure 6, Seaquest). The results in Atari across all games are
    more impressive, but the aggregation and normalization across games
    can hide idiosyncrasies.

Detailed Comments
-----------------

-   Section 3, Experimental setup: For TD-Learning, it says an action is
    sampled. Shouldn't this be an expectation over the policy (and hence
    use every action, without sampling?). If all the actions are
    included in the target through an expectation, the problem of
    divergent action-values should be less pronounced if not absent. The
    problem seems to be that some action-values are left dangling (i.e.
    the action-values of actions that are not sampled). By including
    them all through an expectation, any problematic danging
    action-values should be corrected immediately. The problem remains
    for Q-learning, but further analysis needs to ablate maximization
    bias and other confounders.

-   Sec 3.1, Paragraph 2: What is meant by learned behavior policy?
    Shouldn't the behavior policy be given or fixed?

-   I think there is a leap in reasoning by connecting states aliasing
    and high dot-products. Dot products can increase between two vectors
    without any aliasing. If the magnitude of both vectors increase then
    the dot product will increase without any change in the angle
    between the vectors.

-   Increasing dissimilarity between successive states is intuitively
    reasonable, but this can also be a hindrance if the successive
    states are indeed similar. Is there any way to account for this?

-   Shouldn't a target network prevent the representations from being
    similar? The networks are different and hence the representations
    will be a few gradient steps away.

-   Section 2 Equation 1: While we do take the gradient of this
    expression, it is not correct to say that this is a proper
    optimization objective. It is not minimized in practice (and often
    diverges). In addition, methods that actually minimize this
    objective, such as residual-gradient, do not result in a good
    control policy.

-   Figure 2: The left and center plots communicate the issue clearly,
    except that the training loss plot on the right is not reflective of
    the problem. As you point out, bootstrapping algorithms do not
    minimize any objective function as the gradient does not belong to
    any objective function.

**Summary Of The Paper:**

This paper provides empirical and theoretical evidence that value-based
methods that optimize TD errors with SGD have an implicit "regularizer" that
increases the dot-product of the representation at successive states.
The theoretical analysis shows that this can be offset by a term that
penalizes large dot products, which motivates the proposed regularizer
(DR3). The paper shows empirically that these large dot products are a
source of divergence and that training with the DR3 regularizer can
stabilize learning.


**Summary Of The Review:**

Overall, the paper raises and simultaneously addresses an interesting
problem for RL algorithms that employ function approximation. The paper
is somewhat held back by the empirical analysis, specifically in the
lack of statistical significance in several of their experiments. I am
also not sure of the correctness of the intuition that the dot-products
result in aliasing in reinforcement learning. The paper is still above
the acceptance threshold, but I hope that my comments below contribute
to improving the paper.


Edit: After discussion with the author and reading the back-and-forth between the authors and reviewer 5pgu, I have increased my score from 5 -> 8.

---

> ### Author Response · Authors · 2021-11-16
> **Author Response (Part 1 of 2): Added Statistical Significance, Clarifications in Section 3**
>
> We thank the reviewer for their constructive feedback and for a positive assessment of our work. We have updated the paper in $\textcolor{magenta}{magenta}$ to address their concerns. Specifically, we added significance testing (**Appendix A.6**) and found that DR3 improves over CQL in a statistically significant manner, added diagnostic experiments to understand the effect of target update frequency (**Appendix A.8.1**), and compared co-adaptation in TD-learning and SARSA (**Appendix A.8.2.**). We have also updated the paper to answer various questions.
>
> > **Statistical significance tests:**
>
> To add statistical reliability to our reported results, we have updated the paper to report the average probability of improvement with 95% confidence intervals (CIs)[2,3], as recommended recently by Agarwal et al. NeurIPS 2021 [1] on AntMaze, Kitchen domains from Table 2 in Appendix A.6 to assess the probability that adding DR3 improves over the baseline algorithm. In summary, we find that DR3 ***does*** improve over the baseline algorithm in a statistically significant manner.
>
> The probability of improvement in Appendix A.6. is computed using the [Mann-Whitney U-statistic](https://en.wikipedia.org/wiki/Mann%E2%80%93Whitney_U_test), averaged across tasks. Adding DR3 improves over CQL with probability **0.83** with 95% CI (0.7, 0.96) on AntMaze, with probability **0.8** with 95% CI (0.6, 1.0) on Kitchen. Since the CIs do not contain 0.5, these results are **statistically significant** (p-value < 0.05). Also, since the mean probabilities are larger than 0.75, our results are **statistically meaningful** as per the Neyman-Pearson statistical testing criterion in [4]. Furthermore, statistical significance of the aggregate interquartile mean (IQM) results on Atari datasets can be seen from the fact that there is no overlap in the 95% confidence intervals in Table 1.
>
> [1] Agarwal et al. 2021. Deep reinforcement learning at the edge of the statistical precipice. NeurIPS 2021.
>
> [2] Wasserstein, R et al. 2019. Moving to a world beyond “p< 0.05”. The American Statistician.
>
> [3]  Amrhein V et al. 2019.  Scientists rise up against statistical significance.  Nature.
>
> [4] Bouthillier, Xavier, et al. "Accounting for variance in machine learning benchmarks." Proceedings of Machine Learning and Systems 3 (2021)
>
> **Regarding Figure 6**, we would like to clarify that our primary goal is not to show that CQL + DR3 (red) improves over CQL + DR3 (label noise) (blue), but rather that they are comparable and that the variant of DR3 with label noise also improves over the base CQL (green) and DQN (pink) algorithms. The 95% confidence intervals marked in the figure with DR3 (DQN + DR3 (label noise) and CQL + DR3 (label noise)) do not overlap with that of the base algorithm, indicating statistical significance. If the reviewer suggests, we can add concrete metrics in the final.
>
> ___
>
> We now respond to the detailed comments raised by the reviewer:
>
> > **Section 3, Experiment Setup:**
>
> We apologize for the confusion in this statement and have updated the paper to make this clear. Our implementation of TD-learning computes an **exact** expectation under the behavior policy for computing the target value: $\text{TD-target}(s, a) = r(s, a) + \sum_{a} \pi_\beta(a’|s’) Q_\text{target}(s’, a’)$. While all actions at the next state are used for computing the backup target, note that not all actions for every state $s$ are observed in the dataset $\mathcal{D}$. And therefore the target value for TD-learning does utilize Q-value predictions at out-of-sample (i.e., actions $a’$ such that $(s’, a’) \notin \mathcal{D}$) for computing Bellman targets. These are actions for which the Q-values are never explicitly trained since they do not appear in the dataset. Please let us know if the setup is more clear, we are happy to clarify further.
>
> > **Section 3, other confounding effects (‘’maximization bias’’, ‘’learned behavior policy’’).**
>
> The behavior policy is fixed, but we do not have direct access to the functional form of the behavior policy in our experiments, since it is the mixture policy induced by all the intermediate policy iterates of a run of an online DQN algorithm, and intermediate policy checkpoints are not available with the DQN-Replay dataset. Therefore, we trained a model of the behavior policy on the dataset via supervised learning and used it for TD learning. We have clarified this in the paper now.
>
> Additionally, to remove any potential confounding, we have added a comparative evaluation of the feature dot products (feature co-adaptation) on TD-learning and SARSA where the behavior policy is fully known (**Appendix A.8.2**). This experiment was performed on two sparse-reward gridworld domains from Fu et al. 2019. As shown in **Figure A.12** in the revised paper, we find a similar trend of increasing dot products with TD-learning even when the behavior policy is fully known and a relatively stable trend in dot products for SARSA.

---

> > ### Author Response · Authors · 2021-11-16
> > **Author Response (Part 2): Added clarification for dot product, added experiment to test slow target networks**
> >
> > > **Clarification regarding co-adaptation, dot-products, and aliasing**
> >
> > We would like to clarify that the term co-adaptation refers to a high value of dot-products throughout the paper. Proposition 3.2 indicates that, even in a linear setting, high dot-products can lead to bad effects, whether it be due to high alignment of vectors $\phi(s, a)$ and $\phi(s’, a’)$ or due to high norms. And a high-value dot product is what we call co-adaptation. We have updated the paper to indicate that the word “aliasing” is used informally, and replaced it with the word ‘’co-adaptation’’ to clear the confusion.
> >
> > Further, to understand the contribution of norms and directional alignment (i.e., cosine similarity) in co-adaptation, we have added a new empirical analysis in Appendix. Specifically, we find that the cosine similarities on Atari games and MuJoCo domains (**Appendix A.8.3**) are very high, close to 1 throughout training, and are not indicative of performance degradation on these tasks. However, feature dot-products are continuously increasing and the addition of DR3 improves performance while decreasing both cosine similarity and dot products. We also found a case where the cosine similarities for the base offline RL algorithm (CQL, **Figure A.14**) are decreasing, but the performance is oscillating (**Figure A.3**), indicating that cosine similarities or normalized alignment may not convey the full picture. Therefore, overall, we believe that both norms and directional alignment, combined together, contribute to the co-adaptation issue.
> >
> > ___
> >
> > > **Target networks prevent co-adaptation.**
> >
> > We agree with the reviewer’s intuition that fixing the target network might alleviate co-adaptation, although modern deep RL algorithms do not use this procedure. To empirically investigate this hypothesis, we ran an experiment on a gridworld domain from [Fu et al. ICML 2019](https://arxiv.org/abs/1902.10250), which we have now added to **Appendix A.8.1**. We  aimed at understanding how the trend in feature dot-products for Q-learning changes as a function of the delay in the target update ($N$.) As shown in **Figure A.11**, we find that while slowing down the target update from $N=5$ to $N=10$ does reduce co-adaptation, but slowing down the target update further may increase it (compare $N=10$ vs $N=200$ or $N=500$). This indicates that there are likely other factors that affect the co-adaptation issue and just delaying the target update rate may not fully alleviate it.
> >
> > ___
> >
> > > **Increasing dissimilarity might hurt in some domains.**
> >
> > We agree with the reviewer that reducing dissimilarity is good generally, but it might hurt when done too much in certain domains. In some sense, this is true for any form of regularization, even with regularization schemes in supervised learning. Perhaps an interesting avenue of future work to solve this is by devising cross-validation schemes to identify the right amount of similarity and devising such schemes is a subject of future work.
> >
> >
> > ----------------------------------------------------
> > *We would appreciate it if the reviewer can confirm that their concerns had been addressed and, if so, reconsider their assessment. We’d be happy to engage in further discussions.*

---

> > > ### Comment · Reviewer_T3ZN · 2021-11-18
> > > **Thanks for the thorough response!**
> > >
> > > Thanks for the thorough response! Indeed, many of my concerns were addressed and I enjoy the discussion / speculation regarding my comments. I still think the empirical results are an overall weakness of the *method* but I do not think it should hold the paper back from being accepted. I will continue to monitor the discussion with other reviewers.
> > >
> > > It is almost unfortunate that this paper brings up more questions than it answers. For example, the quote:
> > >
> > > "As shown in Figure A.11, we find that while slowing down the target update from  to  does reduce co-adaptation, but slowing down the target update further may increase it (compare  vs  or ). This indicates that there are likely other factors that affect the co-adaptation issue and just delaying the target update rate may not fully alleviate it."
> > >
> > > really demonstrates the lack of thorough understanding of deep RL algorithms across the literature. This is a very interesting phenemonon and I appreciate it's inclusion.

---

### Author Response · Authors · 2021-11-17
**Summary of Changes So Far**

We thank the reviewers for their detailed feedback which has helped in improving the paper. We have updated the paper to address various of the reviewers’ comments, and the changes are shown in $\textcolor{magenta}{magenta}$. In this summary post, we want to list the main updates we have made to the paper.

1. [**Appendix A.3**] Added experiments on MuJoCo domains (Ant, Hopper, Walker2d) to show efficacy of DR3 in preventing unlearning and poor performance


2. [**Appendix A.6**] Added statistical significance tests for DR3 on AntMaze and Kitchen and found that CQL + DR3 improves over CQL both significantly and meaningfully.


3. [**Appendix A.4**] Added plots showing that srank does ont collapse with DR3 even when applied on DQN and REM. Also added more empirical analysis.


4. [**Appendix F.1**] Added per-game learning curves until 200 iterations (200 $\times$ 62.5k gradient steps)


5. [**Appendix A.8.1**] Added diagnostic experiment to understand the effect of target update frequency on co-adaptation

6. [‘**Appendix A.8.2**] Added more empirical evidence comparing TD learning and SARSA from FIgure 3 on gridworld domains.


7. [**Appendix A.8.3**] Added experiments and discussion of how feature co-adaptation relates to normalized feature similarities


8. [**Appendix A.7**] Added plot running REM on 1% data for twice as long untill 200 iterations.

Additionally, we have made several changes to the paper, revised unclear statements and definitions, released individual runs for each game (as requested by Reviewer 7Brr) and anonymously open-sourced a sample code snippet to run DR3 (as requested by Reviewer 7Brr).

We would appreciate it if the reviewers can check the updates and our responses and tell us if they address their concerns.

---

### Decision · Program_Chairs · 2022-01-20

**Decision:**

Accept (Spotlight)

**Comment:**

The paper proposes an interesting hypothesis about deep nets' generalization behavior inside RL methods: it suggests that the nets' implicit regularization favors a particular form of degeneracy, in which there is excessive aliasing of state-action pairs that tend to co-occur. It proposes a new regularizer to mitigate this problem. It evaluates the hypothesis and the regularizer empirically, and it provides suggestive derivations to motivate both.

The reviewers praised the comprehensive empirical analysis, the insights into learning, and the combination of empirical and theoretical evidence. The authors participated responsively and helpfully in the discussion period, and addressed any concerns raised by the reviewers.

This is a strong paper: it derives and motivates a novel hypothesis about an important problem, and analyzes this hypothesis both mathematically and experimentally.